# MMD-Regularized Unbalanced Optimal Transport

**Piyushi Manupriya**                                          *cs18m20p100002@iith.ac.in*
*Department of Computer Science and Engineering, IIT Hyderabad, INDIA.*

**J. SakethaNath**                                             *saketha@cse.iith.ac.in*
*Department of Computer Science and Engineering, IIT Hyderabad, INDIA.*

**Pratik Jawanpuria**                                          *pratik.jawanpuria@microsoft.com*
*Microsoft, INDIA.*

**Reviewed on OpenReview:** *https://openreview.net/forum?id=eN9CjU3h1b*

## Abstract

We study the unbalanced optimal transport (UOT) problem, where the marginal constraints are enforced using Maximum Mean Discrepancy (MMD) regularization. Our work is motivated by the observation that the literature on UOT is focused on regularization based on $\phi$-divergence (e.g., KL divergence). Despite the popularity of MMD, its role as a regularizer in the context of UOT seems less understood. We begin by deriving a specific dual of MMD-regularized UOT (MMD-UOT), which helps us prove several useful properties. One interesting outcome of this duality result is that MMD-UOT induces novel metrics, which not only lift the ground metric like the Wasserstein but are also sample-wise efficient to estimate like the MMD. Further, for real-world applications involving non-discrete measures, we present an estimator for the transport plan that is supported only on the given ($m$) samples. Under certain conditions, we prove that the estimation error with this finitely-supported transport plan is also $\mathcal{O}(1/\sqrt{m})$. As far as we know, such error bounds that are free from the curse of dimensionality are not known for $\phi$-divergence regularized UOT. Finally, we discuss how the proposed estimator can be computed efficiently using accelerated gradient descent. Our experiments show that MMD-UOT consistently outperforms popular baselines, including KL-regularized UOT and MMD, in diverse machine learning applications.

## 1 Introduction

Optimal transport (OT) is a popular tool for comparing probability measures while incorporating geometry over their support. OT has witnessed a lot of success in machine learning applications (Peyré & Cuturi, 2019), where distributions play a central role. The Kantorovich's formulation for OT aims to find an optimal plan for the transport of mass between the source and the target distributions that incurs the least expected cost of transportation. While classical OT strictly enforces the marginals of the transport plan to be the source and target, one would want to relax this constraint when the measures are noisy (Frogner et al., 2015) or when the source and target are un-normalized (Chizat, 2017; Liero et al., 2018). Unbalanced optimal transport (UOT) (Chizat, 2017), a variant of OT, is employed in such cases, which performs a regularization-based soft-matching of the transport plan's marginals with the source and the target distributions.

Unbalanced optimal transport with Kullback Leibler (KL) divergence and, in general, with $\phi$-divergence (Csiszar, 1967) based regularization is well-explored in literature (Liero et al., 2016; 2018). Entropy regularized UOT with KL divergence (Chizat et al., 2017; 2018) has been employed in applications such as domain adaptation (Fatras et al., 2021), natural language processing (Chen et al., 2020b), and computer vision (De Plaen et al., 2023). Existing works (Piccoli & Rossi, 2014; 2016; Hanin, 1992; Georgiou et al., 2009) have also studied total variation (TV)-regularization-based UOT formulations. While

MMD-based methods have been popularly employed in several machine learning (ML) applications (Gretton et al., 2012; Li et al., 2017; 2021; Nguyen et al., 2021), the applicability of MMD-based regularization for UOT is not well-understood. To the best of our knowledge, interesting questions like the following, have not been answered in prior works:

- Will MMD regularization for UOT also lead to novel metrics over measures, analogous to the ones obtained with the KL divergence (Liero et al., 2018) or the TV distance (Piccoli & Rossi, 2014)?

- What will be the statistical estimation properties of these?

- How can such MMD regularized UOT metrics be estimated in practice such that they are suitable for large-scale applications?

In order to bridge this gap, we study MMD-based regularization for matching the marginals of the transport plan in the UOT formulation (henceforth termed MMD-UOT).

We first derive a specific dual of the MMD-UOT formulation (Theorem 4.1), which helps further analyze its properties. One interesting consequence of this duality result is that the optimal objective of MMD-UOT is a valid distance between the source and target measures (Corollary 4.2), whenever the transport cost is valid (ground) metric over the data points. Popularly, this is known as the phenomenon of lifting metrics to measures. This result is significant as it shows that MMD-regularization in UOT can parallel the metricity-preservation that happens with KL-regularization (Liero et al., 2018) and TV-regularization (Piccoli & Rossi, 2014). Furthermore, our duality result shows that this induced metric is a novel metric belonging to the family of integral probability metrics (IPMs) with a generating set that is the intersection of the generating sets of MMD and the Kantorovich-Wasserstein metric. Because of this important relation, the proposed distance is always smaller than the MMD distance, and hence, estimating MMD-UOT from samples is at least as efficient as that with MMD (Corollary 4.6). This is interesting as minimax estimation rates for MMD can be completely dimension-free. As far as we know, there are no such results that show that estimation with KL/TV-regularized UOT can be as efficient sample-wise. Thus, the proposed metrics not only lift the ground metrics to measures, like the Wasserstein, but also are sample-wise efficient to estimate, like MMD.

However, like any formulation of optimal transport problems, the computation of MMD-UOT involves optimization over all possible joint measures. This may be challenging, especially when the measures are continuous. Hence, we present a convex program-based estimator, which only involves a search over joints supported at the samples. We prove that the proposed estimator is statistically consistent and converges to MMD-UOT between the true measures at a rate $\mathcal{O}\left(m^{-\frac{1}{2}}\right)$, where $m$ is the number of samples. Such efficient estimators are particularly useful in machine learning applications, where typically only samples from the underlying measures are available. Such applications include hypothesis testing, domain adaptation, and model interpolation, to name a few. In contrast, the minimax estimation rate for the Wasserstein distance is itself $\mathcal{O}\left(m^{-\frac{1}{d}}\right)$, where $d$ is the dimensionality of the samples (Niles-Weed & Rigollet, 2019). That is, even if a search over all possible joints is performed, estimating Wasserstein may be challenging. Since MMD-UOT can approximate Wasserstein arbitrarily closely (as the regularization hyperparameter goes $\infty$), our result can also be understood as a way of alleviating the curse of dimensionality problem in Wasserstein. We summarize the comparison between MMD-UOT and relevant OT variants in Table 1.

Finally, our result of MMD-UOT being a metric facilitates its application whenever the metric properties of OT are desired, for example, while computing the barycenter-based interpolation for single-cell RNA sequencing (Tong et al., 2020). Accordingly, we also present a finite-dimensional convex-program-based estimator for the barycenter with MMD-UOT. We prove that this estimator is also consistent with an efficient sample complexity. We discuss how the formulations for estimating MMD-UOT (and barycenter) can be solved efficiently using accelerated (projected) gradient descent. This solver helps us scale well to large datasets. We empirically show the utility of MMD-UOT in several applications including two-sample hypothesis testing, single-cell RNA sequencing, domain adaptation, and prompt learning for few-shot classification. In particular, we observe that MMD-UOT outperforms popular baselines such as KL-regularized UOT and MMD in our experiments.

We summarize our main contributions below:

Table 1: Summarizing interesting properties of MMD and several OT/UOT approaches. $\epsilon$OT (Cuturi, 2013) and $\epsilon$KL-UOT (Chizat, 2017) denote the entropy-regularized scalable variants OT and KL-UOT (Liero et al., 2018), respectively. MMD and the proposed MMD-UOT are shown with characteristic kernels. By 'finite-parameterization bounds' we mean results similar to Theorem 4.10.

| Property | MMD | OT | $\epsilon$OT | TV-UOT | KL-UOT | $\epsilon$KL-UOT | MMD-UOT |
|---|---|---|---|---|---|---|---|
| Metricity | ✓ | ✓ | ✗ | ✓ | ✓ | ✗ | ✓ |
| Lifting of ground metric | ✗ | ✓ | ✓ | ✓ | ✓ | ✓ | ✓ |
| No curse of dimensionality | ✓ | ✗ | ✗ | ✗ | ✗ | ✗ | ✓ |
| Finite-parametrization bounds | N/A | ✗ | ✗ | ✗ | ✗ | ✗ | ✓ |

- Dual of MMD-UOT and its analysis. We prove that MMD-UOT induces novel metrics that not only lift ground metrics like the Wasserstein but also are sample-wise efficient to estimate like the MMD.

- Finite-dimensional convex-program-based estimators for MMD-UOT and the corresponding barycenter. We prove that the estimators are both statistically and computationally efficient.

- We illustrate the efficacy of MMD-UOT in several real-world applications. Empirically, we observe that MMD-UOT consistently outperforms popular baseline approaches.

We present proofs for all our theory results in Appendix B. As a side-remark, we note that most of our results not only hold for MMD-UOT but also for a UOT formulation where a general IPM replaces MMD. Proofs in the appendix are hence written for general IPM-based regularization and then specialized to the case when the IPM is MMD. This generalization to IPMs may itself be of independent interest.

## 2 Preliminaries

**Notations.** Let $\mathcal{X}$ be a set (domain) that forms a compact Hausdorff space. Let $\mathcal{R}^+(\mathcal{X}), \mathcal{R}(\mathcal{X})$ denote the set of all non-negative, signed (finite) Radon measures defined over $\mathcal{X}$; while the set of all probability measures is denoted by $\mathcal{R}_1^+(\mathcal{X})$. For a measure on the product space, $\pi \in \mathcal{R}^+(\mathcal{X} \times \mathcal{X})$, let $\pi_1, \pi_2$ denote the first and second marginals, respectively (i.e., they are the push-forwards under the canonical projection maps onto $\mathcal{X}$). Let $\mathcal{L}(\mathcal{X}), \mathcal{C}(\mathcal{X})$ denote the set of all real-valued measurable functions and all real-valued continuous functions, respectively, over $\mathcal{X}$.

**Integral Probability Metric (IPM):** Given a set $\mathcal{G} \subset \mathcal{L}(\mathcal{X})$, the integral probability metric (IPM) (Muller, 1997; Sriperumbudur et al., 2009; Agrawal & Horel, 2020) associated with $\mathcal{G}$, is defined by:

$$\gamma_{\mathcal{G}}(s_0, t_0) \equiv \max_{f \in \mathcal{G}} \left| \int_{\mathcal{X}} f \, \mathrm{d}s_0 - \int_{\mathcal{X}} f \, \mathrm{d}t_0 \right| \ \forall \ s_0, t_0 \in \mathcal{R}^+(\mathcal{X}). \tag{1}$$

$\mathcal{G}$ is called the generating set of the IPM, $\gamma_{\mathcal{G}}$.

**Maximum Mean Discrepancy (MMD)** Let $k$ be a characteristic kernel (Sriperumbudur et al., 2011) over the domain $\mathcal{X}$, let $\|f\|_k$ denote the norm of $f$ in the canonical reproducing kernel Hilbert space (RKHS), $\mathcal{H}_k$, corresponding to $k$. $\mathrm{MMD}_k$ is the IPM associated with the generating set: $\mathcal{G}_k \equiv \{f \in \mathcal{H}_k | \ \|f\|_k \leq 1\}$. Using a characteristic kernel $k$, MMD metric between $s_0, t_0 \in \mathcal{R}^+(\mathcal{X})$ is defined as:

$$\begin{aligned} \mathrm{MMD}_k(s_0, t_0) & \equiv \max_{f \in \mathcal{G}_k} \left| \int_{\mathcal{X}} f \, \mathrm{d}s_0 - \int_{\mathcal{X}} f \, \mathrm{d}t_0 \right| \\ & = \| \mu_k(s_0) - \mu_k(t_0) \|_k, \end{aligned} \tag{2}$$

where $\mu_k(s) \equiv \int \phi_k(x) \, \mathrm{d}s(x)$, is the kernel mean embedding of $s$ (Muandet et al., 2017), $\phi_k$ is the canonical feature map of $k$. A kernel $k$ is called a characteristic kernel if the map $\mu_k$ is injective. MMD can be computed analytically using evaluations of the kernel $k$. $\mathrm{MMD}_k$ is a metric when the kernel $k$ is characteristic. A continuous positive-definite kernel $k$ on $\mathcal{X}$ is called c-universal if the RKHS $\mathcal{H}_k$ is dense in $\mathcal{C}(\mathcal{X})$ w.r.t. the

sup-norm, i.e., for every function $g \in \mathcal{C}(\mathcal{X})$ and all $\epsilon > 0$, there exists an $f \in \mathcal{H}_k$ such that $\|f - g\|_\infty \leq \epsilon$. Universal kernels are also characteristic. Gaussian kernel (RBF kernel) is an example of a universal kernel over the continuous domain. Dirac delta kernel is an example of a universal kernel over the discrete domain.

**Optimal Transport (OT)** Optimal transport provides a tool to compare distributions while incorporating the underlying geometry of their support points. Given a cost function, $c : \mathcal{X} \times \mathcal{X} \mapsto \mathbb{R}$, and two probability measures $s_0 \in \mathcal{R}_1^+(\mathcal{X}), t_0 \in \mathcal{R}_1^+(\mathcal{X})$, the $p$-Wasserstein Kantorovich OT formulation is given by:

$$\bar{W}_p^p(s_0, t_0) \equiv \min_{\pi \in \mathcal{R}_1^+(\mathcal{X} \times \mathcal{X})} \int c^p \, \mathrm{d}\pi, \text{ s.t. } \pi_1 = s_0, \ \pi_2 = t_0, \tag{3}$$

where $p \geq 1$. An optimal solution of (3) is called an optimal transport plan. Whenever the cost is a metric, $d$, over $\mathcal{X} \times \mathcal{X}$ (ground metric), $\bar{W}_p$ defines a metric over measures, known as the $p$-Wasserstein metric, over $\mathcal{R}_1^+(\mathcal{X}) \times \mathcal{R}_1^+(\mathcal{X})$.

**Kantorovich metric ($\mathcal{K}_c$)** Kantorovich metric also belongs to the family of integral probability metrics associated with the generating set $\mathcal{W}_c \equiv \left\{ f : \mathcal{X} \mapsto \mathbb{R} \mid \max_{x \in \mathcal{X} \neq y \in \mathcal{X}} \frac{|f(x) - f(y)|}{c(x,y)} \leq 1 \right\}$, where $c$ is a metric over $\mathcal{X} \times \mathcal{X}$. The Kantorovich-Rubinstein duality result shows that the 1-Wasserstein metric is the same as the Kantorovich metric when restricted to probability measures (refer for e.g. (5.11) in Villani (2009)):

$$\bar{W}_1(s_0, t_0) \equiv \min_{\substack{\pi \in \mathcal{R}_1^+(\mathcal{X} \times \mathcal{X}) \\ \text{s.t. } \pi_1 = s_0, \ \pi_2 = t_0}} \int c^p \, \mathrm{d}\pi, = \max_{f \in \mathcal{G}} \left| \int_{\mathcal{X}} f \, \mathrm{d}s_0 - \int_{\mathcal{X}} f \, \mathrm{d}t_0 \right| \equiv \mathcal{K}_c(s_0, t_0),$$

where $s_0, t_0 \in \mathcal{R}_1^+(\mathcal{X})$.

## 3 Related Work

Given the source and target measures, $s_0 \in \mathcal{R}^+(\mathcal{X})$ and $t_0 \in \mathcal{R}^+(\mathcal{X})$, respectively, the unbalanced optimal transport (UOT) approach (Liero et al., 2018; Chizat et al., 2018) aims to learn the transport plan by replacing the mass conservation marginal constraints (enforced strictly in 'balanced' OT setting) by a soft regularization/penalization on the marginals. KL-divergence and, in general, $\phi$-divergence (Csiszar, 1967), (Sriperumbudur et al., 2009) based regularizations have been most popularly studied in UOT setting. The $\phi$-divergence regularized UOT formulation may be written as (Frogner et al., 2015), (Chizat, 2017):

$$\min_{\pi \in \mathcal{R}^+(\mathcal{X} \times \mathcal{X})} \int c \, \mathrm{d}\pi + \lambda D_\phi(\pi_1, s_0) + \lambda D_\phi(\pi_2, t_0), \tag{4}$$

where $c$ is the ground cost metric and $D_\phi(\cdot, \cdot)$ denotes the $\phi$-divergence (Csiszar, 1967; Sriperumbudur et al., 2009) between two measures. Since in UOT settings, the measures $s_0$, $t_0$ may be un-normalized, following (Chizat, 2017; Liero et al., 2018) the transport plan is also allowed to be un-normalized. UOT with KL-divergence-based regularization induces the so-called Gaussian Hellinger-Kantorovich metric (Liero et al., 2018) between the measures whenever $0 < \lambda \leq 1$ and the ground cost $c$ is the squared-Euclidean distance. Similar to the balanced OT setup (Cuturi, 2013), an additional entropy regularization in KL-UOT formulation facilitates Sinkhorn iteration (Knight, 2008) based efficient solver for KL-UOT (Chizat et al., 2017) and has been popularly employed in several machine learning applications (Fatras et al., 2021; Chen et al., 2020b; Arase et al., 2023; De Plaen et al., 2023).

Total Variation (TV) distance is another popular metric between measures and is the only common member of the $\phi$-divergence family and the IPM family. UOT formulation with TV regularization (denoted by $|\cdot|_{\mathrm{TV}}$) has been studied in (Piccoli & Rossi, 2014):

$$\min_{\pi \in \mathcal{R}^+(\mathcal{X} \times \mathcal{X})} \int c \, \mathrm{d}\pi + \lambda |\pi_1 - s_0|_{\mathrm{TV}} + \lambda |\pi_2 - t_0|_{\mathrm{TV}}. \tag{5}$$

UOT with TV-divergence-based regularization induces the so-called Generalized Wasserstein metric (Piccoli & Rossi, 2014) between the measures whenever $\lambda > 0$ and the ground cost $c$ is a valid metric. As far as

we know, none of the existing works study the sample complexity of estimating these metrics from samples. More importantly, algorithms for solving (5) with empirical measures that computationally scale well to ML applications seem to be absent in the literature.

Besides the family of $\phi$-divergences, the family of integral probability metrics is popularly used for comparing measures. An important member of the IPM family is the MMD metric, which also incorporates the geometry over supports through the underlying kernel. Due to its attractive statistical properties (Gretton et al., 2006), MMD has been successfully applied in a diverse set of applications including hypothesis testing (Gretton et al., 2012), generative modelling (Li et al., 2017), self-supervised learning (Li et al., 2021), etc.

Recently, (Nath & Jawanpuria, 2020) explored learning the transport plan's kernel mean embeddings in the balanced OT setup. They proposed learning the kernel mean embedding of a joint distribution with the least expected cost and whose marginal embeddings are close to the given-sample-based estimates of the marginal embeddings. As kernel mean embedding induces MMD distance, MMD-based regularization features in the balanced OT formulation of (Nath & Jawanpuria, 2020) as a means to control overfitting. To ensure that valid conditional embeddings are obtained from the learned joint embeddings, (Nath & Jawanpuria, 2020) required additional feasibility constraints that restrict their solvers in scaling well to machine learning applications. We also note that (Nath & Jawanpuria, 2020) neither analyze the dual of their formulation nor study its metric-related properties and their sample complexity result of $\mathcal{O}(m^{-\frac{1}{2}})$ does not apply to our MMD-UOT estimator as their formulation is different from the proposed MMD-UOT formulation (6).

In contrast, we bypass the issues related to the validity of conditional embeddings as our formulation involves directly learning the transport plan and avoids kernel mean embedding of the transport plan. We perform a detailed study of MMD regularization for UOT, which includes analyzing its dual and proving metric properties that are crucial for optimal transport formulations. To the best of our knowledge, the metricity of MMD-regularized UOT formulations has not been studied previously. The proposed algorithm scales well to large-scale machine learning applications. While we also obtain $\mathcal{O}(m^{-\frac{1}{2}})$ estimation error rate, we require a different proof strategy than (Nath & Jawanpuria, 2020). Finally, as discussed in Appendix B, most of our theoretical results apply to a general IPM-regularized UOT formulation and are not limited to the MMD-regularized UOT formulation. This generalization does not hold for (Nath & Jawanpuria, 2020).

Wasserstein auto-encoders (WAE) also employ MMD for regularization. However, there are some important differences. The regularization in WAEs is only performed for one of the marginals, and the other marginal is matched exactly. This not only breaks the symmetry (and hence the metric properties) but also brings back the curse of dimensionality in estimation (for the same reasons as with unregularized OT). Further, their work does not attempt to study any theoretical properties with MMD regularization and merely employs it as a practical tool for matching marginals. Our goal is to theoretically study the metric and estimation properties with MMD regularization. We present more details in Appendix B.18.

We end this section by noting key differences between MMD and OT-based approaches (including MMD-UOT). A distinguishing feature of OT-based approaches is the phenomenon of lifting the ground-metric geometry to that over distributions. One such result is visualized in Figure 2(b), where the MMD-based-interpolate of the two unimodal distributions comes out to be bimodal. This is because MMD's interpolation is the (literal) average of the source and the target densities, irrespective of the kernel. This has been well-established in the literature (Bottou et al., 2017). On the other hand, OT-based approaches obtain a unimodal barycenter. This is a 'geometric' interpolation that captures the characteristic aspects of the source and the target distributions. Another feature of OT-based methods is that we obtain a transport plan between the source and the target points which can be used for various alignment-based applications, e.g., cross-lingual word mapping (Alvarez-Melis & Jaakkola, 2018; Jawanpuria et al., 2020), domain adaptation (Courty et al., 2017; Courty et al., 2017; Gurumoorthy et al., 2021), etc. On the other hand, it is unclear how MMD can be used to align the source and target data points.

## 4 MMD Regularization for UOT

We propose to study the following UOT formulation, where the marginal constraints are enforced using MMD regularization.

$$
\begin{aligned}
\mathcal{U}_{k,c,\lambda_1,\lambda_2}(s_0,t_0) &\equiv \min_{\pi \in \mathcal{R}^+(\mathcal{X} \times \mathcal{X})} \int c \, \mathrm{d}\pi + \lambda_1 \mathrm{MMD}_k(\pi_1, s_0) + \lambda_2 \mathrm{MMD}_k(\pi_2, t_0) \\
&= \min_{\pi \in \mathcal{R}^+(\mathcal{X} \times \mathcal{X})} \int c \, \mathrm{d}\pi + \lambda_1 \|\mu_k(\pi_1) - \mu_k(s_0)\|_k + \lambda_2 \|\mu_k(\pi_2) - \mu_k(t_0)\|_k,
\end{aligned}
\tag{6}
$$

where $\mu_k(s)$ is the kernel mean embedding of $s$ (defined in Section 2) induced by the characteristic kernel $k$ used in the generating set $\mathcal{G}_k \equiv \{f \in \mathcal{H}_k \mid \|f\|_k \leq 1\}$, and $\lambda_1, \lambda_2 > 0$ are the regularization hyper-parameters.

We begin by presenting a key duality result.

**Theorem 4.1. (Duality)** *Whenever $c, k \in \mathcal{C}(\mathcal{X} \times \mathcal{X})$ and $\mathcal{X}$ is compact, we have that:*

$$
\begin{aligned}
\mathcal{U}_{k,c,\lambda_1,\lambda_2}(s_0,t_0) = \max_{f \in \mathcal{G}_k(\lambda_1), g \in \mathcal{G}_k(\lambda_2)} &\int_{\mathcal{X}} f \, \mathrm{d}s_0 + \int_{\mathcal{X}} g \, \mathrm{d}t_0, \\
\text{s.t.} \quad &f(x) + g(y) \leq c(x,y) \; \forall \; x, y \in \mathcal{X}.
\end{aligned}
\tag{7}
$$

*Here, $\mathcal{G}_k(\lambda) \equiv \{g \in \mathcal{H}_k \mid \|g\|_k \leq \lambda\}$.*

The duality result helps us to study several properties of the MMD-UOT (6), discussed in the corollaries below. The proof of Theorem 4.1 is based on an application of Sion's minimax exchange theorem (Sion, 1958) and is detailed in Appendix B.1.

Applications in machine learning often involve comparing distributions for which the Wasserstein metric is a popular choice. While prior works have shown metric-preservation happens under KL-regularization (Liero et al., 2018) and TV-regularization (Piccoli & Rossi, 2016), it is an open question if MMD-regularization in UOT can also lead to valid metrics. The following result answers this affirmatively.

**Corollary 4.2. (Metricity)** *In addition to assumptions in Theorem (4.1), whenever $c$ is a metric, $\mathcal{U}_{k,c,\lambda,\lambda}$ belongs to the family of integral probability metrics (IPMs). Also, the generating set of this IPM is the intersection of the generating set of the Kantorovich metric and the generating set of MMD. Finally, $\mathcal{U}_{k,c,\lambda,\lambda}$ is a valid norm-induced metric over measures whenever $k$ is characteristic. Thus, $\mathcal{U}$ lifts the ground metric $c$ to that over measures.*

The proof of Corollary 4.2 is detailed in Appendix B.2. This result also reveals interesting relationships between $\mathcal{U}_{k,c,\lambda,\lambda}$, the Kantorovich metric, $\mathcal{K}_c$, and the MMD metric used for regularization. This is summarized in the following two results.

**Corollary 4.3. (Interpolant)** *In addition to assumptions in Corollary 4.2, if the kernel is $c$-universal (continuous and universal), then $\forall \; s_0, t_0 \in \mathcal{R}^+(\mathcal{X})$, $\lim_{\lambda \to \infty} \mathcal{U}_{k,c,\lambda,\lambda}(s_0,t_0) = \mathcal{K}_c(s_0,t_0)$. Further, if the cost metric, $c$, dominates the characteristic kernel, $k$, induced metric, i.e., $c(x,y) \geq \sqrt{k(x,x) + k(y,y) - 2k(x,y)} \; \forall \; x, y \in \mathcal{X}$, then $\mathcal{U}_{k,c,\lambda,\lambda}(s_0,t_0) = \lambda \mathrm{MMD}_k(s_0,t_0)$ whenever $0 < \lambda \leq 1$. Finally, when $\lambda \in (0,1)$, MMD-UOT interpolates between the scaled MMD and the Kantorovich metric. The nature of this interpolation is already described in terms of generating sets in Corollary 4.2.*

We illustrate this interpolation result in Figure 1. Our proof of Corollary 4.3, presented in Appendix B.3, also shows that the Euclidean distance satisfies such a dominating cost assumption when the kernel employed is the Gaussian kernel and the inputs lie on a unit-norm ball. The next result presents another relationship between the metrics in the discussion.

**Corollary 4.4.** $\mathcal{U}_{k,c,\lambda,\lambda}(s,t) \leq \min\left(\lambda \mathrm{MMD}_k(s,t), \mathcal{K}_c(s,t)\right).$

The proof of Corollary 4.4 is straightforward and is presented in Appendix B.5. This result enables us to show properties like weak metrization and sample efficiency with MMD-UOT. For a sequence $s_n \in \mathcal{R}_1^+(\mathcal{X})$, $n \geq 1$, we say that $s_n$ weakly converges to $s \in \mathcal{R}_1^+(\mathcal{X})$ (denoted as $s_n \rightharpoonup s$), if and only if $\mathbb{E}_{X \sim s_n}[f(X)] \to \mathbb{E}_{X \sim s}[f(X)]$ for all bounded continuous functions over $\mathcal{X}$. It is natural to ask when is the convergence in metric over measures equivalent to weak convergence on measures. The metric is then said to metrize the

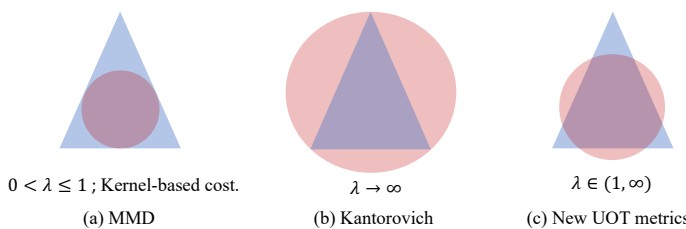

$0 < \lambda \leq 1$ ; Kernel-based cost.          $\lambda \to \infty$          $\lambda \in (1, \infty)$

(a) MMD          (b) Kantorovich          (c) New UOT metrics

Figure 1: For illustration, the generating set of Kantorovich-Wasserstein is depicted as a triangle, and the scaled generating set of MMD is depicted as a disc. The intersection represents the generating set of the IPM metric induced by MMD-UOT. (a) shows the special case when our MMD-UOT metric recovers back the sample-efficient MMD metric, (b) shows the special case when our MMD-UOT metric reduces to the Kantorovich-Wasserstein metric that lifts the ground metric to measures, and (c) shows the resulting family of new UOT metrics which are both sample-efficient and can lift ground metrics to measures.

weak convergence of measures or is equivalently said to weakly metrize measures. The weak metrization properties of the Wasserstein metric and MMD are well-understood (e.g., refer to Theorem 6.9 in (Villani, 2009) and Theorem 7 in (Simon-Gabriel et al., 2020)). The weak metrization property of $\mathcal{U}_{k,c,\lambda,\lambda}$ follows from the above Corollary 4.4.

**Corollary 4.5.** *(Weak Metrization)* $\mathcal{U}_{k,c,\lambda,\lambda}$ *metrizes the weak convergence of normalized measures.*

The proof is presented in Appendix B.6. We now show that the metric induced by MMD-UOT inherits the attractive statistical efficiency of the MMD metric. In typical machine learning applications, only finite samples are given from the measures. Hence, it is important to study statistically efficient metrics that alleviate the curse of dimensionality problem prevalent in OT (Niles-Weed & Rigollet, 2019). Sample complexity result with the metric induced by MMD-UOT is presented as follows.

**Corollary 4.6.** *(Sample Complexity) Let us denote $\mathcal{U}_{k,c,\lambda,\lambda}$, defined in (6), by $\bar{\mathcal{U}}$. Let $\hat{s}_m, \hat{t}_m$ denote the empirical estimates of $s_0, t_0 \in \mathcal{R}^+(\mathcal{X})$ respectively with $m$ samples. Then, $\bar{\mathcal{U}}(\hat{s}_m, \hat{t}_m) \to \bar{\mathcal{U}}(s_0, t_0)$ at a rate (apart from constants) same as that of $\mathrm{MMD}_k(\hat{s}_m, s_0) \to 0$.*

Since the sample complexity of MMD with a normalized characteristic kernel is $\mathcal{O}(m^{-\frac{1}{2}})$ (Smola et al., 2007), the same will be the complexity bound for the corresponding MMD-UOT. The proof of Corollary 4.6 is presented in Appendix B.7. This is interesting because, though MMD-UOT can arbitrarily well approximate Wasserstein (as $\lambda \to \infty$), its estimation can be far more efficient than $\mathcal{O}\left(m^{-\frac{1}{d}}\right)$, which is the minimax estimation rate for the Wasserstein (Niles-Weed & Rigollet, 2019). Here, $d$ is the dimensionality of the samples. Further, in Lemma B4, we show that even when $\mathrm{MMD}_k^q$ ($q \geq 2 \in \mathbb{N}$) is used for regularization, the sample complexity again comes out to be $\mathcal{O}\left(m^{-\frac{1}{2}}\right)$. We conclude this section with a couple of remarks.

**Remark 4.7.** *As a side result, we prove the following theorem (Appendix B.8) that relates our MMD-UOT to the MMD-regularized Kantorovich metric. We believe this connection is interesting as it generalizes the popular Kantorovich-Rubinstein duality result on relating (unregularized) OT to the (unregularized) Kantorovich metric.*

**Theorem 4.8.** *In addition to the assumptions in Theorem 4.1, if $c$ is a valid metric, then*

$$\mathcal{U}_{k,c,\lambda_1,\lambda_2}(s_0, t_0) = \min_{s,t \in \mathcal{R}(\mathcal{X})} \mathcal{K}_c(s,t) + \lambda_1 \mathrm{MMD}_k(s, s_0) + \lambda_2 \mathrm{MMD}_k(t, t_0). \tag{8}$$

**Remark 4.9.** *It is noteworthy that most of our theoretical results presented in this section not only hold with the MMD-UOT formulation (9) but also with a general IPM-regularized UOT formulation, which we discuss in Appendix B. This generalization may be of independent interest for future work.*

Finally, minor results on robustness and connections with spectral normalized GAN (Miyato et al., 2018) are discussed in Appendix B.16 and Appendix B.17, respectively.

### 4.1 Finite-Sample-based Estimation

As noted in Corollary 4.6, MMD-UOT can be efficiently estimated from samples of source and target. However, one needs to solve an optimization problem over all possible joint (un-normalized) measures. This can be computationally expensive[1] (for example, optimization over the set of all joint density functions). Hence, in this section, we propose a simple estimator where the optimization is only over the joint measures supported at sample-based points. We show that our estimator is statistically consistent and that the estimation is free from the curse of dimensionality.

Let $m$ samples be given from the source, target, $s_0$, $t_0 \in \mathcal{R}^+(\mathcal{X})$ respectively[2]. We denote $\mathcal{D}_i = \{x_{i1}, \cdots x_{im}\}, i = 1, 2$ as the set of samples given from $s_0, t_0$ respectively. Let $\hat{s}_m, \hat{t}_m$ denote the empirical measures using samples $\mathcal{D}_1, \mathcal{D}_2$. Let us denote the Gram-matrix of $\mathcal{D}_i$ by $G_{ii}$. Let $\mathcal{C}_{12}$ be the $m \times m$ cost matrix with entries as evaluations of the cost function over $\mathcal{D}_1 \times \mathcal{D}_2$. Following the common practice in OT literature (Chizat et al., 2017; Cuturi, 2013; Damodaran et al., 2018; Fatras et al., 2021; Le et al., 2021; Balaji et al., 2020; Nath & Jawanpuria, 2020; Peyré & Cuturi, 2019), we restrict the transport plan to be supported on the finite samples from each of the measures in order to avoid the computational issues in optimizing over all possible joint densities. More specifically, let $\alpha$ be the $m \times m$ (parameter/variable) matrix with entries as $\alpha_{ij} \equiv \pi(x_{1i}, x_{2j})$ where $i, j \in \{1, \cdots, m\}$. With these notations and the mentioned restricted feasibility set, Problem (6) simplifies to the following, denoted by $\hat{\mathcal{U}}_m(\hat{s}_m, \hat{t}_m)$:

$$\min_{\alpha \geq 0 \in \mathbb{R}^{m \times m}} \mathrm{Tr}\left(\alpha \mathcal{C}_{12}^\top\right) + \lambda_1 \left\| \alpha \mathbf{1} - \frac{\sigma_1}{m} \mathbf{1} \right\|_{G_{11}} + \lambda_2 \left\| \alpha^\top \mathbf{1} - \frac{\sigma_2}{m} \mathbf{1} \right\|_{G_{22}}, \tag{9}$$

where $\mathrm{Tr}(M)$ denotes the trace of matrix $M$, $\|x\|_M \equiv \sqrt{x^\top M x}$, and $\sigma_1, \sigma_2$ are the masses of the source, target measures, $s_0, t_0$, respectively. Since this is a Convex Program over a finite-dimensional variable, it can be solved in a computationally efficient manner (refer Section 4.2).

However, as the transport plan is now supported on the given samples alone, Corollary 4.6 does not apply. The following result shows that our estimator (9) is consistent, and the estimation error decays at a favourable rate.

**Theorem 4.10.** *(Consistency of the proposed estimator) Let us denote $\mathcal{U}_{k,c,\lambda_1,\lambda_2}$, defined in (6), by $\bar{\mathcal{U}}$. Assume the domain $\mathcal{X}$ is compact, ground cost is continuous, $c \in \mathcal{C}(\mathcal{X} \times \mathcal{X})$, and the kernel $k$ is c-universal, normalized. Let the source measure ($s_0$), the target measure ($t_0$), as well as the corresponding MMD-UOT transport plan be absolutely continuous. Also assume $s_0(x), t_0(x) > 0 \ \forall \ x \in \mathcal{X}$. Then, we have w.h.p. and any (arbitrarily small) $\epsilon > 0$ that $\left| \hat{\mathcal{U}}_m(\hat{s}_m, \hat{t}_m) - \bar{\mathcal{U}}(s_0, t_0) \right| \leq \mathcal{O}\left( \frac{\lambda_1 + \lambda_2}{\sqrt{m}} + \frac{g(\epsilon)}{m} + \epsilon \sigma \right)$. Here, $g(\epsilon) \equiv \min_{v \in \mathcal{H}_k \otimes \mathcal{H}_k} \|v\|_k$ s.t. $\|v - c\|_\infty \leq \epsilon$, and $\sigma$ is the mass of the optimal MMD-UOT transport plan. Further, if $c$ belongs to $\mathcal{H}_k \otimes \mathcal{H}_k$, then w.h.p. $\left| \hat{\mathcal{U}}_m(\hat{s}_m, \hat{t}_m) - \bar{\mathcal{U}}(s_0, t_0) \right| \leq \mathcal{O}\left( \frac{\lambda_1 + \lambda_2}{\sqrt{m}} \right)$.*

We discuss the proof of the above theorem in Appendix B.9. Because $k$ is universal, $g(\epsilon) < \infty \ \forall \ \epsilon > 0$. The consistency of our estimator as $m \to \infty$ can be realized, if, for example, one employs the scheme $\lambda_1 = \lambda_2 = \mathcal{O}(m^{1/4})$ and $\epsilon \to 0$ at a slow enough rate such that $\frac{g(\epsilon)}{m} \to 0$. In Appendix B.9.1, we show that even if $\epsilon$ decays as fast as $\mathcal{O}\left(\frac{1}{m^{2/3}}\right)$, then $g(\epsilon)$ blows-up almost as $\mathcal{O}\left(m^{1/3}\right)$. Hence, overall, the estimation error still decays as $\mathcal{O}\left(\frac{1}{m^{1/4}}\right)$. To the best of our knowledge, such consistency results have not been studied in the context of KL-regularized UOT.

### 4.2 Computational Aspects

Problem (9) is an instance of a convex program and can be solved using the mirror descent algorithm detailed in Appendix B.10. In the following, we propose to solve an equivalent optimization problem which helps us leverage faster solvers for MMD-UOT:

$$\min_{\alpha \geq 0 \in \mathbb{R}^{m \times m}} \mathrm{Tr}\left(\alpha \mathcal{C}_{12}^\top\right) + \lambda_1 \left\| \alpha \mathbf{1} - \frac{\sigma_1}{m} \mathbf{1} \right\|_{G_{11}}^2 + \lambda_2 \left\| \alpha^\top \mathbf{1} - \frac{\sigma_2}{m} \mathbf{1} \right\|_{G_{22}}^2. \tag{10}$$

---

[1] Note that this challenge is inherent to OT (and all its variants). It is not a consequence of our choice of MMD regularization.

[2] The no. of samples from source and target need not be the same, in general.

---

**Algorithm 1** Accelerated Projected Gradient Descent for solving Problem (10).

---

**Require:** Lipschitz constant $L$, initial $\alpha_0 \geq 0 \in \mathbb{R}^{m \times m}$.

$\quad f(\alpha) = \text{Tr}\left(\alpha \mathcal{C}_{12}^\top\right) + \lambda_1 \left\|\alpha \mathbf{1} - \frac{\sigma_1}{m}\mathbf{1}\right\|_{G_{11}}^2 + \lambda_2 \left\|\alpha^\top \mathbf{1} - \frac{\sigma_2}{m}\mathbf{1}\right\|_{G_{22}}^2$.

$\quad \gamma_1 = 1$.

$\quad y_1 = \alpha_0$.

$\quad i = 0$.

$\quad$**while** not converged **do**

$\qquad \alpha_i = \text{Project}_{\geq 0}\left(y_i - \frac{1}{L}\nabla f(y_i)\right)$.

$\qquad \gamma_{i+1} = \frac{1+\sqrt{1+4\gamma_i^2}}{2}$.

$\qquad y_{i+1} = \alpha_i + \frac{\gamma_i - 1}{\gamma_{i+1}}(\alpha_i - \alpha_{i-1})$.

$\qquad i = i + 1$.

$\quad$**end while**

**return** $\alpha_i$.

---

The equivalence between (9) and (10) follows from standard arguments and is detailed in Appendix B.11. Our next result shows that the objective in (10) is $L$-smooth (proof provided in Appendix B.12).

**Lemma 4.11.** *The objective in Problem (10) is $L$-smooth with $L = 2\sqrt{(\lambda_1 m)^2\|G_{11}\|_F^2 + (\lambda_2 m)^2\|G_{22}\|_F^2 + 2\lambda_1\lambda_2(\mathbf{1}_m^\top G_{11}\mathbf{1}_m + \mathbf{1}_m^\top G_{22}\mathbf{1}_m)}$.*

The above result enables us to use the accelerated projected gradient descent (APGD) algorithm (Nesterov, 2003; Beck & Teboulle, 2009) with fixed step-size $\tau = 1/L$ for solving (10). The detailed steps are presented in Algorithm 1. The overall computation cost for solving MMD-UOT (10) is $\mathcal{O}(\frac{m^2}{\sqrt{\epsilon}})$, where $\epsilon$ is the optimality gap. In Section 5, we empirically observe that the APGD-based solver for MMD-UOT is indeed computationally efficient.

### 4.3 Barycenter

A related problem is that of barycenter interpolation of measures (Agueh & Carlier, 2011), which has interesting applications (Solomon et al., 2014; 2015; Gramfort et al., 2015). Given measures $s_1, \ldots, s_n$ with total masses $\sigma_1, \ldots, \sigma_n$ respectively, and interpolation weights $\rho_1, \ldots, \rho_n$, the barycenter $s \in \mathcal{R}^+(\mathcal{X})$ is defined as the solution of $\bar{\mathcal{B}}(s_1, \cdots, s_n) \equiv \min_{s \in \mathcal{R}^+(\mathcal{X})} \sum_{i=1}^n \rho_i \mathcal{U}_{k,c,\lambda_1,\lambda_2}(s_i, s)$.

In typical applications, only sample sets, $\mathcal{D}_i$, from $s_i$ are available instead of $s_i$ themselves. Let us denote the corresponding empirical measures by $\hat{s}_1, \ldots, \hat{s}_n$. One way to estimate the barycenter is to consider $\bar{\mathcal{B}}(\hat{s}_1, \cdots, \hat{s}_n)$. However, this may be computationally challenging to optimize, especially when the measures involved are continuous. So we propose estimating the barycenter with the restriction that the transport plan $\pi^i$ corresponding to $\mathcal{U}_{k,c,\lambda_1,\lambda_2}(\hat{s}_i, s)$ is supported on $\mathcal{D}_i \times \cup_{i=1}^n \mathcal{D}_i$. And, let $\alpha_i \geq 0 \in \mathbb{R}^{m_i \times m}$ denote the corresponding probabilities. Following (Cuturi & Doucet, 2014), we also assume that the barycenter, $s$, is supported on $\cup_{i=1}^n \mathcal{D}_i$. Let us denote the barycenter problem with this support restriction on the transport plans and the Barycenter as $\hat{\mathcal{B}}_m(\hat{s}_1, \cdots, \hat{s}_n)$. Let $G$ be the Gram-matrix of $\cup_{i=1}^n \mathcal{D}_i$ and $\mathcal{C}_i$ be the $m_i \times m$ matrix with entries as evaluations of the cost function.

**Lemma 4.12.** *The barycenter problem $\hat{\mathcal{B}}_m(\hat{s}_1, \cdots, \hat{s}_n)$ can be equivalently written as:*

$$\min_{\alpha_1, \cdots, \alpha_n \geq 0} \ \sum_{i=1}^n \rho_i \left(\text{Tr}\left(\alpha_i \mathcal{C}_i^\top\right) + \lambda_1 \|\alpha_i \mathbf{1} - \frac{\sigma_i}{m_i}\mathbf{1}\|_{G_{ii}}^2 + \lambda_2 \|\alpha_i^\top \mathbf{1} - \sum_{j=1}^n \rho_j \alpha_j^\top \mathbf{1}\|_G^2\right). \tag{11}$$

We present the proof in Appendix B.14.1. Similar to Problem (10), the objective in Problem (11) is a smooth quadratic program in each $\alpha_i$ and is jointly convex in $\alpha_i$'s. In Appendix B.14.2, we also present the details for solving Problem (11) using APGD as well as its statistical consistency in Appendix B.14.3.

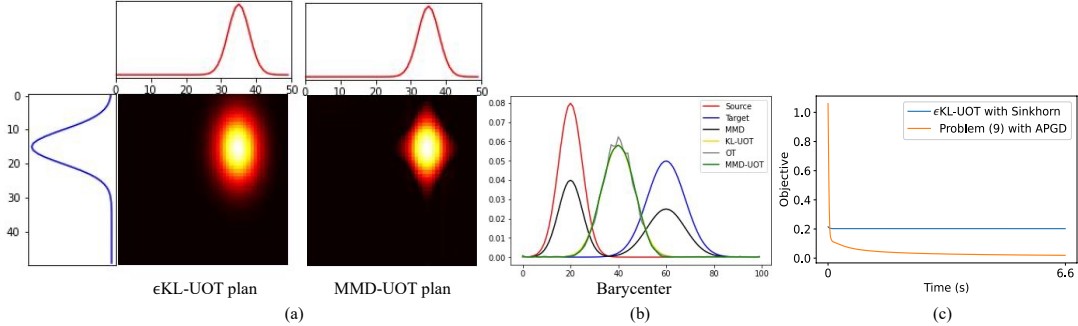

Figure 2: (a) Optimal Transport plans of $\epsilon$KL-UOT and MMD-UOT; (b) Barycenter interpolating between Gaussian measures. For the chosen hyperparameter, the barycenters of $\epsilon$KL-UOT and MMD-UOT overlap and can be looked as smooth approximations of the OT barycenter; (c) Objective vs Time plot comparing $\epsilon$KL-UOT solved using the popular Sinkhorn algorithm (Chizat et al., 2017; Pham et al., 2020) and MMD-UOT (10) solved using APGD. A plot showing $\epsilon$KL-UOT's progress at the initial phase is given in Figure 4.

## 5 Experiments

In Section 4, we examined the theoretical properties of the proposed MMD-UOT formulation. In this section, we show that MMD-UOT is a good practical alternative to the popular entropy-regularized $\epsilon$KL-UOT. We emphasize that our purpose is not to benchmark state-of-the-art performance. Our codes are publicly available at https://github.com/Piyushi-0/MMD-reg-OT.

### 5.1 Synthetic Experiments

We present some synthetic experiments to visualize the quality of our solution. Please refer to Appendix C.1 for more details.

**Transport Plan and Barycenter** We perform synthetic experiments with the source and target as Gaussian measures. We compare the OT plan of $\epsilon$KL-UOT and MMD-UOT in Figure 2(a). We observe that the MMD-UOT plan is sparser compared to the $\epsilon$KL-UOT plan. In Figure 2(b), we visualize the barycenter interpolating between the source and target, obtained with MMD, $\epsilon$KL-UOT and MMD-UOT. While MMD barycenter is an empirical average of the measures and hence has two modes, the geometry of measures is considered in both $\epsilon$KL-UOT and MMD-UOT formulations. Barycenters obtained by these methods have the same number of modes (one) as in the source and the target. Moreover, they appear to smoothly approximate the barycenter obtained with OT (solved using a linear program).

**Visualizing the Level Sets** Applications like generative modeling deal with optimization over the parameter ($\theta$) of the source distribution to match the target distribution. In such cases, it is desirable that the level sets of the distance function over the measures show a lesser number of stationary points that are not global optima (Bottou et al., 2017). Similar to (Bottou et al., 2017), we consider a model family for source distributions as $\mathcal{F} = \{P_\theta = \frac{1}{2}(\delta_\theta + \delta_{-\theta}) : \theta \in [-1,1] \text{ x } [-1,1]\}$ and a fixed target distribution $Q$ as $P_{(2,2)} \notin \mathcal{F}$. We compute the distances between $P_\theta$ and $Q$ according to various divergences. Figure 3 presents level sets showing the set of distances $\{d(P_\theta, Q) : \theta \in [-1,1] \text{ x } [-1,1]\}$ where the distance $d(.,.)$ is measured using MMD, Kantorovich metric, $\epsilon$KL-UOT, and MMD-UOT (9), respectively. While all methods correctly identify global minima (green arrow), level sets with MMD-UOT and $\epsilon$KL-UOT show no local minima (encircled in red for MMD) and have a lesser number of non-optimal stationary points (marked with black arrows) compared to the Kantorovich metric in Figure 3(b).

**Computation Time** In Figure 2(c), we present the objective versus time plot. The source and target measures are chosen to be the same, in which case the optimal objective is 0. MMD-UOT (10) solved using

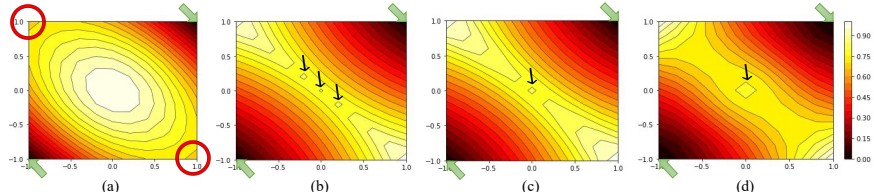

Figure 3: Level sets of distance function between a family of source distributions and a fixed target distribution with the task of finding the source distribution closest to the target distribution using (a) MMD, (b) $\bar{W}_2$, (c) $\epsilon$KL-UOT, and (d) MMD-UOT. While all methods correctly identify global minima (green arrows), level sets with MMD-UOT and $\epsilon$KL-UOT show no local minima (encircled in red for MMD) and have a lesser number of non-optimal stationary points (marked with black arrows) compared to (b).

Table 2: Average Test Power (between 0 and 1; higher is better) on MNIST. MMD-UOT obtains the highest average test power at all timesteps.

| N | MMD | $\epsilon$KL-UOT | MMD-UOT |
|---|---|---|---|
| 100 | 0.137 | 0.099 | **0.154** |
| 200 | 0.258 | 0.197 | **0.333** |
| 300 | 0.467 | 0.242 | **0.588** |
| 400 | 0.656 | 0.324 | **0.762** |
| 500 | 0.792 | 0.357 | **0.873** |
| 1000 | **0.909** | 0.506 | **0.909** |

APGD (described in Section 4.2) gives a much faster rate of decrease in objective compared to the Sinkhorn algorithm used for solving KL-UOT.

## 5.2 Two-Sample Hypothesis Test

Given two sets of samples $\{x_1, \ldots, x_m\} \sim s_0$ and $\{y_1, \ldots, y_m\} \sim t_0$, the two-sample test aims to determine whether the two sets of samples are drawn from the same distributions, viz., to predict if $s_0 = t_0$. The performance evaluation in the two-sample test relies on two types of errors. Type-I error occurs when $s_0 = t_0$, but the algorithm predicts otherwise. Type-II error occurs when the algorithm incorrectly predicts $s_0 = t_0$. The probability of Type-I error is called the significance level. The significance level can be controlled using permutation test-based setups (Ernst, 2004; Liu et al., 2020). Algorithms are typically compared based on the empirical estimate of their test power (higher is better), defined as the probability of not making a Type-II error and the average Type-I error (lower is better).

**Dataset and experimental setup.** Following (Liu et al., 2020), we consider the two sets of samples, one from the true MNIST (LeCun & Cortes, 2010) and another from fake MNIST generated by the DCGAN (Bian et al., 2019). The data lies in 1024 dimensions. We take an increasing number of samples ($N$) and compute the average test power over 100 pairs of sets for each value of $N$. We repeat the experiment 10 times and report the average test power in Table 2 for the significance level $\alpha = 0.05$. By the design of the test, the average Type-I error was upper-bounded, and we noted the Type-II error in our experiment. We detail the procedure for choosing the hyperparameters and the list of chosen hyperparameters for each method in Appendix C.2.

**Results.** In Table 2, we observe that MMD-UOT obtains the highest test power for all values of $N$. The average test power of MMD-UOT is $1.5 - 2.4$ times better than that of $\epsilon$KL-UOT across $N$. MMD-UOT also outperforms EMD and 2-Wasserstein, which suffer from the curse of dimensionality, for all values of $N$. Our results match the sample efficient MMD metric's result on increasing $N$ to 1000, but for lesser sample-size, MMD-UOT is always better than MMD.

Table 3: MMD distance (lower is better) between computed barycenter and the ground truth distribution. A sigma-heuristics based RBF kernel is used to compute the MMD distance. We observe that MMD-UOT's results are closer to the ground truth than the baselines' results at all timesteps.

| Timestep | MMD | $\epsilon$KL-UOT | MMD-UOT |
|---|---|---|---|
| $t_1$ | 0.375 | 0.391 | **0.334** |
| $t_2$ | 0.190 | 0.184 | **0.179** |
| $t_3$ | 0.125 | 0.138 | **0.116** |
| Avg. | 0.230 | 0.238 | **0.210** |

### 5.3 Single-Cell RNA Sequencing

We empirically evaluate the quality of our barycenter in the Single-cell RNA sequencing experiment. Single-cell RNA sequencing technique (scRNA-seq) helps us understand how the expression profile of the cells changes (Schiebinger et al., 2019). Barycenter estimation in the OT framework offers a principled approach to estimate the trajectory of a measure at an intermediate timestep $t$ ($t_i < t < t_j$) when we have measurements available only at $t_i$ (source) and $t_j$ (target) time steps.

**Dataset and experimental setup.** We perform experiments on the Embryoid Body (EB) single-cell dataset (Moon et al., 2019). The dataset has samples available at five timesteps ($t_j$ with $j = 0, \ldots, 4$), which were collected during a 25-day period of development of the human embryo. Following (Tong et al., 2020), we project the data onto two-dimensional space and associate uniform measures to the source and the target samples given at different timesteps. We consider the samples at timestep $t_i$ and $t_{i+2}$ as the samples from the source and target measures where $0 \leq i \leq 2$ and aim at estimating the measure at $t_i$ timestep as their barycenter with equal interpolation weights $\rho_1 = \rho_2 = 0.5$.

We compute the barycenters using MMD-UOT (11) and the $\epsilon$KL-UOT (Chizat et al., 2018; Liero et al., 2018) approaches. For both, a simplex constraint is used to cater to the case of uniform measures. We also compare against the empirical average of the source and target measures, which is the barycenter obtained with the MMD metric. The computed barycenter is evaluated against the measure corresponding to the ground truth samples available at the corresponding timestep. We compute the distance between the two using the MMD metric with RBF kernel (Gretton et al., 2012). The hyperparameters are chosen based on the leave-one-out validation protocol. More details and some additional results are in Appendix C.3.

**Results.** Table 3 shows that MMD-UOT achieves the lowest distance from the ground truth for all the timesteps, illustrating its superior interpolation quality.

### 5.4 Domain Adaptation in JUMBOT framework

OT has been widely employed in domain adaptation problems (Courty et al., 2017; Courty et al., 2017; Seguy et al., 2018; Damodaran et al., 2018). JUMBOT (Fatras et al., 2021) is a popular domain adaptation method based on $\epsilon$KL-UOT that outperforms OT-based baselines. JUMBOT's loss function involves a cross-entropy term and $\epsilon$KL-UOT discrepancy term between the source and target distributions. We showcase the utility of MMD-UOT (10) in the JUMBOT (Fatras et al., 2021) framework.

**Dataset and experimental setup:** We perform the domain adaptation experiment with and Digits datasets comprising of MNIST (LeCun & Cortes, 2010), M-MNIST (Ganin et al., 2016), SVHN (Netzer et al., 2011), USPS (Hull, 1994) datasets. We replace the $\epsilon$KL-UOT based loss with the MMD-UOT loss (10), keeping the other experimental set-up the same as JUMBOT. We obtain JUMBOT's result with $\epsilon$KL-UOT with the best-reported hyperparameters (Fatras et al., 2021). Following JUMBOT, we tune hyperparameters of MMD-UOT for the Digits experiment on USPS to MNIST (U$\mapsto$M) domain adaptation task and use the same hyperparameters for the rest of the domain adaptation tasks on Digits. More details are in Appendix C.4.

Table 4: Target domain accuracy (higher is better) obtained in domain adaptation experiments. Results for $\epsilon$KL-UOT are reproduced from the code open-sourced for JUMBOT in (Fatras et al., 2021). MMD-UOT outperforms $\epsilon$KL-UOT in all the domain adaptation tasks considered.

| Source | Target | $\epsilon$KL-UOT | MMD-UOT |
|--------|--------|---------|---------|
| M-MNIST | USPS | 91.53 | **94.97** |
| M-MNIST | MNIST | 99.35 | **99.50** |
| MNIST | M-MNIST | 96.51 | **96.96** |
| MNIST | USPS | 96.51 | **97.01** |
| SVHN | M-MNIST | 94.26 | **95.35** |
| SVHN | MNIST | 98.68 | **98.98** |
| SVHN | USPS | 92.78 | **93.22** |
| USPS | MNIST | 96.76 | **98.53** |
| Avg. | | 95.80 | **96.82** |

**Results:** Table 4 reports the accuracy obtained on target datasets. We observe that MMD-UOT-based loss performs better than $\epsilon$KL-UOT-based loss for all the domain adaptation tasks. In Figure 8 (appendix), we also compare the t-SNE plot of the embeddings learned with the MMD-UOT and the $\epsilon$KL-UOT-based loss functions. The clusters learned with MMD-UOT are better separated (e.g., red- and cyan-colored clusters).

### 5.5 More Results on Domain Adaptation

In Section 5.4, we compared the proposed MMD-UOT-based loss function with the $\epsilon$KL-UOT based loss function in the JUMBOT framework (Fatras et al., 2021). It should be noted that JUMBOT has a ResNet-50 backbone. Hence, in this section, we also compare with popular domain adaptation baselines having ResNet-50 backbone. These include DANN (Ganin et al., 2015), CDANN-E (Long et al., 2017), DEEPJ-DOT (Damodaran et al., 2018), ALDA (Chen et al., 2020a), ROT (Balaji et al., 2020), and BombOT (Nguyen et al., 2022). BombOT is a recent state-of-the-art OT-based method for unsupervised domain adaptation (UDA). As in JUMBOT (Fatras et al., 2021), BombOT also employs $\epsilon$KL-UOT based loss function. We also include the results of the baseline ResNet-50 model, where the model is trained on the source and is evaluated on the target without employing any adaptation techniques.

**Office-Home dataset:** We evaluate the proposed method on the Office-Home dataset (Venkateswara et al., 2017), popular for unsupervised domain adaptation. We use the backbone network of ResNet-50 following. The Office-Home dataset has 15,500 images from four domains: Artistic images (A), Clip Art (C), Product images (P) and Real-World (R). The dataset contains images of 65 object categories common in office and home scenarios for each domain. Following (Fatras et al., 2021; Nguyen et al., 2022), evaluation is done in 12 adaptation tasks. Following JUMBOT, we validate the proposed method on the A→C task and use the chosen hyperparameters for the rest of the tasks.

Table 5 reports the target accuracies obtained by different methods. The results of the BombOT method are quoted from (Nguyen et al., 2022), and the results of other baselines are quoted from (Fatras et al., 2021). We observe that the proposed MMD-UOT-based method achieves the best target accuracy in 11 out of 12 adaptation tasks.

**VisDA-2017 dataset:** We next consider the next domain adaptation task between the training and validation sets of the VisDA-2017 (Recht et al., 2018) dataset. We follow the experimental setup detailed in (Fatras et al., 2021). The source domain of VisDA has 152,397 synthetic images, while the target domain has 55,388 real-world images. Both the domains have 12 object categories.

Table 6 compares the performance of different methods. The results of the BombOT method are quoted from (Nguyen et al., 2022), and the results of other baselines are quoted from (Fatras et al., 2021). The proposed

Table 5: Target accuracies (higher is better) on the Office-Home dataset in the UDA setting. The letters denote different domains: 'A' for Artistic images, 'P' for Product images, 'C' for Clip art and 'R' for Real-World images. The proposed method achieves the highest accuracy on almost all the domain adaptation tasks and achieves the best accuracy averaged across the tasks.

| Method | A→C | A→P | A→R | C→A | C→P | C→R | P→A | P→C | P→R | R→A | R→C | R→P | **Avg** |
|---|---|---|---|---|---|---|---|---|---|---|---|---|---|
| ResNet-50 | 34.9 | 50.0 | 58.0 | 37.4 | 41.9 | 46.2 | 38.5 | 31.2 | 60.4 | 53.9 | 41.2 | 59.9 | 46.1 |
| DANN (Ganin et al., 2015) | 44.3 | 59.8 | 69.8 | 48.0 | 58.3 | 63.0 | 49.7 | 42.7 | 70.6 | 64.0 | 51.7 | 78.3 | 58.3 |
| CDAN-E (Long et al., 2017) | 52.5 | 71.4 | 76.1 | 59.7 | 69.9 | 71.5 | 58.7 | 50.3 | 77.5 | 70.5 | 57.9 | 83.5 | 66.6 |
| DEEPJDOT (Damodaran et al., 2018) | 50.7 | 68.7 | 74.4 | 59.9 | 65.8 | 68.1 | 55.2 | 46.3 | 73.8 | 66.0 | 54.9 | 78.3 | 63.5 |
| ALDA (Chen et al., 2020a) | 52.2 | 69.3 | 76.4 | 58.7 | 68.2 | 71.1 | 57.4 | 49.6 | 76.8 | 70.6 | 57.3 | 82.5 | 65.8 |
| ROT (Balaji et al., 2020) | 47.2 | 71.8 | 76.4 | 58.6 | 68.1 | 70.2 | 56.5 | 45.0 | 75.8 | 69.4 | 52.1 | 80.6 | 64.3 |
| $\epsilon$KL-UOT (JUMBOT) (Fatras et al., 2021) | 55.2 | 75.5 | 80.8 | 65.5 | 74.4 | 74.9 | 65.2 | 52.7 | 79.2 | 73.0 | 59.9 | 83.4 | 70.0 |
| BombOT (Nguyen et al., 2022) | 56.2 | 75.2 | 80.5 | 65.8 | 74.6 | 75.4 | 66.2 | 53.2 | 80.0 | 74.2 | **60.1** | 83.3 | 70.4 |
| Proposed | **56.5** | **77.2** | **82.0** | **70.0** | **77.1** | **77.8** | **69.3** | **55.1** | **82.0** | **75.5** | 59.3 | **84.0** | **72.2** |

Table 6: Target accuracies (higher is better) on the VisDA-2017 dataset in the UDA setting. The proposed MMD-UOT method achieves the highest accuracy.

| Dataset | CDAN-E | ALDA | DEEPJDOT | ROT | $\epsilon$KL-UOT (JUMBOT) | BombOT | Proposed |
|---|---|---|---|---|---|---|---|
| VisDA-2017 | 70.1 | 70.5 | 68.0 | 66.3 | 72.5 | 74.6 | **77.0** |

method achieves the best performance, improving the accuracy obtained by $\epsilon$KL-UOT based JUMBOT and BombOT methods by 4.5% and 2.4%, respectively.

## 5.6 Prompt Learning for Few-Shot Classification

The task of learning prompts (e.g. "a tall bird of [class]") for vision-language models has emerged as a promising approach to adapt large pre-trained models like CLIP (Radford et al., 2021) for downstream tasks. The similarity between prompt features (which are class-specific) and visual features of a given image can help us classify the image. A recent OT-based prompt learning approach, PLOT (Chen et al., 2023), obtained state-of-the-art results on the $K$-shot recognition task in which only $K$ images per class are available during training. We evaluate the performance of MMD-UOT following the setup of (Chen et al., 2023) on the benchmark EuroSAT (Helber et al., 2018) dataset consisting of satellite images, DTD (Cimpoi et al., 2014) dataset having images of textures and Oxford-Pets (Parkhi et al., 2012) dataset having images of pets.

**Results** With the same evaluation protocol as in (Chen et al., 2023), we report the classification accuracy averaged over three seeds in Table 7. We note that MMD-UOT-based prompt-learning achieves better results than PLOT, especially when $K$ is less (more challenging case due to lesser training data). With the EuroSAT dataset, the improvement is as high as 4% for a challenging case of $K$=1. More details are in Appendix C.5.

Table 7: Average and standard deviation (over 3 runs) of accuracy (higher is better) on the $k$-shot classification task, shown for different values of shots ($k$) in the state-of-the-art PLOT framework. The proposed method replaces OT with MMD-UOT in PLOT, keeping all other hyperparameters the same. The results of PLOT are taken from their paper (Chen et al., 2023).

| Dataset | Method | 1 | 2 | 4 | 8 | 16 |
|---------|--------|---|---|---|---|-----|
| EuroSAT | PLOT | $54.05 \pm 5.95$ | $64.21 \pm 1.90$ | $\mathbf{72.36 \pm 2.29}$ | $78.15 \pm 2.65$ | $82.23 \pm 0.91$ |
|  | Proposed | $\mathbf{58.47 \pm 1.37}$ | $\mathbf{66.0 \pm 0.93}$ | $71.97 \pm 2.21$ | $\mathbf{79.03 \pm 1.91}$ | $\mathbf{83.23 \pm 0.24}$ |
| DTD | PLOT | $46.55 \pm 2.62$ | $\mathbf{51.24 \pm 1.95}$ | $56.03 \pm 0.43$ | $61.70 \pm 0.35$ | $65.60 \pm 0.82$ |
|  | Proposed | $\mathbf{47.27 \pm 1.46}$ | $51.0 \pm 1.71$ | $\mathbf{56.40 \pm 0.73}$ | $\mathbf{63.17 \pm 0.69}$ | $\mathbf{65.90 \pm 0.29}$ |

## 6 Conclusion

The literature on unbalanced optimal transport (UOT) has largely focused on $\phi$-divergence-based regularization. Our work provides a comprehensive analysis of MMD-regularization in UOT, answering many open questions. We prove novel results on the metricity and the sample efficiency of MMD-UOT, propose consistent estimators which can be computed efficiently, and illustrate its empirical effectiveness on several machine learning applications. Our theoretical and empirical contributions for MMD-UOT and its corresponding barycenter demonstrate the potential of MMD-regularization in UOT as an effective alternative to $\phi$-divergence-based regularization. Interesting directions of future work include exploring applications of IPM-regularized UOT (Remark 4.9) and the generalization of Kantorovich-Rubinstein duality (Remark 4.7).

## 7 Funding Disclosure and Acknowledgements

We thank Kilian Fatras for the discussions on the JUMBOT baseline, and Bharath Sriperumbudur (PSU) and G. Ramesh (IITH) for discussions related to Appendix B.9.1. We are grateful to Rudraram Siddhi Vinayaka. We also thank the anonymous reviewers for constructive feedback. PM and JSN acknowledge the support of Google PhD Fellowship and Fujitsu Limited (Japan), respectively.

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

# A  Preliminaries

## A.1  Integral Probability Metric (IPM):

Given a set $\mathcal{G} \subset \mathcal{L}(\mathcal{X})$, the integral probability metric (IPM) (Muller, 1997; Sriperumbudur et al., 2009; Agrawal & Horel, 2020) associated with $\mathcal{G}$, is defined by:

$$\gamma_{\mathcal{G}}(s_0, t_0) \equiv \max_{f \in \mathcal{G}} \left| \int_{\mathcal{X}} f \ \mathrm{d}s_0 - \int_{\mathcal{X}} f \ \mathrm{d}t_0 \right| \ \forall \ s_0, t_0 \in \mathcal{R}^+(\mathcal{X}). \tag{12}$$

$\mathcal{G}$ is called the generating set of the IPM, $\gamma_{\mathcal{G}}$.

In order that the IPM metrizes weak convergence, we assume the following (Muller, 1997):

**Assumption A.1.** $\mathcal{G} \subseteq \mathcal{C}(\mathcal{X})$ *and is compact.*

Since the IPM generated by $\mathcal{G}$ and its absolute convex hull is the same (without loss of generality), we additionally assume the following:

**Assumption A.2.** $\mathcal{G}$ *is absolutely convex.*

**Remark A.3.** *We note that the assumptions A.1 and A.2 are needed only to generalize our theoretical results to an IPM-regularized UOT formulation (Formulation 13). These assumptions are satisfied whenever the IPM employed for regularization is the MMD (Formulation 6) with a kernel that is continuous and universal (i.e., c-universal).*

## A.2  Classical Examples of IPMs

- **Maximum Mean Discrepancy (MMD):** Let $k$ be a characteristic kernel (Sriperumbudur et al., 2011) over the domain $\mathcal{X}$, let $\|f\|_k$ denote the norm of $f$ in the canonical reproducing kernel Hilbert space (RKHS), $\mathcal{H}_k$, corresponding to $k$. $\mathrm{MMD}_k$ is the IPM associated with the generating set: $\mathcal{G}_k \equiv \{f \in \mathcal{H}_k | \ \|f\|_k \leq 1\}$.

$$\mathrm{MMD}_k(s_0, t_0) \equiv \max_{f \in \mathcal{G}_k} \left| \int_{\mathcal{X}} f \ \mathrm{d}s_0 - \int_{\mathcal{X}} f \ \mathrm{d}t_0 \right|.$$

- **Kantorovich metric ($\mathcal{K}_c$):** Kantorovich metric also belongs to the family of integral probability metrics associated with the generating set $\mathcal{W}_c \equiv \left\{ f : \mathcal{X} \mapsto \mathbb{R} \ | \ \max_{x \in \mathcal{X} \neq y \in \mathcal{X}} \frac{|f(x) - f(y)|}{c(x,y)} \leq 1 \right\}$, where $c$ is a metric over $\mathcal{X}$. The Kantorovich-Fenchel duality result shows that the 1-Wasserstein metric is the same as the Kantorovich metric when restricted to probability measures.

- **Dudley:** This is the IPM associated with the generating set: $\mathcal{D}_d \equiv \{f : \mathcal{X} \mapsto \mathbb{R} \ | \ \|f\|_\infty + \|f\|_d \leq 1\}$, where $d$ is a ground metric over $\mathcal{X} \times \mathcal{X}$. The so-called **Flat** metric is related to the Dudley metric. It's generating set is: $\mathcal{F}_d \equiv \{f : \mathcal{X} \mapsto \mathbb{R} \ | \ \|f\|_\infty \leq 1, \|f\|_d \leq 1\}$.

- **Kolmogorov:** Let $\mathcal{X} = \mathbb{R}^n$. Then, the Kolmogorov metric is the IPM associated with the generating set: $\bar{\mathcal{K}} \equiv \left\{ 1_{(-\infty, x)} \ | \ x \in \mathbb{R}^n \right\}$.

- **Total Variation (TV):** This is the IPM associated with the generating set: $\mathcal{T} \equiv \{f : \mathcal{X} \mapsto \mathbb{R} \mid \|f\|_\infty \leq 1\}$, where $\|f\|_\infty \equiv \max\limits_{x \in \mathcal{X}} |f(x)|$. Total Variation metric over measures $s_0, t_0 \in \mathcal{R}^+(\mathcal{X})$ is defined as:

$$\text{TV}(s,t) \equiv \int_{\mathcal{Y}} \mathrm{d}|s-t|(y), \text{ where } |s-t|(y) \equiv \begin{cases} s(y) - t(y) & \text{if } s(y) \geq t(y) \\ t(y) - s(y) & \text{otherwise} \end{cases}$$

## B  Proofs and Additional Theory Results

As mentioned in the main paper and Remark 4.9, most of our proofs hold even with a general IPM-regularized UOT formulation (13) under mild assumptions. **We restate such results and give a general proof that holds for IPM-regularized UOT (Formulation 13), of which MMD-regularized UOT (Formulation 6) is a special case.**

The proposed **IPM-regularized UOT** formulation is presented as follows.

$$\mathcal{U}_{\mathcal{G},c,\lambda_1,\lambda_2}(s_0,t_0) \equiv \min_{\pi \in \mathcal{R}^+(\mathcal{X} \times \mathcal{X})} \int c \,\mathrm{d}\pi + \lambda_1 \gamma_{\mathcal{G}}(\pi_1, s_0) + \lambda_2 \gamma_{\mathcal{G}}(\pi_2, t_0), \tag{13}$$

where $\gamma_{\mathcal{G}}$ is defined in equation (12).

We now present the theoretical results and proofs with IPM-regularized UOT (Formulation 13), of which MMD-regularized UOT (Formulation 6) is a special case. To the best of our knowledge, such an analysis for IPM-regularized UOT has not been done before.

### B.1  Proof of Theorem 4.1

**Theorem 4.1.** *(Duality) Whenever $\mathcal{G}$ satisfies Assumptions A.1 and A.2, $c, k \in \mathcal{C}(\mathcal{X} \times \mathcal{X})$ and $\mathcal{X}$ is compact, we have that:*

$$\mathcal{U}_{\mathcal{G},c,\lambda_1,\lambda_2}(s_0,t_0) = \max_{f \in \mathcal{G}(\lambda_1), g \in \mathcal{G}(\lambda_2)} \int_{\mathcal{X}} f \,\mathrm{d}s_0 + \int_{\mathcal{X}} g \,\mathrm{d}t_0,$$

$$\text{s.t.} \quad f(x) + g(y) \leq c(x,y) \ \forall \ x,y \in \mathcal{X}. \tag{14}$$

*Proof.* We begin by re-writing the RHS of (13) using the definition of IPMs given in (12):

$$\mathcal{U}_{\mathcal{G},c,\lambda_1,\lambda_2}(s_0,t_0) \equiv \min_{\pi \in \mathcal{R}^+(\mathcal{X} \times \mathcal{X})} \int_{\mathcal{X} \times \mathcal{X}} c \,\mathrm{d}\pi + \lambda_1 \left( \max_{f \in \mathcal{G}} \left| \int_{\mathcal{X}} f \,\mathrm{d}s_0 - \int_{\mathcal{X}} f \,\mathrm{d}\pi_1 \right| \right) + \lambda_2 \left( \max_{g \in \mathcal{G}} \left| \int_{\mathcal{X}} g \,\mathrm{d}t_0 - \int_{\mathcal{X}} g \,\mathrm{d}\pi_2 \right| \right)$$

$$\overset{\because (A.2)}{=} \min_{\pi \in \mathcal{R}^+(\mathcal{X} \times \mathcal{X})} \int_{\mathcal{X} \times \mathcal{X}} c \,\mathrm{d}\pi + \lambda_1 \left( \max_{f \in \mathcal{G}} \int_{\mathcal{X}} f \,\mathrm{d}s_0 - \int_{\mathcal{X}} f \,\mathrm{d}\pi_1 \right) + \lambda_2 \left( \max_{g \in \mathcal{G}} \int_{\mathcal{X}} g \,\mathrm{d}t_0 - \int_{\mathcal{X}} g \,\mathrm{d}\pi_2 \right)$$

$$= \min_{\pi \in \mathcal{R}^+(\mathcal{X} \times \mathcal{X})} \int_{\mathcal{X} \times \mathcal{X}} c \,\mathrm{d}\pi + \left( \max_{f \in \mathcal{G}(\lambda_1)} \int_{\mathcal{X}} f \,\mathrm{d}s_0 - \int_{\mathcal{X}} f \,\mathrm{d}\pi_1 \right) + \left( \max_{g \in \mathcal{G}(\lambda_2)} \int_{\mathcal{X}} g \,\mathrm{d}t_0 - \int_{\mathcal{X}} g \,\mathrm{d}\pi_2 \right)$$

$$= \max_{f \in \mathcal{G}(\lambda_1), g \in \mathcal{G}(\lambda_2)} \int_{\mathcal{X}} f \,\mathrm{d}s_0 + \int_{\mathcal{X}} g \,\mathrm{d}t_0 + \min_{\pi \in \mathcal{R}^+(\mathcal{X} \times \mathcal{X})} \int_{\mathcal{X} \times \mathcal{X}} c \,\mathrm{d}\pi - \int_{\mathcal{X}} f \,\mathrm{d}\pi_1 - \int_{\mathcal{X}} g \,\mathrm{d}\pi_2$$

$$= \max_{f \in \mathcal{G}(\lambda_1), g \in \mathcal{G}(\lambda_2)} \int_{\mathcal{X}} f \,\mathrm{d}s_0 + \int_{\mathcal{X}} g \,\mathrm{d}t_0 + \min_{\pi \in \mathcal{R}^+(\mathcal{X} \times \mathcal{X})} \int_{\mathcal{X} \times \mathcal{X}} c - \bar{f} - \bar{g} \,\mathrm{d}\pi$$

$$= \max_{f \in \mathcal{G}(\lambda_1), g \in \mathcal{G}(\lambda_2)} \int_{\mathcal{X}} f \,\mathrm{d}s_0 + \int_{\mathcal{X}} g \,\mathrm{d}t_0 + \begin{cases} 0 & \text{if } f(x) + g(y) \leq c(x,y) \ \forall \ x,y \in \mathcal{X}, \\ -\infty & \text{otherwise.} \end{cases}$$

$$= \max_{f \in \mathcal{G}(\lambda_1), g \in \mathcal{G}(\lambda_2)} \int_{\mathcal{X}} f \,\mathrm{d}s_0 + \int_{\mathcal{X}} g \,\mathrm{d}t_0,$$

$$\text{s.t.} \quad f(x) + g(y) \leq c(x,y) \ \forall \ x,y \in \mathcal{X}. \tag{15}$$

Here, $\bar{f}(x,y) \equiv f(x)$, $\bar{g}(x,y) \equiv g(y)$. The min-max interchange in the third equation is due to Sion's minimax theorem: (i) since $\mathcal{R}(\mathcal{X})$ is a topological dual of $\mathcal{C}(\mathcal{X})$ whenever $\mathcal{X}$ is compact, the objective is

bilinear (inner-product in this duality), whenever $c, f, g$ are continuous. This is true from Assumption A.1 and $c \in \mathcal{C}(\mathcal{X} \times \mathcal{X})$. (ii) one of the feasibility sets involves $\mathcal{G}$, which is convex compact by Assumptions A.1, A.2. The other feasibility set is convex (the closed conic set of non-negative measures). □

**Remark B.1.** *Whenever the kernel, $k$, employed is continuous, the generating set of the corresponding MMD satisfies assumptions A.2 and $\mathcal{G}_k \subseteq \mathcal{C}(\mathcal{X})$. Hence, the above proof also works in our case of MMD-regularized UOT (i.e., to prove Theorem 4.1 in the main paper).*

## B.2 Proof of Corollary 4.2

We first derive an equivalent re-formulation of 13, which will be used in our proof.

**Lemma B1.**
$$\mathcal{U}_{\mathcal{G},c,\lambda_1,\lambda_2}(s_0, t_0) \equiv \min_{s,t \in \mathcal{R}^+(\mathcal{X})} |s| W_1(s,t) + \lambda_1 \gamma_{\mathcal{G}}(s, s_0) + \lambda_2 \gamma_{\mathcal{G}}(t, t_0), \tag{16}$$

*where $W_1(s,t) \equiv \begin{cases} \bar{W}_1(\frac{s}{|s|}, \frac{t}{|t|}) & \text{if } |s| = |t|, \\ \infty & \text{otherwise.} \end{cases}$, with $\bar{W}_1$ as the 1-Wasserstein metric.*

*Proof.*

$$\min_{s,t \in \mathcal{R}^+(\mathcal{X})} |s| W_1(s,t) + \lambda_1 \gamma_{\mathcal{G}}(s, s_0) + \lambda_2 \gamma_{\mathcal{G}}(t, t_0)$$

$$= \min_{s,t \in \mathcal{R}^+(\mathcal{X}); \ |s|=|t|} |s| \min_{\bar{\pi} \in \mathcal{R}_1^+(\mathcal{X} \times \mathcal{X})} \int c \, d\bar{\pi} + \lambda_1 \gamma_{\mathcal{G}}(s, s_0) + \lambda_2 \gamma_{\mathcal{G}}(t, t_0) \ \text{ s.t. } \bar{\pi}_1 = \frac{s}{|s|}, \bar{\pi}_2 = \frac{t}{|t|}$$

$$= \min_{\eta > 0} \eta \min_{\bar{\pi} \in \mathcal{R}_1^+(\mathcal{X} \times \mathcal{X})} \int c \, d\bar{\pi} + \lambda_1 \gamma_{\mathcal{G}}(\eta \bar{\pi}_1, s_0) + \lambda_2 \gamma_{\mathcal{G}}(\eta \bar{\pi}_2, t_0)$$

$$= \min_{\eta > 0} \min_{\bar{\pi} \in \mathcal{R}_1^+(\mathcal{X} \times \mathcal{X})} \int c \, \eta d\bar{\pi} + \lambda_1 \gamma_{\mathcal{G}}(\eta \bar{\pi}_1, s_0) + \lambda_2 \gamma_{\mathcal{G}}(\eta \bar{\pi}_2, t_0)$$

$$= \min_{\pi \in \mathcal{R}^+(\mathcal{X} \times \mathcal{X})} \int c \, d\pi + \lambda_1 \gamma_{\mathcal{G}}(\pi_1, s_0) + \lambda_2 \gamma_{\mathcal{G}}(\pi_2, t_0)$$

The first equality holds from the definition of $W_1$: $W_1(s,t) \equiv \begin{cases} \bar{W}_1(\frac{s}{|s|}, \frac{t}{|t|}) & \text{if } |s| = |t|, \\ \infty & \text{otherwise.} \end{cases}$. Eliminating normalized versions $s$ and $t$ using the equality constraints and introducing $\eta$ to denote their common mass gives the second equality. The last equality comes after changing the variable of optimization to $\pi \in \mathcal{R}^+(\mathcal{X} \times \mathcal{X}) \equiv \eta \bar{\pi}$. Recall that $\mathcal{R}^+(\mathcal{X})$ denotes the set of all non-negative Radon measures defined over $\mathcal{X}$; while the set of all probability measures is denoted by $\mathcal{R}_1^+(\mathcal{X})$. □

Corollary 4.2 in the main paper is restated below with the IPM-regularized UOT formulation (13), followed by its proof.

**Corollary 4.2.** *(Metricity) In addition to assumptions in Theorem (4.1), whenever $c$ is a metric, $\mathcal{U}_{\mathcal{G},c,\lambda,\lambda}$ belongs to the family of integral probability metrics (IPMs). Also, the generating set of this IPM is the intersection of the generating set of the Kantorovich metric and the generating set of the IPM used for regularization. Finally, $\mathcal{U}_{\mathcal{G},c,\lambda,\lambda}$ is a valid norm-induced metric over measures whenever the IPM used for regularization is norm-induced (e.g. MMD with a characteristic kernel). Thus, $\mathcal{U}$ lifts the ground metric $c$ to that over measures.*

*Proof.* The constraints in dual, (7), are equivalent to: $g(y) \leq \min_{x \in \mathcal{X}} c(x, y) - f(x) \ \forall \ y \in \mathcal{X}$. The RHS is nothing but the $c$-conjugate ($c$-transform) of $f$. From Proposition 6.1 in (Peyré & Cuturi, 2019), whenever $c$ is a metric we have: $\min_{x \in \mathcal{X}} c(x, y) - f(x) = \begin{cases} -f(y) & \text{if } f \in \mathcal{W}_c, \\ -\infty & \text{otherwise.} \end{cases}$ Here, $\mathcal{W}_c$ is the generating set of the Kantorovich metric lifting $c$. Thus the constraints are equivalent to: $g(y) \leq -f(y) \ \forall \ y \in \mathcal{X}, f \in \mathcal{W}_c$.

Now, since the dual, (7), seeks to maximize the objective with respect to $g$, and monotonically increases with values of $g$; at optimality, we have that $g(y) = -f(y) \ \forall \ y \in \mathcal{X}$. Note that this equality is possible to achieve as both $g, -f \in \mathcal{G}(\lambda) \cap \mathcal{W}_c$ (these sets are absolutely convex). Eliminating $g$, one obtains:

$$\mathcal{U}_{\mathcal{G},c,\lambda,\lambda}(s_0, t_0) = \max_{f \in \mathcal{G}(\lambda) \cap \mathcal{W}_c} \int_{\mathcal{X}} f \ \mathrm{d}s_0 - \int_{\mathcal{X}} f \ \mathrm{d}t_0,$$

Comparing this and the definition of IPMs 12, we have that $\mathcal{U}_{\mathcal{G},c,\lambda,\lambda}$ belongs to the family of IPMs. Since any IPM is a pseudo-metric (induced by a semi-norm) over measures (Muller, 1997), the only condition left to be proved is positive definiteness with $\mathcal{U}_{\mathcal{G},c,\lambda,\lambda}(s_0, t_0)$. Following Lemma B1, we have that for optimal $s^*, t^*$ in (16), $\mathcal{U}_{\mathcal{G},c,\lambda,\lambda}(s_0, t_0) = 0 \iff (i) \ W_1(s^*, t^*) = 0, (ii) \ \gamma_{\mathcal{G}}(s^*, s_0) = 0, (iii) \ \gamma_{\mathcal{G}}(t^*, t_0) = 0$ as each term in the RHS is non-negative. When the IPM used for regularization is a norm-induced metric (e.g. the MMD metric or the Dudley metric), the conditions $(i), (ii), (iii) \iff s^* = t^* = s_0 = t_0$, which proves the positive definiteness. Hence, we proved that $\mathcal{U}_{\mathcal{G},c,\lambda,\lambda}$ is a norm-induced metric over measures whenever the IPM used for regularization is a metric. $\qquad\square$

**Remark B.2.** *Recall that MMD is a valid norm-induced IPM metric whenever the kernel employed is characteristic. Hence, our proof above also shows the metricity of the MMD-regularized UOT (as per corollary 4.2 in the main paper).*

**Remark B.3.** *If $\mathcal{G}$ is the unit uniform-norm ball (corresponding to TV), our result specializes to that in (Piccoli & Rossi, 2016), which proves that $\mathcal{U}_{\mathcal{G},c,\lambda,\lambda}$ coincides with the so-called Flat metric (or the bounded Lipschitz distance).*

**Remark B.4.** *If the regularizer is the Kantorovich metric[3], i.e., $\mathcal{G} = \mathcal{W}_c$, and $\lambda_1 = \lambda_2 = \lambda \geq 1$, then $\mathcal{U}_{\mathcal{W}_c,c,\lambda,\lambda}$ coincides with the Kantorovich metric. In other words, the Kantorovich-regularized OT is the same as the Kantorovich metric. Hence providing an OT interpretation for the Kantorovich metric that is valid for potentially un-normalized measures in $\mathcal{R}^+(\mathcal{X})$.*

### B.3 Proof of Corollary 4.3

*Proof.* As discussed in Theorem 4.1 and Corollary 4.2, the MMD-regularized UOT (Formulation 6) is an IPM with the generating set as an intersection of the generating sets of the MMD and the Kantorovich-Wasserstein metrics. We now present special cases when MMD-regularized UOT (Formulation 6) recovers back the Kantorovich-Wasserstein metric and the MMD metric.

**Recovering Kantorovich.** Recall that $\mathcal{G}_k(\lambda) = \{\lambda g \mid g \in \mathcal{G}_k\}$. From the definition of $\mathcal{G}_k(\lambda)$, $f \in \mathcal{G}_k(\lambda) \implies f \in \mathcal{H}_k$, $\|f\|_k \leq \lambda$. Hence, as $\lambda \to \infty, \mathcal{G}_k(\lambda) = \mathcal{H}_k$. Using this in the duality result of Theorem 4.1, we have the following.

$$\lim_{\lambda \to \infty} \mathcal{U}_{k,c,\lambda,\lambda}(s_0, t_0) = \lim_{\lambda \to \infty} \max_{f \in \mathcal{G}_k(\lambda) \cap \mathcal{W}_c} \int f \mathrm{d}s_0 - \int f \mathrm{d}t_0 = \max_{f \in \mathcal{H}_k \cap \mathcal{W}_c} \int f \mathrm{d}s_0 - \int f \mathrm{d}t_0$$
$$\overset{(1)}{=} \max_{f \in \mathcal{C}(\mathcal{X}) \cap \mathcal{W}_c} \int f \mathrm{d}s_0 - \int f \mathrm{d}t_0$$
$$\overset{(2)}{=} \max_{f \in \mathcal{W}_c} \int f \mathrm{d}s_0 - \int f \mathrm{d}t_0$$

Equality (1) holds because $\mathcal{H}_k$ is dense in the set of continuous functions, $\mathcal{C}(\mathcal{X})$. For equality (2), we use that $\mathcal{W}_c$ consists of only 1-Lipschitz continuous functions. Thus, $\forall s_0, t_0 \in \mathcal{R}^+(\mathcal{X})$, $\lim_{\lambda \to \infty} \mathcal{U}_{k,c,\lambda,\lambda}(s_0, t_0) = \mathcal{K}_c(s_0, t_0)$.

**Recovering MMD.** We next show that when $0 < \lambda_1 = \lambda_2 = \lambda \leq 1$ and the cost metric $c$ is such that $c(x, y) \geq \sqrt{k(x, x) + k(y, y) - 2k(x, y)} = \|\phi(x) - \phi(y)\|_k \ \forall x, y$ (Dominating cost assumption discussed in B.4), then $\forall s_0, t_0 \in \mathcal{R}^+(\mathcal{X})$, $\mathcal{U}_{k,c,\lambda,\lambda}(s_0, t_0) = \mathrm{MMD}_k(s_0, t_0)$.

---

[3]The ground metric in $\mathcal{U}_{\mathcal{G},c,\lambda,\lambda}$ must be the same as that defining the Kantorovich regularizer.

Let $f \in \mathcal{G}_k(\lambda) \implies f = \lambda g$ where $g \in \mathcal{H}_k, \|g\| \le 1$. This also implies that $\lambda g \in \mathcal{H}_k$ as $\lambda \in (0, 1]$.

$$
\begin{aligned}
|f(x) - f(y)| &= |\langle \lambda g, \phi(x) - \phi(y) \rangle| \text{ (RKHS property)} \\
&\le |\langle g, \phi(x) - \phi(y) \rangle| \ (\because 0 < \lambda \le 1) \\
&\le \|g\|_k \|\phi(x) - \phi(y)\|_k \text{ (Cauchy Schwarz)} \\
&\le \|\phi(x) - \phi(y)\|_k \ (\because \|g\| \le 1) \\
&\le c(x, y) \text{ (Dominating cost assumption, discussed in B.4)} \\
&\implies f \in \mathcal{W}_c
\end{aligned}
$$

Therefore, $\mathcal{G}_k(\lambda) \subseteq \mathcal{W}_C$ and hence, $\mathcal{G}_k(\lambda) \cap \mathcal{W}_C = \mathcal{G}_k(\lambda)$. This relation, together with the metricity result shown in Corollary 4.2, implies that $\mathcal{U}_{k,c,\lambda,\lambda}(s_0, t_0) = \lambda \mathrm{MMD}_k(s_0, t_0)$. In B.4, we show that the Euclidean distance satisfies the dominating cost assumption when the kernel employed is the Gaussian kernel and the inputs lie on a unit-norm ball. $\qquad\square$

## B.4 Dominating Cost Assumption with Euclidean cost and Gaussian Kernel

We present a sufficient condition for the Dominating cost assumption (used in Corollary 4.3) to be satisfied while using a Euclidean cost and a Gaussian kernel based MMD. We consider the characteristic RBF kernel, $k(x, y) = \exp\left(-s\|x - y\|^2\right)$, and show that for the hyper-parameter, $0 < s \le 0.5$, the Euclidean cost is greater than the Kernel cost when the inputs are normalized, i.e., $\|x\| = \|y\| = 1$.

$$
\begin{aligned}
\|x - y\|^2 &\ge k(x, x) + k(y, y) - 2k(x, y) \\
\iff \|x\|^2 + \|y\|^2 - 2\langle x, y \rangle &\ge 2 - 2k(x, y) \\
\iff \langle x, y \rangle &\le \exp\left(-2s(1 - \langle x, y \rangle)\right) \text{ (Assuming normalized inputs)}
\end{aligned}
\tag{17}
$$

From Cauchy Schwarz inequality, $-\|x\|\|y\| \le \langle x, y \rangle \le \|x\|\|y\|$. With the assumption of normalized inputs, we have that $-1 \le \langle x, y \rangle \le 1$. We consider two cases based on this.

**Case 1:** $\langle x, y \rangle \in [-1, 0]$   In this case, condition (17) is satisfied $\forall s \ge 0$ because $k(x, y) \ge 0 \ \forall x, y$ with a Gaussian kernel.

**Case 2:** $\langle x, y \rangle \in (0, 1]$   In this case, our problem in condition (17) is to find $s \ge 0$ such that $\ln \langle x, y \rangle \le -2s(1 - \langle x, y \rangle)$. We further consider two sub-cases and derive the required condition as follows:

**Case 2A:** $\langle x, y \rangle \in \left(0, \frac{1}{e}\right]$   We re-parameterize $\langle x, y \rangle = e^{-n}$ for $n \ge 1$. With this, we need to find $s \ge 0$ such that $-n \le -2s(1 - e^{-n}) \iff n \ge 2s(1 - e^{-n})$. This is satisfied when $0 < s \le 0.5$ because $e^{-n} \ge 1 - n$.

**Case 2B:** $\langle x, y \rangle \in \left(\frac{1}{e}, \infty\right)$   We re-parameterize $\langle x, y \rangle = e^{-\frac{1}{n}}$ for $n > 1$. With this, we need to find $s \ge 0$ such that $\frac{1}{n\left(1 - e^{-\frac{1}{n}}\right)} \ge 2s$. We consider the function $f(n) = n\left(1 - e^{-\frac{1}{n}}\right)$ for $n \ge 1$. We now show that $f$ is an increasing function by showing that the gradient $\frac{df}{dn} = 1 - \left(1 + \frac{1}{n}\right)e^{-\frac{1}{n}}$ is always non-negative.

$$
\begin{aligned}
\frac{df}{dn} &\ge 0 \\
\iff e^{\frac{1}{n}} &\ge \left(1 + \frac{1}{n}\right) \\
\iff \frac{1}{n} - \ln\left(1 + \frac{1}{n}\right) &\ge 0 \\
\iff \frac{1}{n} - (\ln(n+1) - \ln(n)) &\ge 0
\end{aligned}
$$

Applying the Mean Value Theorem on $g(n) = \ln n$, we get

$$\ln(n+1) - \ln n = (n+1-n)\frac{1}{z}, \text{ where } n \le z \le n+1$$

$$\implies \ln\left(1 + \frac{1}{n}\right) = \frac{1}{z} \le \frac{1}{n}$$

$$\implies \frac{\mathrm{d}f}{\mathrm{d}n} = \frac{1}{n} - \ln\left(1 + \frac{1}{n}\right) \ge 0$$

The above shows that $f$ is an increasing function of $n$. We note that $\lim_{n\to\infty} f(n) = 1$, hence, $\frac{1}{f(n)} = \frac{1}{n\left(1 - e^{-\frac{1}{n}}\right)} \ge 1$ which implies that condition (17) is satisfied by taking $0 < s \le 0.5$.

## B.5 Proof of Corollary 4.4

Corollary 4.4 in the main paper is restated below with the IPM-regularized UOT formulation (13), followed by its proof.

**Corollary 4.4.** $\mathcal{U}_{\mathcal{G},c,\lambda,\lambda}(s,t) \le \min\left(\lambda\gamma_{\mathcal{G}}(s,t), \mathcal{K}_c(s,t)\right).$

*Proof.* Theorem 4.1 shows that $\mathcal{U}_{\mathcal{G},c,\lambda,\lambda}$ is an IPM whose generating set is the intersection of the generating sets of Kantorovich and the scaled version of the IPM used for regularization. Thus, from the definition of max, we have that $\mathcal{U}_{\mathcal{G},c,\lambda,\lambda}(s,t) \le \lambda\gamma_{\mathcal{G}}(s,t)$ and $\mathcal{U}_{\mathcal{G},c,\lambda,\lambda}(s,t) \le \mathcal{K}_c(s,t)$. This implies that $\mathcal{U}_{\mathcal{G},c,\lambda,\lambda}(s,t) \le \min\left(\lambda\gamma_{\mathcal{G}}(s,t), \mathcal{K}_c(s,t)\right)$. As a special case, $\mathcal{U}_{k,c,\lambda,\lambda}(s,t) \le \min\left(\lambda\mathrm{MMD}_k(s,t), \mathcal{K}_c(s,t)\right)$. $\square$

## B.6 Proof of Corollary 4.5

Corollary 4.5 in the main paper is restated below with the IPM-regularized UOT formulation (13), followed by its proof.

**Corollary 4.5.** *(Weak Metrization)* $\mathcal{U}_{\mathcal{G},c,\lambda,\lambda}$ *metrizes the weak convergence of normalized measures.*

*Proof.* For convenience of notation, we denote $\mathcal{U}_{\mathcal{G},c,\lambda,\lambda}$ by $\mathcal{U}$. From Corollary 4.4 in the main paper,

$$0 \le \mathcal{U}(\beta_n, \beta) \le \mathcal{K}_c(\beta_n, \beta)$$

From Sandwich theorem, $\lim_{\beta_n \to \beta} \mathcal{U}(\beta_n, \beta) \to 0$ as $\lim_{\beta_n \to \beta} \mathcal{K}_c(\beta_n, \beta)) \to 0$ by Theorem 6.9 in (Villani, 2009). $\square$

## B.7 Proof of Corollary 4.6

Corollary 4.6 in the main paper is restated below with the IPM-regularized UOT formulation (13), followed by its proof.

**Corollary 4.6.** *(Sample Complexity) Let us denote* $\mathcal{U}_{\mathcal{G},c,\lambda,\lambda}$, *defined in 13, by* $\bar{\mathcal{U}}$. *Let* $\hat{s}_m, \hat{t}_m$ *denote the empirical estimates of* $s_0, t_0 \in \mathcal{R}^+(\mathcal{X})$ *respectively with* $m$ *samples. Then,* $\bar{\mathcal{U}}(\hat{s}_m, \hat{t}_m) \to \bar{\mathcal{U}}(s_0, t_0)$ *at a rate (apart from constants) same as that of* $\gamma_{\mathcal{G}}(\hat{s}_m, s_0) \to 0$.

*Proof.* We use metricity of $\bar{\mathcal{U}}$ proved in Corrolary 4.2. From triangle inequality of the metric $\bar{\mathcal{U}}$ and Corollary 4.4 in the main paper, we have that
$0 \le |\bar{\mathcal{U}}(\hat{s}_m, \hat{t}_m) - \bar{\mathcal{U}}(s_0, t_0)| \le \bar{\mathcal{U}}(\hat{s}_m, s_0) + \bar{\mathcal{U}}(t_0, \hat{t}_m) \le \lambda\left(\gamma_{\mathcal{G}}(\hat{s}_m, s_0) + \gamma_{\mathcal{G}}(\hat{t}_m, t_0)\right)$.

Hence, by Sandwich theorem, $\bar{\mathcal{U}}(\hat{s}_m, \hat{t}_m) \to \bar{\mathcal{U}}(s_0, t_0)$ at a rate at which $\gamma_{\mathcal{G}}(\hat{s}_m, s_0) \to 0$ and $\gamma_{\mathcal{G}}(\hat{t}_m, t_0) \to 0$. If the IPM used for regularization is MMD with a normalized kernel, then $\mathrm{MMD}_k(s_0, \hat{s}_m) \le \sqrt{\frac{1}{m}} + \sqrt{\frac{2\log(1/\delta)}{m}}$ with probability at least $1 - \delta$ (Smola et al., 2007).

From the union bound, with probability at least $1 - \delta$, $|\bar{\mathcal{U}}(s_m, t_m) - \bar{\mathcal{U}}(s_0, t_0)| \le 2\lambda\left(\sqrt{\frac{1}{m}} + \sqrt{\frac{2\log(2/\delta)}{m}}\right)$. $\square$

### B.8 Proof of Theorem 4.8

We first restate the standard Moreau-Rockafellar theorem, which we refer to in this discussion.

**Theorem B2.** *Let $X$ be a real Banach space and $f, g : X \mapsto \mathbb{R} \cup \{\infty\}$ be closed convex functions such that $dom(f) \cap dom(g)$ is not empty, then: $(f+g)^*(y) = \min\limits_{x_1 + x_2 = y} f^*(x_1) + g^*(x_2) \; \forall y \in X^*$. Here, $f^*$ is the Fenchel conjugate of $f$, and $X^*$ is the topological dual space of $X$.*

Theorem 4.8 in the main paper is restated below with the IPM-regularized UOT formulation 13, followed by its proof.

**Theorem 4.8.** *In addition to the assumptions in Theorem 4.1, if $c$ is a valid metric, then*

$$\mathcal{U}_{\mathcal{G}, c, \lambda_1, \lambda_2}(s_0, t_0) = \min_{s, t \in \mathcal{R}(\mathcal{X})} \mathcal{K}_c(s, t) + \lambda_1 \gamma_{\mathcal{G}}(s, s_0) + \lambda_2 \gamma_{\mathcal{G}}(t, t_0). \tag{18}$$

*Proof.* Firstly, the result in the theorem is not straightforward and is not a consequence of Kantorovich-Rubinstein duality. This is because the regularization terms in our original formulation (13, 16) enforce closeness to the marginals of a transport plan and hence necessarily must be of the same mass and must belong to $\mathcal{R}^+(\mathcal{X})$. Whereas in the RHS of 18, the regularization terms enforce closeness to marginals that belong to $\mathcal{R}(\mathcal{X})$ and more importantly, they could be of different masses.

We begin the proof by considering indicator functions $F_c$ and $F_{\mathcal{G}}$ defined over $\mathcal{C}(\mathcal{X}) \times \mathcal{C}(\mathcal{X})$ as:
$F_c(f, g) = \left\{ \begin{array}{ll} 0 & \text{if } f(x) + g(y) \le c(x, y) \; \forall \; x, y \in \mathcal{X}, \\ \infty & \text{otherwise.} \end{array} \right.$ , $F_{\mathcal{G}, \lambda_1, \lambda_2}(f, g) = \left\{ \begin{array}{ll} 0 & \text{if } f \in \mathcal{G}(\lambda_1), g \in \mathcal{G}(\lambda_2), \\ \infty & \text{otherwise} \end{array} \right.$ .

Recall that the topological dual of $\mathcal{C}(\mathcal{X})$ is the set of regular Radon measures $\mathcal{R}(\mathcal{X})$ and the duality product $\langle f, s \rangle \equiv \int f \, ds \; \forall \; f \in \mathcal{C}(\mathcal{X}), s \in \mathcal{R}(\mathcal{X})$. Now, from the definition of Fenchel conjugate in the (direct sum) space $\mathcal{C}(\mathcal{X}) \oplus \mathcal{C}(\mathcal{X})$, we have: $F_c^*(s, t) = \max\limits_{f \in \mathcal{C}(\mathcal{X}), g \in \mathcal{C}(\mathcal{X})} \int f \, ds + \int g \, dt, \text{s.t.} f(x) + g(y) \le c(x, y) \; \forall \; x, y \in \mathcal{X}$, where $s, t \in \mathcal{R}(\mathcal{X})$. Under the assumptions that $\mathcal{X}$ is compact and $c$ is a continuous metric, Proposition 6.1 in (Peyré & Cuturi, 2019) shows that $F_c^*(s, t) = \max\limits_{f \in \mathcal{W}_c} \int f \, ds - \int f \, dt = \mathcal{K}_c(s, t)$.

On the other hand, $F_{\mathcal{G}, \lambda_1, \lambda_2}(f, g) = \left( \max\limits_{f \in \mathcal{G}(\lambda_1)} \int f \, ds + \max\limits_{g \in \mathcal{G}(\lambda_2)} \int g \, dt \right) = \lambda_1 \gamma_{\mathcal{G}}(s, 0) + \lambda_2 \gamma_{\mathcal{G}}(t, 0)$. Now, we have that the RHS of 18 is $\min\limits_{s, t, s_1, t_1 \in \mathcal{R}(\mathcal{X}):(s,t)+(s_1,t_1)=(s_0,t_0)} F_c^*(s, t) + F_{\mathcal{G}, \lambda_1, \lambda_2}^*(s_1, t_1)$. This is because $\gamma_{\mathcal{G}}(s_0 - s, 0) = \gamma_{\mathcal{G}}(s_0, s)$. Now, observe that the indicator functions $F_{\mathcal{G}, \lambda_1, \lambda_2}, F_c$ are closed, convex functions because their domains are closed, convex sets. Indeed, $\mathcal{G}$ is a closed, convex set by Assumptions A.1, A.2. Also, it is simple to verify that the set $\{(f, g) \mid f(x) + g(y) \le c(x, y) \; \forall \; x, y \in \mathcal{X}\}$ is closed and convex. Hence by applying the Moreau-Rockafellar formula (Theorem B2), we have that the RHS of 18 is equal to $(F_c + F_{\mathcal{G}, \lambda_1, \lambda_2})^*(s_0, t_0)$. But from the definition of conjugate, we have that $(F_c + F_{\mathcal{G}, \lambda_1, \lambda_2})^*(s_0, t_0) \equiv \max\limits_{f \in \mathcal{C}(\mathcal{X}), g \in \mathcal{C}(\mathcal{X})} \int_{\mathcal{X}} f \, ds_0 + \int_{\mathcal{X}} g \, dt_0 - F_c(f, g) - F_{\mathcal{G}, \lambda_1, \lambda_2}(f, g)$. Finally, from the definition of the indicator functions $F_c$, $F_{\mathcal{G}, \lambda_1, \lambda_2}$, this is same as the final RHS in 15. Hence Proved. □

**Remark B.5.** *Whenever the kernel, $k$, employed is continuous, the generating set of the corresponding MMD satisfies assumptions A.1, A.2 and $\mathcal{G}_k \subseteq \mathcal{C}(\mathcal{X})$. Hence, the above proof also works in our case of MMD-UOT.*

### B.9 Proof of Theorem 4.10: Consistency of the Proposed Estimator

*Proof.* From triangle inequality,

$$|\hat{\mathcal{U}}_m(\hat{s}_m, \hat{t}_m) - \bar{\mathcal{U}}(s_0, t_0)| \le |\hat{\mathcal{U}}_m(\hat{s}_m, \hat{t}_m) - \hat{\mathcal{U}}_m(s_0, t_0)| + |\hat{\mathcal{U}}_m(s_0, t_0) - \bar{\mathcal{U}}(s_0, t_0)|, \tag{19}$$

where $\hat{\mathcal{U}}_m(s_0, t_0)$ is same as $\bar{\mathcal{U}}(s_0, t_0)$ except that it employs the restricted feasibility set, $\mathcal{F}(\hat{s}_m, \hat{t}_m)$, for the transport plan: set of all joints supported using the samples in $\hat{s}_m, \hat{t}_m$ alone i.e.,

$\mathcal{F}(\hat{s}_m, \hat{t}_m) \equiv \left\{ \sum_{i=1}^m \sum_{j=1}^m \alpha_{ij} \delta_{(x_{1i}, x_{2j})} \mid \alpha_{ij} \ge 0 \; \forall \; i, j = 1, \ldots, m \right\}$. Here, $\delta_z$ is the Dirac measure at $z$. We begin by bounding the first term in RHS of (19).

We denote the (common) objective in $\hat{\mathcal{U}}_m(\cdot,\cdot), \bar{\mathcal{U}}(\cdot,\cdot)$ as a function of the transport plan, $\pi$, by $h(\pi,\cdot,\cdot)$. Then,

$$
\begin{aligned}
\hat{\mathcal{U}}_m(\hat{s}_m, \hat{t}_m) - \hat{\mathcal{U}}_m(s_0, t_0) &= \min_{\pi \in \mathcal{F}(\hat{s}_m, \hat{t}_m)} h(\pi, \hat{s}_m, \hat{t}_m) - \min_{\pi \in \mathcal{F}(\hat{s}_m, \hat{t}_m)} h(\pi, s_0, t_0) \\
&\leq h(\pi^{0*}, \hat{s}_m, \hat{t}_m) - h(\pi^{0*}, s_0, t_0) \left( \text{where } \pi^{0*} = \underset{\pi \in \mathcal{F}(\hat{s}_m, \hat{t}_m)}{\arg\min} \; h(\pi, s_0, t_0) \right) \\
&= \lambda_1 \left( \mathrm{MMD}_k(\pi_1^{0*}, \hat{s}_m) - \mathrm{MMD}_k(\pi_1^{0*}, s_0) \right) + \lambda_2 \left( \mathrm{MMD}_k(\pi_2^{0*}, \hat{t}_m) - \mathrm{MMD}_k(\pi_2^{0*}, t_0) \right) \\
&\leq \lambda_1 \mathrm{MMD}_k(s_0, \hat{s}_m) + \lambda_2 \mathrm{MMD}_k(t_0, \hat{t}_m) \; (\because \mathrm{MMD}_k \text{ satisfies triangle inequality})
\end{aligned}
$$

Similarly, one can show that $\hat{\mathcal{U}}_m(s_0, t_0) - \hat{\mathcal{U}}_m(\hat{s}_m, \hat{t}_m) \leq \lambda_1 \mathrm{MMD}_k(s_0, \hat{s}_m) + \lambda_2 \mathrm{MMD}_k(t_0, \hat{t}_m)$. Now, (Muandet et al., 2017, Theorem 3.4) shows that, with probability at least $1 - \delta$, $\mathrm{MMD}_k(s_0, \hat{s}_m) \leq \frac{1}{\sqrt{m}} + \sqrt{\frac{2\log(1/\delta)}{m}}$, where $k$ is a normalized kernel. Hence, the first term in inequality (19) is upper-bounded by $(\lambda_1 + \lambda_2)\left( \frac{1}{\sqrt{m}} + \sqrt{\frac{2\log 2/\delta}{m}} \right)$, with probability at least $1 - \delta$.

We next look at the second term in inequality (19): $|\hat{\mathcal{U}}_m(s_0, t_0) - \bar{\mathcal{U}}(s_0, t_0)|$. Let $\bar{\pi}^m$ be the optimal transport plan in definition of $\hat{\mathcal{U}}_m(s_0, t_0)$. Let $\pi^*$ be the optimal transport plan in the definition of $\bar{\mathcal{U}}(s_0, t_0)$. Consider another transport plan: $\hat{\pi}^m \in \mathcal{F}(\hat{s}_m, \hat{t}_m)$ such that $\hat{\pi}^m(x_i, y_j) = \frac{\eta(x_i, y_j)}{m^2}$ where $\eta(x_i, y_j) = \frac{\pi^*(x_i, y_j)}{s_0(x_i) t_0(y_j)}$ for $i, j \in [1, m]$.

$$
\begin{aligned}
|\hat{\mathcal{U}}_m(s_0, t_0) - \bar{\mathcal{U}}(s_0, t_0)| &= \hat{\mathcal{U}}_m(s_0, t_0) - \bar{\mathcal{U}}(s_0, t_0) \\
&= h(\bar{\pi}^m, s_0, t_0) - h(\pi^*, s_0, t_0) \\
&\leq h(\hat{\pi}^m, s_0, t_0) - h(\pi^*, s_0, t_0) \; (\because \bar{\pi}^m \text{ is optimal,}) \\
&\leq \int c \, \mathrm{d}\hat{\pi}^m - \int c \, \mathrm{d}\pi^* + \lambda_1 \|\mu_k(\hat{\pi}_1^m) - \mu_k(\pi_1^*)\|_k + \lambda_2 \|\mu_k(\hat{\pi}_2^m) - \mu_k(\pi_2^*)\|_k \\
&\quad (\because \text{Triangle inequality})
\end{aligned}
$$

To upper bound these terms, we utilize the fact that the RKHS, $\mathcal{H}_k$, corresponding to a c-universal kernel, $k$, is dense in $\mathcal{C}(\mathcal{X})$ wrt. the supnorm (Sriperumbudur et al., 2011) and like-wise the direct-product space, $\mathcal{H}_k \otimes \mathcal{H}_k$, is dense in $\mathcal{C}(\mathcal{X} \times \mathcal{X})$ (Gretton, 2015). Given any $f \in \mathcal{C}(\mathcal{X}) \times \mathcal{C}(\mathcal{X})$, and arbitrarily small $\epsilon > 0$, we denote by $f_\epsilon, f_{-\epsilon}$ the functions in $\mathcal{H}_k \otimes \mathcal{H}_k$ that satisfy the condition:

$$
f - \epsilon/2 \leq f_{-\epsilon} \leq f \leq f_\epsilon \leq f + \epsilon/2.
$$

Such an $f_\epsilon \in \mathcal{H}_k \otimes \mathcal{H}_k$ will exist because: i) $f + \epsilon/4 \in \mathcal{C}(\mathcal{X}) \times \mathcal{C}(\mathcal{X})$ and ii) $\mathcal{H}_k \otimes \mathcal{H}_k \subseteq \mathcal{C}(\mathcal{X}) \times \mathcal{C}(\mathcal{X})$ is dense. So there must exist some $f_\epsilon \in \mathcal{H}_k \otimes \mathcal{H}_k$ such that $|f(x,y) + \epsilon/4 - f_\epsilon(x,y)| \leq \epsilon/4 \; \forall \; x, y \in \mathcal{X} \iff f(x,y) \leq f_\epsilon(x,y) \leq f(x,y) + \epsilon/2 \; \forall \; x, y \in \mathcal{X}$. Analogously, $f_{-\epsilon}$ exists. In other words, $f_\epsilon, f_{-\epsilon} \in \mathcal{H}_k \otimes \mathcal{H}_k$ are arbitrarily close upper-bound (majorant), lower-bound (minorant) of $f \in \mathcal{C}(\mathcal{X}) \times \mathcal{C}(\mathcal{X})$.

We now upper-bound the first of the set of terms (denote $s_0(x) t_0(y)$ by $\xi(x,y)$ and $\hat{\xi}^m(x,y)$ is the corresponding empirical measure):

$$
\begin{aligned}
\int c \, \mathrm{d}\hat{\pi}^m - \int c \, \mathrm{d}\pi^* &\leq \int c_\epsilon \, \mathrm{d}\hat{\pi}^m - \int c_{-\epsilon} \, \mathrm{d}\pi^* \\
&= \langle c_\epsilon, \mu_k(\hat{\pi}^m) \rangle - \langle c_{-\epsilon}, \mu_k(\pi^*) \rangle \\
&= \langle c_\epsilon, \mu_k(\hat{\pi}^m) \rangle - \langle c_\epsilon, \mu_k(\pi^*) \rangle + \langle c_\epsilon, \mu_k(\pi^*) \rangle - \langle c_{-\epsilon}, \mu_k(\pi^*) \rangle \\
&= \langle c_\epsilon, \mu_k(\hat{\pi}^m) - \mu_k(\pi^*) \rangle + \langle c_\epsilon - c_{-\epsilon}, \mu_k(\pi^*) \rangle \\
&\leq \langle c_\epsilon, \mu_k(\hat{\pi}^m) - \mu_k(\pi^*) \rangle + \epsilon \sigma_{\pi^*} \\
&\quad (\because \|c_\epsilon - c_{-\epsilon}\|_\infty \leq \epsilon \text{ and define } \sigma_s \text{ as the mass of measure } s) \\
&\leq \|c_\epsilon\|_k \|\mu_k(\hat{\pi}^m) - \mu_k(\pi^*)\|_k + \epsilon \sigma_{\pi^*}
\end{aligned}
$$

One can obtain the tightest upper bound by choosing $c_\epsilon \equiv \arg\min_{v \in \mathcal{H}_k \otimes \mathcal{H}_k} \|v\|_k$ s.t. $c \le v \le c + \epsilon/2$. Accordingly, we replace $\|c\|_k$ by $g(\epsilon)$ in the theorem statement[4]. Further, we have:

$$
\begin{aligned}
\|\mu_k(\hat{\pi}^m) - \mu_k(\pi^*)\|_k^2 &= \left\| \int \phi_k(x) \otimes \phi_k(y) \mathrm{d}\hat{\pi}^m(x,y) - \int \phi_k(x) \otimes \phi_k(y) \mathrm{d}\pi^*(x,y) \right\|_k^2 \\
&= \left\| \int \phi_k(x) \otimes \phi_k(y) \mathrm{d}\left(\hat{\pi}^m(x,y) - \pi^*(x,y)\right) \right\|_k^2 \\
&= \left\langle \int \phi_k(x) \otimes \phi_k(y) \mathrm{d}\left(\hat{\pi}^m(x,y) - \pi^*(x,y)\right), \int \phi_k(x') \otimes \phi_k(y') \mathrm{d}\left(\hat{\pi}^m(x',y') - \pi^*(x',y')\right) \right\rangle \\
&= \left\langle \int \phi_k(x) \otimes \phi_k(y)\eta(x,y)\mathrm{d}\left(\hat{\xi}^m(x,y) - \xi(x,y)\right), \int \phi_k(x') \otimes \phi_k(y')\eta(x',y')\mathrm{d}\left(\hat{\xi}^m(x',y') - \xi(x',y')\right) \right\rangle \\
&= \int \int \langle \phi_k(x) \otimes \phi_k(y), \phi_k(x') \otimes \phi_k(y') \rangle \eta(x,y)\eta(x',y')\mathrm{d}\left(\hat{\xi}^m(x,y) - \xi(x,y)\right) \ \mathrm{d}\left(\hat{\xi}^m(x',y') - \xi(x',y')\right) \\
&= \int \int \langle \phi_k(x), \phi_k(x') \rangle \langle \phi_k(y), \phi_k(y') \rangle \eta(x,y)\eta(x',y')\mathrm{d}\left(\hat{\xi}^m(x,y) - \xi(x,y)\right) \ \mathrm{d}\left(\hat{\xi}^m(x',y') - \xi(x',y')\right) \\
&= \int \int k(x,x')k(y,y')\eta(x,y)\eta(x',y')\mathrm{d}\left(\hat{\xi}^m(x,y) - \xi(x,y)\right) \ \mathrm{d}\left(\hat{\xi}^m(x',y') - \xi(x',y')\right)
\end{aligned}
$$

Now, observe that $\tilde{k} : \mathcal{X} \times \mathcal{X} \times \mathcal{X} \times \mathcal{X}$ defined by $\tilde{k}\left((x,y),(x',y')\right) \equiv k(x,x')k(y,y')\eta(x,y)\eta(x',y')$ is a valid kernel. This is because $\tilde{k} = k_a k_b k_c$, where $k_a\left((x,y),(x',y')\right) \equiv k(x,x')$ is a kernel, $k_b\left((x,y),(x',y')\right) \equiv k(y,y')$ is a kernel, and $k_c\left((x,y),(x',y')\right) \equiv \eta(x,y)\eta(x',y')$ is a kernel (the unit-rank kernel), and product of kernels is indeed a kernel. Let $\psi(x,y)$ be the feature map corresponding to $\tilde{k}$. Then, the final RHS in the above set of equations is:

$$
\begin{aligned}
&= \int \int \langle \psi(x,y), \psi(x',y') \rangle \mathrm{d}\left(\hat{\xi}^m(x,y) - \xi(x,y)\right) \ \mathrm{d}\left(\hat{\xi}^m(x',y') - \xi(x',y')\right) \\
&= \left\langle \int \psi(x,y)\mathrm{d}\left(\hat{\xi}^m(x,y) - \xi(x,y)\right), \int \psi(x',y')\mathrm{d}\left(\hat{\xi}^m(x',y') - \xi(x',y')\right) \right\rangle.
\end{aligned}
$$

Hence, we have that: $\|\mu_k(\hat{\pi}^m) - \mu_k(\pi^*)\|_k = \left\|\mu_{\tilde{k}}(\hat{\xi}^m) - \mu_{\tilde{k}}(\xi)\right\|_{\tilde{k}}$. Again, using (Muandet et al., 2017, Theorem 3.4), with probability at least $1 - \delta$, $\left\|\mu_{\tilde{k}}(\hat{\xi}^m) - \mu_{\tilde{k}}(\xi)\right\|_{\tilde{k}} \le \frac{C_{\tilde{k}}}{m} + \frac{\sqrt{2C_{\tilde{k}}\log(1/\delta)}}{m}$, where $C_{\tilde{k}} = \max_{x,y,x',y' \in \mathcal{X}} \tilde{k}\left((x,y),(x',y')\right)$. Note that $C_{\tilde{k}} < \infty$ as $\mathcal{X}$ is compact and $s_0, t_0$ are assumed to be positive measures and $k$ is normalized.

Now the MMD-regularizer terms can be bounded using a similar strategy. Recall that, $\hat{\pi}_1^m(x_i) = \sum_{j=1}^n \frac{\pi^*(x_i,y_j)}{m^2 s_0(x_i)t_0(y_j)}$, so we have the following.

$$
\begin{aligned}
\|\mu_k(\hat{\pi}_1^m) - \mu_k(\pi_1^*)\|_k^2 &= \left\| \int \phi_k(x)\mathrm{d}\hat{\pi}_1^m(x) - \int \phi_k(x)\mathrm{d}\pi_1^*(x) \right\|_k^2 \\
&= \left\| \int \phi_k(x)\mathrm{d}\left(\hat{\pi}_1^m(x) - \pi_1^*(x)\right) \right\|_k^2 \\
&= \left\langle \int \phi_k(x)\mathrm{d}\left(\hat{\pi}_1^m(x) - \pi_1^*(x)\right), \int \phi_k(x')\mathrm{d}\left(\hat{\pi}_1^m(x') - \pi_1^*(x')\right) \right\rangle \\
&= \left\langle \int \phi_k(x)\eta(x,y)\mathrm{d}\left(\hat{\xi}^m(x,y) - \xi(x,y)\right), \int \phi_k(x')\eta(x',y')\mathrm{d}\left(\hat{\xi}^m(x',y') - \xi(x',y')\right) \right\rangle \\
&= \int \int \langle \phi_k(x), \phi_k(x') \rangle \eta(x,y)\eta(x',y')\mathrm{d}\left(\hat{\xi}^m(x,y) - \xi(x,y)\right) \ \mathrm{d}\left(\hat{\xi}^m(x',y') - \xi(x',y')\right) \\
&= \int \int k(x,x')\eta(x,y)\eta(x',y')\mathrm{d}\left(\hat{\xi}^m(x,y) - \xi(x,y)\right) \ \mathrm{d}\left(\hat{\xi}^m(x',y') - \xi(x',y')\right).
\end{aligned}
$$

---

[4]This leads to a slightly weaker bound, but we prefer it for ease of presentation

Now, observe that $\bar{k} : \mathcal{X} \times \mathcal{X} \times \mathcal{X} \times \mathcal{X}$ defined by $\bar{k}\left((x,y),(x',y')\right) \equiv k(x,x')\eta(x,y)\eta(x',y')$ is a valid kernel. This is because $\bar{k} = k_1 k_2$, where $k_1\left((x,y),(x',y')\right) \equiv k(x,x')$ is a kernel and $k_2\left((x,y),(x',y')\right) \equiv \eta(x,y)\eta(x',y')$ is a kernel (the unit-rank kernel), and product of kernels is indeed a kernel. Hence, we have that: $\|\mu_k(\hat{\pi}_1^m) - \mu_k(\pi_1^*)\|_k = \left\|\mu_{\bar{k}}(\hat{\xi}^m) - \mu_{\bar{k}}(\xi)\right\|_{\bar{k}}$. Similarly, we have: $\|\mu_k(\hat{\pi}_2^m) - \mu_k(\pi_2^*)\|_k = \left\|\mu_{\bar{k}}(\hat{\xi}^m) - \mu_{\bar{k}}(\xi)\right\|_{\bar{k}}$. Again, using (Muandet et al., 2017, Theorem 3.4), with probability at least $1 - \delta$, $\left\|\mu_{\bar{k}}(\hat{\xi}^m) - \mu_{\bar{k}}(\xi)\right\|_{\bar{k}} \leq \frac{C_{\bar{k}}}{m} + \frac{\sqrt{2C_{\bar{k}}\log(1/\delta)}}{m}$, where $C_{\bar{k}} = \max_{x,y,x',y' \in \mathcal{X}} \bar{k}\left((x,y),(x',y')\right)$. Note that $C_{\bar{k}} < \infty$ as $\mathcal{X}$ is compact, $s_0, t_0$ are assumed to be positive measures, and $k$ is normalized. From the union bound, we have: $\left|\hat{\mathcal{U}}_m(\hat{s}_m, \hat{t}_m) - \bar{\mathcal{U}}(s_0, t_0)\right| \leq (\lambda_1 + \lambda_2)\left(\frac{1}{\sqrt{m}} + \sqrt{\frac{2\log(5/\delta)}{m}} + \frac{C_{\bar{k}}}{m} + \frac{\sqrt{2C_{\bar{k}}\log(5/\delta)}}{m}\right) + g(\epsilon)\left(\frac{C_{\bar{k}}}{m} + \frac{\sqrt{2C_{\bar{k}}\log(5/\delta)}}{m}\right) + \epsilon\sigma_{\pi^*}$, with probability at least $1 - \delta$. In other words, w.h.p. we have: $\left|\hat{\mathcal{U}}_m(\hat{s}_m, \hat{t}_m) - \bar{\mathcal{U}}(s_0, t_0)\right| \leq \mathcal{O}\left(\frac{\lambda_1 + \lambda_2}{\sqrt{m}} + \frac{g(\epsilon)}{m} + \epsilon\sigma_{\pi^*}\right)$ for any $\epsilon > 0$. Hence proved. $\qquad\square$

### B.9.1 Bounding $g(\epsilon)$

Let the target function to be approximated be $h^* \in \mathcal{C}(\mathcal{X}) \subset \mathcal{L}^2(\mathcal{X})$, which is the set of square-integrable functions (wrt. some measure). Since $\mathcal{X}$ is compact, $k$ being c-universal, it is also $\mathcal{L}^2$-universal.

Consider the inclusion map $\iota : \mathcal{H}_k \mapsto \mathcal{L}^2(\mathcal{X})$, defined by $\iota\, g = g$. Let's denote the adjoint of $\iota$ by $\iota^*$. Consider the regularized least square approximation of $h^*$ defined by $h_t \equiv (\iota^*\iota + t)^{-1}\iota^* h^* \in \mathcal{H}_k$, where $t > 0$. Now, using standard results, we have:

$$
\begin{aligned}
\|\iota h_t - h^*\|_{\mathcal{L}^2} &= \|\left(\iota(\iota^*\iota + t)^{-1}\iota^* - I\right)h^*\|_{\mathcal{L}^2} \\
&= \|\left(\iota\,\iota^*(\iota\,\iota^* + t)^{-1} - I\right)h^*\|_{\mathcal{L}^2} \\
&= \|\left(\iota\,\iota^*(\iota\,\iota^* + t)^{-1} - (\iota\,\iota^* + t)(\iota\,\iota^* + t)^{-1}\right)h^*\|_{\mathcal{L}^2} \\
&= t\|\left(\iota\,\iota^* + t\right)^{-1}h^*\|_{\mathcal{L}^2} \\
&\leq t\|\left(\iota\,\iota^*\right)^{-1}h^*\|_{\mathcal{L}^2}
\end{aligned}
$$

The last inequality is true because the operator $\iota\,\iota^*$ is PD and $t > 0$. Thus, if $t \equiv \hat{t} = \frac{\epsilon}{\|(\iota\,\iota^*)^{-1}h^*\|_{\mathcal{L}^2}}$, then $\|\iota h_{\hat{t}} - h^*\|_\infty \leq \|\iota h_{\hat{t}} - h^*\|_{\mathcal{L}^2} \leq \epsilon$. Clearly,

$$
\begin{aligned}
g(\epsilon) &\leq \|h_{\hat{t}}\|_{\mathcal{H}_k} \\
&= \sqrt{\langle h_{\hat{t}}, h_{\hat{t}}\rangle_{\mathcal{H}_k}} \\
&= \sqrt{\langle(\iota^*\iota + \hat{t})^{-1}\iota^* h^*, (\iota^*\iota + \hat{t})^{-1}\iota^* h^*\rangle_{\mathcal{H}_k}} \\
&= \sqrt{\langle \iota^*(\iota\,\iota^* + \hat{t})^{-1}h^*, \iota^*(\iota\,\iota^* + \hat{t})^{-1}h^*\rangle_{\mathcal{H}_k}} \\
&= \sqrt{\langle(\iota\,\iota^* + \hat{t})^{-1}\iota\,\iota^*(\iota\,\iota^* + \hat{t})^{-1}h^*, h^*\rangle_{\mathcal{L}^2}} \\
&= \sqrt{\langle(\iota\,\iota^*)^{\frac{1}{2}}(\iota\,\iota^* + \hat{t})^{-1}h^*, (\iota\,\iota^*)^{\frac{1}{2}}(\iota\,\iota^* + \hat{t})^{-1}h^*\rangle_{\mathcal{L}^2}} \\
&= \|(\iota\,\iota^*)^{\frac{1}{2}}(\iota\,\iota^* + \hat{t})^{-1}h^*\|_{\mathcal{L}^2}.
\end{aligned}
$$

Now, consider the spectral function $f(\lambda) = \frac{\lambda^{\frac{1}{2}}}{\lambda + \hat{t}}$. This is maximized when $\lambda = \hat{t}$. Hence, $f(\lambda) \leq \frac{1}{2\sqrt{\hat{t}}}$. Thus, $g(\epsilon) \leq \frac{\|h^*\|_{\mathcal{L}^2}\sqrt{\|(\iota\,\iota^*)^{-1}h^*\|_{\mathcal{L}^2}}}{2\sqrt{\epsilon}}$. Therefore, as $\epsilon$ decays as $\frac{1}{m^{2/3}}$, then, $\frac{g(\epsilon)}{m} \leq \mathcal{O}\left(\frac{1}{m^{2/3}}\right)$.

### B.10 Solving Problem (9) using Mirror Descent

Problem (9) is an instance of a convex program and can be solved using Mirror Descent (Ben-Tal & Nemirovski, 2021), presented in Algorithm 2.

---
**Algorithm 2** Mirror Descent for solving Problem (9)

---
**Require:** Initial $\alpha_1 \geq 0$, max iterations $N$.
  $f(\alpha) = \text{Tr}\left(\alpha \mathcal{C}_{12}^\top\right) + \lambda_1 \left\|\alpha \mathbf{1} - \frac{\sigma_1}{m} \mathbf{1}\right\|_{G_{11}} + \lambda_2 \left\|\alpha^\top \mathbf{1} - \frac{\sigma_2}{m} \mathbf{1}\right\|_{G_{22}}$.
  **for** $i \leftarrow 1$ to $N$ **do**
    **if** $\|\nabla f(\alpha_i)\| \neq \mathbf{0}$ **then**
      $s_i = 1/\|\nabla f(\alpha_i)\|_\infty$.
    **else**
      **return** $\alpha_i$.
    **end if**
    $\alpha_{i+1} = \alpha_i \odot e^{-s_i \nabla f(\alpha_i)}$.
  **end for**
**return** $\alpha_{i+1}$.

---

### B.11 Equivalence between Problems (9) and (10)

We comment on the equivalence between Problems (9) and (10) based on the equivalence of their Ivanov forms:

Ivanov form for Problem (9) is

$$\min_{\alpha \geq 0 \in \mathbb{R}^{m_1 \times m_2}} \text{Tr}\left(\alpha \mathcal{C}_{12}^\top\right) \text{ s.t. } \left\|\alpha \mathbf{1} - \frac{\sigma_1}{m_1} \mathbf{1}\right\|_{G_{11}} \leq r_1, \left\|\alpha^\top \mathbf{1} - \frac{\sigma_2}{m_2} \mathbf{1}\right\|_{G_{22}} \leq r_2,$$

where $r_1, r_2 > 0$.

Similarly, the Ivanov form for Problem (10) is

$$\min_{\alpha \geq 0 \in \mathbb{R}^{m_1 \times m_2}} \text{Tr}\left(\alpha \mathcal{C}_{12}^\top\right) \text{ s.t. } \left\|\alpha \mathbf{1} - \frac{\sigma_1}{m_1} \mathbf{1}\right\|_{G_{11}}^2 \leq \bar{r}_1, \left\|\alpha^\top \mathbf{1} - \frac{\sigma_2}{m_2} \mathbf{1}\right\|_{G_{22}}^2 \leq \bar{r}_2,$$

where $\bar{r}_1, \bar{r}_2 > 0$.

As we can see, the Ivanov forms are the same with $\bar{r}_1 = r_1^2, \bar{r}_2 = r_2^2$, the solutions obtained for Problems (9) and (10) are the same.

### B.12 Proof of Lemma 4.11

*Proof.* Let $f(\alpha)$ denote the objective of Problem (10), $G_{11}, G_{22}$ are the Gram matrices over the source and target samples, respectively and $m_1, m_2$ as the number of source and target samples respectively.

$$\nabla f(\alpha) = \mathcal{C}_{12} + 2\left(\lambda_1 G_{11}\left(\alpha \mathbf{1}_{m_2} - \frac{\sigma_1}{m_1} \mathbf{1}_{m_1}\right)\mathbf{1}_{m_2}^\top + \lambda_2 \mathbf{1}_{m_1}\left(\mathbf{1}_{m_1}^\top \alpha - \mathbf{1}_{m_2}^\top \frac{\sigma_2}{m_2}\right) G_{22}\right)$$

We now derive the Lipschitz constant of this gradient.

$$\nabla f(\alpha) - \nabla f(\beta) = 2\left(\lambda_1 G_{11}\left(\alpha - \beta\right)\mathbf{1}_{m_2}\mathbf{1}_{m_2}^\top + \mathbf{1}_{m_1}\mathbf{1}_{m_1}^\top \lambda_2\left(\alpha - \beta\right) G_{22}\right)$$

$$\text{vec}\left(\left(\nabla f(\alpha) - \nabla f(\beta)\right)^\top\right) = 2\left(\lambda_1 \text{vec}\left(\left(G_{11}\left(\alpha - \beta\right)\mathbf{1}_{m_2}\mathbf{1}_{m_2}^\top\right)^\top\right) + \lambda_2 \text{vec}\left(\left(\mathbf{1}_{m_1}\mathbf{1}_{m_1}^\top\left(\alpha - \beta\right) G_{22}\right)^\top\right)\right)$$

$$= 2\left(\lambda_1 \mathbf{1}_{m_2}\mathbf{1}_{m_2}^\top \otimes G_{11} + \lambda_2 G_{22} \otimes \mathbf{1}_{m_1}\mathbf{1}_{m_1}^\top\right)\text{vec}(\alpha - \beta)$$

where $\otimes$ denotes Kronecker product.

$$\|\text{vec}(\nabla f(\alpha) - \nabla f(\beta))\|_F = \|\text{vec}\left((\nabla f(\alpha) - \nabla f(\beta))^\top\right)\|_F$$

$$\leq 2\|\lambda_1 \mathbf{1}_{m_2}\mathbf{1}_{m_2}^\top \otimes G_{11} + \lambda_2 G_{22} \otimes \mathbf{1}_{m_1}\mathbf{1}_{m_1}^\top\|_F \|\text{vec}(\alpha - \beta)\|_F \text{ (Cauchy Schwarz)}.$$

This implies the Lipschitz smoothness constant

$$L = 2\|\lambda_1 \mathbf{1}_{m_2}\mathbf{1}_{m_2}^\top \otimes G_{11} + \lambda_2 G_{22} \otimes \mathbf{1}_{m_1}\mathbf{1}_{m_1}^\top\|_F$$

$$= 2\sqrt{(\lambda_1 m_2)^2\|G_{11}\|_F^2 + (\lambda_2 m_1)^2\|G_{22}\|_F^2 + 2\lambda_1\lambda_2 \left\langle \mathbf{1}_{m_2}\mathbf{1}_{m_2}^\top \otimes G_{11}, G_{22} \otimes \mathbf{1}_{m_1}\mathbf{1}_{m_1}^\top \right\rangle_F}$$

$$= 2\sqrt{(\lambda_1 m_2)^2\|G_{11}\|_F^2 + (\lambda_2 m_1)^2\|G_{22}\|_F^2 + 2\lambda_1\lambda_2(\mathbf{1}_{m_1}^\top G_{11}\mathbf{1}_{m_1})\ (\mathbf{1}_{m_2}^\top G_{22}\mathbf{1}_{m_2})}.$$

For the last equality, we use the following properties for Kronecker products-
Mixed product property: $(A \otimes B)^\top = A^\top \otimes B^\top$, $(A \otimes B)(C \otimes D) = (AC) \otimes (BD)$ and
Spectrum property: $\text{Tr}\left((AC) \otimes (BD)\right) = \text{Tr}(AC)\text{Tr}(BD)$. □

### B.13  Solving Problem (10) using Accelerated Projected Gradient Descent

In Algorithm 1, we present the accelerated projected gradient descent (APGD) algorithm that we use to solve Problem (10), as discussed in Section 4.2. The projection operation involved is $\text{Project}_{\geq 0}(\mathbf{x}) = \max(\mathbf{x}, 0)$.

### B.14  More on the Barycenter problem

#### B.14.1  Proof of Lemma 4.12

*Proof.* Recall that we estimate the barycenter with the restriction that the transport plan $\pi^i$ corresponding to $\hat{\mathcal{U}}(\hat{s}_i, s)$ is supported on $\mathcal{D}_i \times \cup_{i=1}^n \mathcal{D}_i$. Let $\beta \geq 0 \in \mathbb{R}^m$ denote the probabilities parameterizing the barycenter, $s$. With $\hat{\mathcal{U}}_m$ as defined in Equation (9), the MMD-UOT barycenter formulation, $\hat{\mathcal{B}}_m(\hat{s}_1, \cdots, \hat{s}_n) = \min_{\beta \geq 0} \sum_{i=1}^n \rho_i \hat{\mathcal{U}}_m(\hat{s}_i, s(\beta))$, becomes

$$\min_{\alpha_1, \cdots, \alpha_n, \beta \geq 0} \sum_{i=1}^n \rho_i \left\{ \text{Tr}\left(\alpha_i \mathcal{C}_i^\top\right) + \lambda_1\|\alpha_i\mathbf{1} - \frac{\sigma_i}{m_i}\mathbf{1}\|_{G_{ii}} + \lambda_2\|\alpha_i^\top\mathbf{1} - \beta\|_G \right\}. \tag{20}$$

Following our discussion in Sections 4.2 and B.11, we present an equivalent barycenter formulation with squared-MMD regularization. This not only makes the objective smooth, allowing us to exploit accelerated solvers, but also simplifies the problem, as we discuss next.

$$\mathcal{B}'_m(\hat{s}_1, \cdots, \hat{s}_n) \equiv \min_{\alpha_1, \cdots, \alpha_n, \beta \geq 0} \sum_{i=1}^n \rho_i \left\{ \text{Tr}\left(\alpha_i \mathcal{C}_i^\top\right) + \lambda_1\|\alpha_i\mathbf{1} - \frac{\sigma_i}{m_i}\mathbf{1}\|_{G_{ii}}^2 + \lambda_2\|\alpha_i^\top\mathbf{1} - \beta\|_G^2 \right\}. \tag{21}$$

The above problem is a least squares problem in terms of $\beta$ with a non-negativity constraint. Equating the gradient wrt $\beta$ as 0, we get $G(\beta - \sum_{j=1}^n \rho_j \alpha_j^\top \mathbf{1}) = 0$. As the Gram matrices of universal kernels are full-rank (Song, 2008, Corollary 32), this implies $\beta = \sum_{j=1}^n \rho_j \alpha_j^\top \mathbf{1}$, which also satisfies the non-negativity constraint. Substituting $\beta = \sum_{j=1}^n \rho_j \alpha_j^\top \mathbf{1}$ in 21 gives us the MMD-UOT barycenter formulation:

$$\mathcal{B}'_m(\hat{s}_1, \cdots, \hat{s}_n) \equiv \min_{\alpha_1, \cdots, \alpha_n, \beta \geq 0} \sum_{i=1}^n \rho_i \left\{ \text{Tr}\left(\alpha_i \mathcal{C}_i^\top\right) + \lambda_1\|\alpha_i\mathbf{1} - \frac{\sigma_i}{m_i}\mathbf{1}\|_{G_{ii}}^2 + \lambda_2\|\alpha_i^\top\mathbf{1} - \sum_{j=1}^n \rho_j \alpha_j^\top \mathbf{1}\|_G^2 \right\}. \tag{22}$$

□

### B.14.2 Solving the Barycenter Formulation

The objective of 22, as a function of $\alpha_i$, has the following smoothness constant (derivation analogous to Lemma 4.11 in the main paper).

$$L_i = 2\rho_i\sqrt{(\lambda_1 m)^2 \|G_{ii}\|_F^2 + (\eta_i m_i)^2 \|G\|_F^2 + 2\lambda_1 \eta_i (\mathbf{1}_{m_i}^\top G_{ii} \mathbf{1}_{m_i})(\mathbf{1}_m^\top G \mathbf{1}_m)}$$

where $\eta_i = \lambda_2(1-\rho_i)$. We jointly optimize for $\alpha_i$'s using accelerated projected gradient descent with step-size $1/L_i$.

### B.14.3 Consistency of the Barycenter estimator

Similar to Theorem 4.10, we show the consistency of the proposed sample-based barycenter estimator. Let $\hat{s}_i$ be the empirical measure supported over $m$ samples from $s_i$. From the proof of Lemma 4.12 and 22, recall that,

$$\mathcal{B}'_m(s_1, \cdots, s_n) = \min_{\alpha_1, \cdots, \alpha_n \geq 0} \sum_{i=1}^{n} \rho_i\Big(\text{Tr}\left(\alpha_i \mathcal{C}_i^\top\right) + \lambda_1 \|\alpha_i \mathbf{1} - \hat{s}_i\|_{G_{ii}}^2 + \lambda_2 \|\alpha_i^\top \mathbf{1} - \sum_{j=1}^{n} \rho_j \alpha_j^\top \mathbf{1}\|_G^2\Big).$$

Now let us denote the true Barycenter with squared-MMD regularization by $\mathcal{B}(s_1, \cdots, s_n) \equiv \min_{s \in \mathcal{R}^+(\mathcal{X})} \sum_{i=1}^{n} \rho_i \mathcal{U}(s_i, s)$ where $\mathcal{U}(s_i, s) \equiv \min_{\pi^i \in \mathcal{R}^+(\mathcal{X})} \int c \ \mathrm{d}\pi^i + \lambda_1 \text{MMD}_k^2(\pi_1^i, s_i) + \lambda_2 \text{MMD}_k^2(\pi_2^i, s)$. Let $\pi^{1*}, \ldots, \pi^{n*}, s^*$ be the optimal solutions corresponding to $\mathcal{B}(s_1, \cdots, s_n)$. It is easy to see that $s^* = \sum_{j=1}^{n} \rho_j \pi_2^{j*}$ (for e.g. refer (Cohen et al., 2020, Sec C)). After eliminating $s$, we have: $\mathcal{B}(s_1, \cdots, s_n) = \min_{\pi^1, \ldots, \pi^n \in \mathcal{R}^+(\mathcal{X})} \sum_{i=1}^{n} \rho_i \left(\int c \ \mathrm{d}\pi^i + \lambda_1 \text{MMD}_k^2(\pi_1^i, s_i) + \lambda_2 \text{MMD}_k^2(\pi_2^i, \sum_{j=1}^{n} \rho_j \pi_2^j)\right)$.

**Theorem B3.** *Let* $\eta^i(x, z) \equiv \frac{\pi^{i*}(x,z)}{s_i(x)s'(z)}$ *where $s'$ is the mixture density* $s' \equiv \sum_{i=1}^{n} \frac{1}{n} s_i$. *Under mild assumptions that the functions,* $\eta^i, c \in \mathcal{H}_k \otimes \mathcal{H}_k$, *we have that w.h.p., the estimation error,* $|\mathcal{B}'_m(\hat{s}_1, \cdots, \hat{s}_m) - \mathcal{B}(s_1, \cdots, s_n)| \leq \mathcal{O}(\max_{i \in [1,n]} \left(\|\eta^i\|_k \|c\|_k\right)/m)$.

*Proof.* From triangle inequality,

$$|\mathcal{B}'_m(\hat{s}_1, \cdots, \hat{s}_n) - \mathcal{B}(s_1, \cdots, s_n)| \leq |\mathcal{B}'_m(\hat{s}_1, \cdots, \hat{s}_n) - \mathcal{B}'_m(s_1, \cdots, s_n)| + |\mathcal{B}'_m(s_1, \cdots, s_n) - \mathcal{B}(s_1, \cdots, s_n)|, \quad (23)$$

where $\mathcal{B}'_m(s_1, \cdots, s_n)$ is the same as $\mathcal{B}(s_1, \cdots, s_n)$ except that it employs restricted feasibility sets, $\mathcal{F}_i(\hat{s}_1, \cdots, \hat{s}_n)$ for corresponding $\alpha_i$ as the set of all joints supported at the samples in $\hat{s}_1, \cdots, \hat{s}_n$ alone. Let $\mathcal{D}_i = \{x_{i1}, \cdots, x_{im}\}$ and the union of all samples, $\cup \mathcal{D}_{i=1}^n = \{z_1, \cdots, z_{mn}\}$. $\mathcal{F}_i(\hat{s}_1, \cdots, \hat{s}_n) \equiv \left\{\sum_{l=1}^{m} \sum_{j=1}^{mn} \alpha_{lj} \delta_{(x_{il}, z_j)} \mid \alpha_{lj} \geq 0 \ \forall \ l = 1, \ldots, m; j = 1, \ldots, mn\right\}$. Here, $\delta_r$ is the Dirac measure at $r$. We begin by bounding the first term.

We denote the (common) objective in $\mathcal{B}'_m(\cdot)$, $\mathcal{B}(\cdot)$ as a function of the transport plans, $(\pi^1, \cdots, \pi^n)$, by $h(\pi^1, \cdots, \pi^n, \cdot)$.

$$
\begin{aligned}
\mathcal{B}'_m(\hat{s}_1, \cdots, \hat{s}_n) - \mathcal{B}'_m(s_1, \cdots, s_n) &= \min_{\pi^i \in \mathcal{F}_i(\hat{s}_1, \cdots, \hat{s}_n)} h(\pi^1, \cdots, \pi^n, \hat{s}_1, \cdots, \hat{s}_n) - \min_{\pi^i \in \mathcal{F}_i(\hat{s}_1, \cdots, \hat{s}_n)} h(\pi^1, \cdots, \pi^n, s_1, \cdots, s_n) \\
&\leq h(\bar{\pi}^{1*}, \cdots, \bar{\pi}^{n*}, \hat{s}_1, \cdots, \hat{s}_n) - h(\bar{\pi}^{1*}, \cdots, \bar{\pi}^{n*}, s_1, \cdots, s_n) \\
&\quad \Big(\text{where } \bar{\pi}^{i*} = \arg\min_{\pi^i \in \mathcal{F}_i(\hat{s}_1, \cdots, \hat{s}_n)} h(\pi^1, \cdots, \pi^n, s_1, \cdots, s_n) \text{ for } i \in [1, n]\Big) \\
&= \sum_{i=1}^{n} \lambda_1 \rho_i \left(\text{MMD}_k^2(\bar{\pi}_1^{i*}, \hat{s}_i) - \text{MMD}_k^2(\bar{\pi}_1^{i*}, s_i)\right) \\
&= \sum_{i=1}^{n} \rho_i \lambda_1 \left(\text{MMD}_k(\bar{\pi}_1^{i*}, \hat{s}_i) - \text{MMD}_k(\bar{\pi}_1^{i*}, s_i)\right)\left(\text{MMD}_k(\bar{\pi}_1^{i*}, \hat{s}_i) + \text{MMD}_k(\bar{\pi}^{i*}, s_i)\right) \\
&\overset{(1)}{\leq} 2\lambda_1 M \sum_{i=1}^{n} \rho_i \left(\text{MMD}_k(\bar{\pi}_1^{i*}, \hat{s}_i) - \text{MMD}_k(\bar{\pi}_1^{i*}, s_i)\right) \\
&\leq 2\lambda_1 M \sum_{i=1}^{n} \rho_i \text{MMD}_k(\hat{s}_i, s_i) \text{ (As MMD satisfies Triangle Inequality)}, \\
&\leq 2\lambda_1 M \max_{i \in [1,n]} \text{MMD}_k(\hat{s}_i, s_i)
\end{aligned}
$$

where for inequality (1) we use that $\max\limits_{s,t\in\mathcal{R}_1^+(\mathcal{X})} \mathrm{MMD}_k(s,t) = M < \infty$ as the generating set of MMD is compact.

As with probability at least $1-\delta$, $\mathrm{MMD}_k(\hat{s}_i, s_i) \leq \frac{1}{\sqrt{m}} + \sqrt{\frac{2\log(1/\delta)}{m}}$ (Smola et al., 2007), with union bound, we get that the first term in inequality (23) is upper-bounded by $2\lambda_1 M\left(\frac{1}{\sqrt{m}} + \sqrt{\frac{2\log 2n/\delta}{m}}\right)$, with probability at least $1-\delta$.

We next look at the second term in inequality (23): $|\mathcal{B}'_m(s_1, \cdots, s_n) - \mathcal{B}(s_1, \cdots, s_n)|$. Let $(\bar{\pi}^1, \cdots, \bar{\pi}^n)$ be the solutions of $\mathcal{B}'_m(s_1, \cdots, s_n)$. Let $(\pi^{1*}, \cdots, \pi^{n*})$ be the solutions of $\mathcal{B}(s_1, \cdots, s_n)$. Recall that $s'$ denotes the mixture density $s' \equiv \sum_{i=1}^n \frac{1}{n} s_i$. Let us denote the empirical distribution of $s'$ by $\hat{s}'$ (i.e., uniform samples from $\cup_{i=1}^n \mathcal{D}_i$). Consider the transport plans: $\hat{\pi}^{im} \in \mathcal{F}_i(\hat{s}_1, \cdots, \hat{s}_n)$ such that $\hat{\pi}^{im}(l,j) = \frac{\eta^i(x_l, z_j)}{m^2 n}$ where

$$\eta^i(x_l, z_j) = \frac{\pi^{i*}(x_l, z_j)}{s_i(x_l)s'(z_j)}, \text{ for } l \in [1, m]; j \in [1, mn].$$

$$
\begin{aligned}
|\mathcal{B}'_m(s_1, \cdots, s_n) - \mathcal{B}(s_1, \cdots, s_n)| &= \mathcal{B}'_m(s_1, \cdots, s_n) - \mathcal{B}(s_1, \cdots, s_n) \\
&= h(\bar{\pi}^{1m}, \cdots, \bar{\pi}^{nm}, s_1, \cdots, s_n) - h(\pi^{1*}, \cdots, \pi^{n*}, s_1, \cdots, s_n) \\
&\leq h(\hat{\pi}^{1m}, \cdots, \hat{\pi}^{nm}, s_1, \cdots, s_n) - h(\pi^{1*}, \cdots, \pi^{n*}, s_1, \cdots, s_n) \\
&= \sum_{i=1}^{n} \rho_i \Bigg\{ \langle \mu_k(\hat{\pi}^{im}) - \mu_k(\pi^{i*}), c_i \rangle + 2\lambda_1 M \Big( \|\mu_k(\hat{\pi}_1^{im}) - \mu_k(s_i)\|_k \\
&\qquad\qquad\qquad\qquad\qquad - \|\mu_k(\pi_1^{i*}) - \mu_k(s_i)\|_k \Big) + \\
&\quad 2\lambda_2 M \left( \left\| \mu_k(\hat{\pi}_2^{im}) - \mu_k \left( \sum_{j=1}^{n} \rho_j \hat{\pi}_2^{jm} \right) \right\|_k - \left\| \mu_k(\pi_2^{i*}) - \mu_k \left( \sum_{j=1}^{n} \rho_j \pi_2^{j*} \right) \right\|_k \right) \Bigg\}
\end{aligned}
$$

(Upper-bounding the sum of two MMD terms by $2M$)

$$
\begin{aligned}
&\leq \sum_{i=1}^{n} \rho_i \Bigg\{ \langle \mu_k(\hat{\pi}^{im}) - \mu_k(\pi^{i*}), c_i \rangle + 2\lambda_1 M \left\| \mu_k(\hat{\pi}_1^{im}) - \mu_k(\pi_1^{i*}) \right\|_k + \\
&\quad 2\lambda_2 M \left( \left\| \mu_k(\hat{\pi}_2^{im}) - \mu_k \left( \sum_{j=1}^{n} \rho_j \hat{\pi}_2^{jm} \right) \right\|_k - \left\| \mu_k(\pi_2^{i*}) - \mu_k \left( \sum_{j=1}^{n} \rho_j \pi_2^{j*} \right) \right\|_k \right) \Bigg\}
\end{aligned}
$$

(Using triangle inequality)

$$
\begin{aligned}
&\leq \sum_{i=1}^{n} \rho_i \Bigg\{ \langle \mu_k(\hat{\pi}^{im}) - \mu_k(\pi^{i*}), c_i \rangle + 2\lambda_1 M \|\mu_k(\hat{\pi}_1^{im}) - \mu_k(\pi_1^{i*})\|_k + \\
&\quad 2\lambda_2 M \left( \|\mu_k(\hat{\pi}_2^{im}) - \mu_k(\pi_2^{i*})\|_k + \sum_{j=1}^{n} \rho_j \|\mu_k(\hat{\pi}_2^{jm}) - \mu_k(\pi_2^{j*})\|_k \right) \Bigg\}
\end{aligned}
$$

(Triangle Inequality and linearity of the kernel mean embedding)

$$
\begin{aligned}
&\leq \sum_{i=1}^{n} \rho_i \Bigg\{ \|\mu_k(\hat{\pi}^{im}) - \mu_k(\pi^{i*})\|_k \, \|c_i\|_k + 2\lambda_1 M \|\mu_k(\hat{\pi}_1^{im}) - \mu_k(\pi_1^{i*})\|_k + \\
&\quad 2\lambda_2 M \left( \|\mu_k(\hat{\pi}_2^{im}) - \mu_k(\pi_2^{i*})\|_k + \sum_{j=1}^{n} \rho_j \|\mu_k(\hat{\pi}_2^{jm}) - \mu_k(\pi_2^{j*})\|_k \right) \Bigg\}
\end{aligned}
$$

(Cauchy Schwarz)

$$
\begin{aligned}
&\leq \max_{i \in [1,n]} \Bigg\{ \|\mu_k(\hat{\pi}^{im}) - \mu_k(\pi^{i*})\|_k \, \|c_i\|_k + 2\lambda_1 M \|\mu_k(\hat{\pi}_1^{im}) - \mu_k(\pi_1^{i*})\|_k + \\
&\quad 2\lambda_2 M \left( \|\mu_k(\hat{\pi}_2^{im}) - \mu_k(\pi_2^{i*})\|_k + \max_{j \in [1,n]} \|\mu_k(\hat{\pi}_2^{jm}) - \mu_k(\pi_2^{j*})\|_k \right) \Bigg\}
\end{aligned}
$$

We now repeat the steps similar to B.9 (for bounding the second term in the proof of Theorem 4.10) and get the following.

$$
\begin{aligned}
\|\mu_k(\hat{\pi}^{im}) - \mu_k(\pi^{i*})\|_k &= \max_{f \in \mathcal{H}_k, \|f\|_k \leq 1} \left| \int f \, d\hat{\pi}^{im} - \int f \, d\pi^{i*} \right| \\
&= \max_{f \in \mathcal{H}_k, \|f\|_k \leq 1} \int f \, d\hat{\pi}^{im} - \int f \, d\pi^{i*} \\
&= \max_{f \in \mathcal{H}_k, \|f\|_k \leq 1} \sum_{l=1}^{m} \sum_{j=1}^{mn} f(x_l, z_j) \frac{\pi^{i*}(x_l, z_j)}{m^2 n s_i(x_l) s'(z_j)} - \int \int f(x, z) \frac{\pi^{i*}(x, z)}{s_i(x) s'(z)} s_i(x) s'(z) \, dx \, dz \\
&= \max_{f \in \mathcal{H}_k, \|f\|_k \leq 1} \mathbb{E}_{X \sim \hat{s}_i - s_i, Z \sim \hat{s}' - s'} \left[ f(X, Z) \frac{\pi^{i*}(X, Z)}{s_i(X) s'(Z)} \right] \\
&= \max_{f \in \mathcal{H}_k, \|f\|_k \leq 1} \mathbb{E}_{X \sim \hat{s}_i - s_i, Z \sim \hat{s}' - s'} \left[ f(X, Z) \eta^i(X, Z) \right] \\
&= \max_{f \in \mathcal{H}_k, \|f\|_k \leq 1} \mathbb{E}_{X \sim \hat{s}_i - s_i, Z \sim \hat{s}' - s'} \left[ \langle f \otimes \eta^i, \phi(X) \otimes \phi(Z) \otimes \phi(X) \otimes \phi(Z) \rangle \right] \\
&= \max_{f \in \mathcal{H}_k, \|f\|_k \leq 1} \langle f \otimes \eta^i, \mathbb{E}_{X \sim \hat{s}_i - s_i, Z \sim \hat{s}' - s'} \left[ \phi(X) \otimes \phi(Z) \otimes \phi(X) \otimes \phi(Z) \right] \rangle \\
&\leq \max_{f \in \mathcal{H}_k, \|f\|_k \leq 1} \|f \otimes \eta^i\|_k \| \mathbb{E}_{X \sim \hat{s}_i - s_i, Z \sim \hat{s}' - s'} \left[ \phi(X) \otimes \phi(Z) \otimes \phi(X) \otimes \phi(Z) \right] \|_k \\
&\quad (\because \text{ Cauchy Schwarz}) \\
&= \max_{f \in \mathcal{H}_k, \|f\|_k \leq 1} \|f\|_k \|\eta^i\|_k \| \mathbb{E}_{X \sim \hat{s}_i - s_i, Z \sim \hat{s}' - s'} \left[ \phi(X) \otimes \phi(X) \otimes \phi(Z) \otimes \phi(Z) \right] \|_k \\
&\quad (\because \text{ properties of norm of tensor product}) \\
&= \max_{f \in \mathcal{H}_k, \|f\|_k \leq 1} \|f\|_k \|\eta^i\|_k \| \mathbb{E}_{X \sim \hat{s}_i - s_i} \left[ \phi(X) \otimes \phi(X) \right] \otimes \mathbb{E}_{Z \sim \hat{s}' - s'} \left[ \phi(Z) \otimes \phi(Z) \right] \|_k \\
&\leq \|\eta^i\|_k \| \mathbb{E}_{X \sim \hat{s}_i - s_i} \left[ \phi(X) \otimes \phi(X) \right] \|_k \| \mathbb{E}_{Z \sim \hat{s}' - s'} \left[ \phi(Z) \otimes \phi(Z) \right] \|_k \\
&= \|\eta^i\|_k \|\mu_{k^2}(\hat{s}_i) - \mu_{k^2}(s_i)\|_{k^2} \|\mu_{k^2}(\hat{s}') - \mu_{k^2}(s')\|_{k^2} \\
&\quad (\because \ \phi(\cdot) \otimes \phi(\cdot) \text{ is the feature map corresponding to } k^2.)
\end{aligned}
$$

Similarly, we have the following for the marginals.

$$
\begin{aligned}
\|\mu_k(\hat{\pi}_1^{im}) - \mu_k(\pi_1^{i*})\|_k &= \max_{f \in \mathcal{H}_k, \|f\|_k \leq 1} \left| \int f \, d\hat{\pi}_1^{im} - \int f \, d\pi_1^{i*} \right| \\
&= \max_{f \in \mathcal{H}_k, \|f\|_k \leq 1} \int f \, d\hat{\pi}_1^{im} - \int f \, d\pi_1^{i*} \\
&= \max_{f \in \mathcal{H}_k, \|f\|_k \leq 1} \sum_{l=1}^{m} \sum_{j=1}^{mn} f(x_l) \frac{\pi^{i*}(x_l, z_j)}{m^2 n s_i(x_l) s'(z_j)} - \int \int f(x) \frac{\pi^{i*}(x, z)}{s_i(x) s'(z)} s_i(x) s'(z) \, dx \, dz \\
&= \max_{f \in \mathcal{H}_k, \|f\|_k \leq 1} \mathbb{E}_{X \sim \hat{s}_i - s_i, Z \sim \hat{s}' - s'} \left[ f(X) \frac{\pi^{i*}(X, Z)}{s_i(X) s'(Z)} \right] \\
&= \max_{f \in \mathcal{H}_k, \|f\|_k \leq 1} \mathbb{E}_{X \sim \hat{s}_i - s_i, Z \sim \hat{s}' - s'} \left[ f(X) \eta^i(X, Z) \right] \\
&= \max_{f \in \mathcal{H}_k, \|f\|_k \leq 1} \mathbb{E}_{X \sim \hat{s}_i - s_i, Z \sim \hat{s}' - s'} \left[ \langle f \otimes \eta^i, \phi(X) \otimes \phi(X) \otimes \phi(Z) \rangle \right] \\
&= \max_{f \in \mathcal{H}_k, \|f\|_k \leq 1} \langle f \otimes \eta^i, \mathbb{E}_{X \sim \hat{s}_i - s_i, Z \sim \hat{s}' - s'} \left[ \phi(X) \otimes \phi(X) \otimes \phi(Z) \right] \rangle \\
&\leq \max_{f \in \mathcal{H}_k, \|f\|_k \leq 1} \|f \otimes \eta^i\|_k \| \mathbb{E}_{X \sim \hat{s}_i - s_i, Z \sim \hat{s}' - s'} \left[ \phi(X) \otimes \phi(X) \otimes \phi(Z) \right] \|_k \\
&\quad (\because \text{ Cauchy Schwarz}) \\
&= \max_{f \in \mathcal{H}_k, \|f\|_k \leq 1} \|f\|_k \|\eta^i\|_k \| \mathbb{E}_{X \sim \hat{s}_i - s_i, Z \sim \hat{s}' - s'} \left[ \phi(X) \otimes \phi(X) \otimes \phi(Z) \right] \|_k \\
&\quad (\because \text{ properties of norm of tensor product}) \\
&= \max_{f \in \mathcal{H}_k, \|f\|_k \leq 1} \|f\|_k \|\eta^i\|_k \| \mathbb{E}_{X \sim \hat{s}_i - s_i} \left[ \phi(X) \otimes \phi(X) \right] \otimes \mathbb{E}_{Z \sim \hat{s}' - s'} \left[ \phi(Z) \right] \|_k \\
&\leq \|\eta^i\|_k \| \mathbb{E}_{X \sim \hat{s}_i - s_i} \left[ \phi(X) \otimes \phi(X) \right] \|_k \| \mathbb{E}_{Z \sim \hat{s}' - s'} \left[ \phi(Z) \right] \|_k \\
&= \|\eta^i\|_k \|\mu_{k^2}(\hat{s}_i) - \mu_{k^2}(s_i)\|_{k^2} \|\mu_k(\hat{s}') - \mu_k(s')\|_k \\
&\quad (\because \ \phi(\cdot) \otimes \phi(\cdot) \text{ is the feature map corresponding to } k^2.)
\end{aligned}
$$

Thus, with probability at least $1 - \delta$, $|\mathcal{B}'_m(s_1, \cdots, s_n) - \mathcal{B}(s_1, \cdots, s_n)| \leq$ $\left( \max_{i \in [1,n]} \left\{ \|\eta^i\|_k \|c_i\|_k + 2\lambda_1 M \|\eta^i\|_k + 2\lambda_2 M (\|\eta^i\|_k + \max_{j \in [1,n]} \|\eta^j\|_k) \right\} \right) \left( \frac{1}{\sqrt{m}} + \sqrt{\frac{2 \log{(2n+2)/\delta}}{m}} \right)^2$. Applying union bound again for the inequality in 23, we get that with probability at least $1 - \delta$, $|\mathcal{B}'_m(\hat{s}_1, \cdots, \hat{s}_n) - \mathcal{B}(s_1, \cdots, s_n)| \leq \left( \frac{1}{\sqrt{m}} + \sqrt{\frac{2 \log{(2n+4)/\delta}}{m}} \right) \left( 2\lambda_1 M + \zeta \left( \frac{1}{\sqrt{m}} + \sqrt{\frac{2 \log{(2n+4)/\delta}}{m}} \right) \right)$, where

$$\zeta = \left( \max_{i \in [1,n]} \left\{ \|\eta^i\|_k \|c_i\|_k + 2\lambda_1 M \|\eta^i\|_k + 2\lambda_2 M (\|\eta^i\|_k + \max_{j \in [1,n]} \|\eta^j\|_k) \right\} \right). \qquad \square$$

### B.15 More on Formulation (10)

Analogous to Formulation (10) in the main paper, we consider the following formulation where an IPM raised to the $q^{th}$ power with $q > 1 \in \mathbb{Z}$ is used for regularization.

$$U_{\mathcal{G}, c, \lambda_1, \lambda_2, q}(s_0, t_0) \equiv \min_{\pi \in \mathcal{R}^+(\mathcal{X} \times \mathcal{X})} \int c \, d\pi + \lambda_1 \gamma_{\mathcal{G}}^q(\pi_1, s_0) + \lambda_2 \gamma_{\mathcal{G}}^q(\pi_2, t_0) \qquad (24)$$

Formulation (10) in the main paper is a special case of Formulation (24), when IPM is MMD and $q = 2$.

Following the proof in Lemma B1, one can easily show that

$$U_{\mathcal{G}, c, \lambda_1, \lambda_2, q}(s_0, t_0) \equiv \min_{s, t \in \mathcal{R}^+(\mathcal{X})} |s| W_1(s, t) + \lambda_1 \gamma_{\mathcal{G}}^q(s, s_0) + \lambda_2 \gamma_{\mathcal{G}}^q(t, t_0). \qquad (25)$$

To simplify notations, we denote $U_{\mathcal{G}, c, \lambda, \lambda, 2}$ by $U$ in the following. It is easy to see that $U$ satisfies the following properties by inheritance.

1. $U \geq 0$ as each of the terms in the objective in Formulation (25) is greater than 0.

2. $U(s_0, t_0) = 0 \iff s_0 = t_0$, whenever the IPM used for regularization is a norm-induced metric. As $W_1, \gamma_{\mathcal{G}}$ are non-negative terms, $U(s_0, t_0) = 0 \iff s = t, \gamma_{\mathcal{G}}(s, s_0) = 0, \gamma_{\mathcal{G}}(t, t_0) = 0$. If IPM used for regularization is a norm-induced metric, the above condition reduces to $s_0 = t_0$.

3. $U(s_0, t_0) = U(t_0, s_0)$ as each term in Formulation (25) is symmetric.

We now derive sample complexity with Formulation (24).

**Lemma B4.** *Let us denote $U_{\mathcal{G}, c, \lambda_1, \lambda_2, q}$ defined in Formulation (9) by $U$, where $q > 1 \in \mathbb{Z}$. Let $\hat{s}_m, \hat{t}_m$ denote the empirical estimates of $s_0, t_0 \in \mathcal{R}_1^+(\mathcal{X})$ respectively with $m$ samples. Then, $U(\hat{s}_m, \hat{t}_m) \to U(s_0, t_0)$ at a rate same as that of $\gamma_{\mathcal{G}}(\hat{s}_m, s_0) \to 0$.*

*Proof.*

$$U(s_0, t_0) \equiv \min_{\pi \in \mathcal{R}^+(\mathcal{X} \times \mathcal{X})} h(\pi, s_0, t_0) \equiv \int c \, d\pi + \lambda \gamma_{\mathcal{G}}^q(\pi_1, s_0) + \lambda \gamma_{\mathcal{G}}^q(\pi_2, t_0)$$

We have,

$$U(s_m, t_m) - U(s_0, t_0) = \min_{\pi \in \mathcal{R}^+(\mathcal{X} \times \mathcal{X})} h(\pi, \hat{s}_m, \hat{t}_m) - \min_{\pi \in \mathcal{R}^+(\mathcal{X} \times \mathcal{X})} h(\pi, s_0, t_0)$$

$$\leq h(\pi^*, \hat{s}_m, \hat{t}_m) - h(\pi^*, s_0, t_0) \left( \text{where } \pi^* = \arg\min_{\pi \in \mathcal{R}^+(\mathcal{X} \times \mathcal{X})} h(\pi, s_0, t_0) \right)$$

$$= \lambda \left( \gamma_{\mathcal{G}}^q(\pi_1^*, \hat{s}_m) - \gamma_{\mathcal{G}}^q(\pi_1^*, s_0) + \gamma_{\mathcal{G}}^q(\pi_2^*, \hat{t}_m) - \gamma_{\mathcal{G}}^q(\pi_2^*, t_0) \right)$$

$$= \lambda \left( (\gamma_{\mathcal{G}}(\pi_1^*, \hat{s}_m) - \gamma_{\mathcal{G}}(\pi_1^*, s_0)) \left( \sum_{i=0}^{q-1} \gamma_{\mathcal{G}}^i(\pi_1^*, \hat{s}_m) \gamma_{\mathcal{G}}^{q-1-i}(\pi_1^*, s_0) \right) \right) +$$

$$\lambda \left( (\gamma_{\mathcal{G}}(\pi_2^*, \hat{t}_m) - \gamma_{\mathcal{G}}(\pi_2^*, t_0)) \left( \sum_{i=0}^{q-1} \gamma_{\mathcal{G}}^i(\pi_2^*, \hat{t}_m) \gamma_{\mathcal{G}}^{q-1-i}(\pi_2^*, t_0) \right) \right)$$

$$
\begin{aligned}
&\leq \lambda \left( \gamma_{\mathcal{G}}(s_0, \hat{s}_m) \left( \sum_{i=0}^{q-1} \gamma_{\mathcal{G}}^i(\pi_1^*, \hat{s}_m) \gamma_{\mathcal{G}}^{q-1-i}(\pi_1^*, s_0) \right) \right) + \\
&\quad \lambda \left( \gamma_{\mathcal{G}}(t_0, \hat{t}_m) \left( \sum_{i=0}^{q-1} \gamma_{\mathcal{G}}^i(\pi_2^*, \hat{t}_m) \gamma_{\mathcal{G}}^{q-1-i}(\pi_2^*, t_0) \right) \right) \quad (\because \gamma_{\mathcal{G}} \text{ satisfies triangle inequality}) \\
&\leq \lambda \left( \gamma_{\mathcal{G}}(s_0, \hat{s}_m) \sum_{i=0}^{q-1} \left( \binom{q-1}{i} \gamma_{\mathcal{G}}^i(\pi_1^*, \hat{s}_m) \gamma_{\mathcal{G}}^{q-1-i}(\pi_1^*, s_0) \right) \right) + \\
&\quad \lambda \left( (\gamma_{\mathcal{G}}(t_0, \hat{t}_m) \left( \sum_{i=0}^{q-1} \binom{q-1}{i} \gamma_{\mathcal{G}}^i(\pi_2^*, \hat{t}_m) \gamma_{\mathcal{G}}^{q-1-i}(\pi_2^*, t_0) \right) \right) \\
&= \lambda \Big( \gamma_{\mathcal{G}}(s_0, \hat{s}_m) \left( \gamma_{\mathcal{G}}(\pi_1^*, \hat{s}_m) + \gamma_{\mathcal{G}}(\pi_1^*, s_0) \right)^{q-1} \\
&\quad + \gamma_{\mathcal{G}}(t_0, \hat{t}_m) \left( \gamma_{\mathcal{G}}(\pi_2^*, \hat{t}_m) + \gamma_{\mathcal{G}}(\pi_2^*, t_0) \right)^{q-1} \Big) \\
&\leq \lambda (2M)^{q-1} \left( \gamma_{\mathcal{G}}(s_0, \hat{s}_m) + \gamma_{\mathcal{G}}(t_0, \hat{t}_m) \right).
\end{aligned}
$$

For the last inequality, we use that $\max_{a \in \mathcal{R}_1^+(\mathcal{X})} \max_{b \in \mathcal{R}_1^+(\mathcal{X})} \gamma_{\mathcal{G}}(a, b) = M < \infty$ as the domain is compact.

Similarly, one can show the other way inequality, resulting in the following.

$$
|U(s_0, t_0) - U(s_m, t_m)| \leq \lambda (2M)^{q-1} \left( \gamma_{\mathcal{G}}(s_0, \hat{s}_m) + \gamma_{\mathcal{G}}(t_0, \hat{t}_m) \right). \tag{26}
$$

The rate at which $|U(s_m, t_m) - U(s_0, t_0)|$ goes to zero is hence the same as that with which either of the IPM terms goes to zero. For example, if the IPM used for regularization is MMD with a normalized kernel, then $\text{MMD}_k(s_0, \hat{s}_m) \leq \sqrt{\frac{1}{m}} + \sqrt{\frac{2 \log(1/\delta)}{m}}$ with probability at least $1 - \delta$ (Smola et al., 2007).

From the union bound, with probability at least $1 - \delta$, $|U(s_m, t_m) - U(s_0, t_0)| \leq 2\lambda(2M)^{q-1} \left( \sqrt{\frac{1}{m}} + \sqrt{\frac{2 \log(2/\delta)}{m}} \right)$. Thus, $O\left( \frac{1}{\sqrt{m}} \right)$ is the common bound for the rate at which the LHS as well as the $\text{MMD}_k(s_0, \hat{s}_m)$ decays to zero. $\qquad \square$

### B.16 Robustness

We show the robustness property of IPM-regularized UOT 13 with the same assumptions on the noise model as used in (Fatras et al., 2021, Lemma 1) for KL-regularized UOT.

**Lemma B5.** *(Robustness) Let $s_0, t_0 \in \mathcal{R}_1^+(\mathcal{X})$. Consider $s_c = \rho s_0 + (1 - \rho)\delta_z$ ($\rho \in [0,1]$), a distribution perturbed by a Dirac outlier located at some $z$ outside of the support of $t_0$. Let $m(z) = \int c(z, y) \mathrm{d}t_0(y)$. We have that, $\mathcal{U}_{\mathcal{G},c,\lambda_1,\lambda_2}(s_c, t_0) \leq \rho \, \mathcal{U}_{\mathcal{G},c,\lambda_1,\lambda_2}(s_0, t_0) + (1 - \rho)m(z)$.*

*Proof.* Let $\pi$ be the solution of $\mathcal{U}_{\mathcal{G},c,\lambda_1,\lambda_2}(s_0,t_0)$. Consider $\tilde{\pi} = \rho\pi + (1-\rho)\delta_z \otimes t_0$. It is easy to see that $\tilde{\pi}_1 = \rho\pi_1 + (1-\rho)\delta_z$ and $\tilde{\pi}_2 = \rho\pi_2 + (1-\rho)t_0$.

$$\mathcal{U}_{\mathcal{G},c,\lambda_1,\lambda_2}(s_c,t_0) \leq \int c(x,y)\mathrm{d}\tilde{\pi}(x,y) + \lambda_1\gamma_{\mathcal{G}}(\tilde{\pi}_1,s_c) + \lambda_2\gamma_{\mathcal{G}}(\tilde{\pi}_2,t_0) \text{ (Using the definition of min)}$$

$$\leq \int c(x,y)\mathrm{d}\tilde{\pi}(x,y) + \lambda_1\left(\rho\gamma_{\mathcal{G}}(\pi_1,s_0) + (1-\rho)\gamma_{\mathcal{G}}(\delta_z,\delta_z)\right) + \lambda_2\left(\rho\gamma_{\mathcal{G}}(\pi_2,t_0) + (1-\rho)\gamma_{\mathcal{G}}(t_0,t_0)\right)$$

$$(\because \text{ IPMs are jointly convex})$$

$$= \int c(x,y)\mathrm{d}\tilde{\pi}(x,y) + \rho\left(\lambda_1\gamma_{\mathcal{G}}(\pi_1,s_0) + \lambda_2\gamma_{\mathcal{G}}(\pi_2,t_0)\right)$$

$$= \rho\int c(x,y)\mathrm{d}\pi(x,y) + \int (1-\rho)c(z,y)\mathrm{d}(\delta_z \otimes t_0)(z,y) + \rho\left(\lambda_1\gamma_{\mathcal{G}}(\pi_1,s_0) + \lambda_2\gamma_{\mathcal{G}}(\pi_2,t_0)\right)$$

$$= \rho\int c(x,y)\mathrm{d}\pi(x,y) + \int (1-\rho)c(z,y)\mathrm{d}t_0(y) + \rho\left(\lambda_1\gamma_{\mathcal{G}}(\pi_1,s_0) + \lambda_2\gamma_{\mathcal{G}}(\pi_2,t_0)\right)$$

$$= \rho\,\mathcal{U}_{\mathcal{G},c,\lambda_1,\lambda_2}(s_0,t_0) + (1-\rho)m(z).$$

We note that $m(z)$ is finite as $t_0 \in \mathcal{R}_1^+(\mathcal{X})$. $\qquad\square$

We now present robustness guarantees with a different noise model.

**Corollary B6.** *We say a measure $q \in \mathcal{R}^+(\mathcal{X})$ is corrupted with $\rho \in [0,1]$ fraction of noise when $q = (1-\rho)q_c + \rho q_n$, where $q_c$ is the clean measure and $q_n$ is the noisy measure.*
*Let $s_0, t_0 \in \mathcal{R}^+(\mathcal{X})$ be corrupted with $\rho$ fraction of noise such that $|s_c - s_n|_{TV} \leq \epsilon_1$ and $|t_c - t_n|_{TV} \leq \epsilon_2$. We have that $\mathcal{U}_{\mathcal{G},c,\lambda,\lambda}(s_0,t_0) \leq \mathcal{U}_{\mathcal{G},c,\lambda,\lambda}(s_c,t_c) + \rho\beta(\epsilon_1 + \epsilon_2)$, where $\beta = \max_{f\in\mathcal{G}(\lambda)\cap\mathcal{W}_c}\|f\|_{\infty}$.*

*Proof.* We use our duality result of $\mathcal{U}_{\mathcal{G},c,\lambda,\lambda}$, from Theorem 4.1. We first upper-bound $\mathcal{U}_{\mathcal{G},c,\lambda,\lambda}(s_n,t_n)$ which is later used in the proof.

$$\mathcal{U}_{\mathcal{G},c,\lambda,\lambda}(s_n,t_n) = \max_{f\in\mathcal{G}(\lambda)\cap\mathcal{W}_c} \int f\mathrm{d}s_n - \int f\mathrm{d}t_n$$

$$= \max_{f\in\mathcal{G}(\lambda)\cap\mathcal{W}_c} \int f\mathrm{d}(s_n-s_c) + \int f\mathrm{d}s_c - \int f\mathrm{d}(t_n-t_c) - \int f\mathrm{d}t_c$$

$$\leq \max_{f\in\mathcal{G}(\lambda)\cap\mathcal{W}_c} \int f\mathrm{d}(s_n-s_c) + \max_{f\in\mathcal{G}(\lambda)\cap\mathcal{W}_c} \int f\mathrm{d}(t_n-t_c) + \max_{f\in\mathcal{G}(\lambda)\cap\mathcal{W}_c}\left(\int f\mathrm{d}s_c - \int f\mathrm{d}t_c\right)$$

$$\leq \beta(|s_c - s_n|_{TV} + |t_c - t_n|_{TV}) + \mathcal{U}_{\mathcal{G},c,\lambda,\lambda}(s_c,t_c)$$

$$= \beta(\epsilon_1 + \epsilon_2) + \mathcal{U}_{\mathcal{G},c,\lambda,\lambda}(s_c,t_c). \tag{27}$$

We now show the robustness result as follows.

$$\mathcal{U}_{\mathcal{G},c,\lambda,\lambda}(s_0,t_0) = \max_{f\in\mathcal{G}(\lambda)\cap\mathcal{W}_c} \int f\mathrm{d}s_0 - \int f\mathrm{d}t_0$$

$$= \max_{f\in\mathcal{G}(\lambda)\cap\mathcal{W}_c} (1-\rho)\int f\mathrm{d}s_c + \rho\int f\mathrm{d}s_n - (1-\rho)\int f\mathrm{d}t_c - \rho\int f\mathrm{d}t_n$$

$$= \max_{f\in\mathcal{G}(\lambda)\cap\mathcal{W}_c} (1-\rho)\left(\int f\mathrm{d}s_c - \int f\mathrm{d}t_c\right) + \rho\left(\int f\mathrm{d}s_n - \int f\mathrm{d}t_n\right)$$

$$\leq \max_{f\in\mathcal{G}(\lambda)\cap\mathcal{W}_c} (1-\rho)\left(\int f\mathrm{d}s_c - \int f\mathrm{d}t_c\right) + \max_{f\in\mathcal{G}(\lambda)\cap\mathcal{W}_c} \rho\left(\int f\mathrm{d}s_n - \int f\mathrm{d}t_n\right)$$

$$= (1-\rho)\,\mathcal{U}_{\mathcal{G},c,\lambda,\lambda}(s_c,t_c) + \rho\,\mathcal{U}_{\mathcal{G},c,\lambda,\lambda}(s_n,t_n)$$

$$\leq (1-\rho)\,\mathcal{U}_{\mathcal{G},c,\lambda,\lambda}(s_c,t_c) + \rho\left(\mathcal{U}_{\mathcal{G},c,\lambda,\lambda}(s_c,t_c) + \beta(\epsilon_1 + \epsilon_2)\right) \text{ (Using 27)}$$

$$= \mathcal{U}_{\mathcal{G},c,\lambda,\lambda}(s_c,t_c) + \rho\beta(\epsilon_1 + \epsilon_2).$$

We note that $\beta = \max\limits_{f \in \mathcal{G}(\lambda) \cap \mathcal{W}_c} \|f\|_\infty \leq \max\limits_{f \in \mathcal{W}_c} \|f\|_\infty < \infty$. Also, as $\beta \leq \min\left(\max\limits_{f \in \mathcal{G}_k(\lambda)} \|f\|_\infty, \max\limits_{f \in \mathcal{W}_c} \|f\|_\infty\right) \leq$ $\min\left(\lambda, \max\limits_{f \in \mathcal{W}_c} \|f\|_\infty\right)$ (for a normalized kernel).

$\square$

### B.17 Connections with Spectral Normalized GAN

We comment on the applicability of MMD-UOT in generative modelling and draw connections with the Spectral Norm GAN (SN-GAN) (Miyato et al., 2018) formulation.

A popular approach in generative modelling is to define a parametric function $g_\theta : \mathcal{Z} \mapsto \mathcal{X}$ that takes a noise distribution and generates samples from $P_\theta$ distribution. We then learn $\theta$ to make $P_\theta$ closer to the real distribution, $P_r$. On formulating this problem with the dual of MMD-UOT derived in Theorem 4.1, we get

$$\min_\theta \max_{f \in \mathcal{W}_c \cap \mathcal{G}_k(\lambda)} \int f \mathrm{d}P_\theta - \int f \mathrm{d}P_r \tag{28}$$

We note that in the above optimization problem, the critic function or the discriminator $f$ should satisfy $\|f\|_c \leq 1$ and $\|f\|_k \leq \lambda$ where $\|f\|_c$ denotes the Lipschitz norm under the cost function $c$. Let the critic function be $f_W$, parametrized using a deep convolution neural network (CNN) with weights $W = \{W_1, \cdots, W_L\}$, where $L$ is the depth of the network. Let $\mathcal{F}$ be the space of all such CNN models, then Problem (28) can be approximated as follows.

$$\min_\theta \max_{f_W \in \mathcal{F}; \|f_W\|_c \leq 1, \|f_W\|_k \leq \lambda} \int f_W \mathrm{d}P_\theta - \int f_W \mathrm{d}P_r \tag{29}$$

The constraint $\|f\|_c \leq 1$ is popularly handled using a penalty on the gradient, $\|\nabla f_W\|$ (Gulrajani et al., 2017). The constraint on the RKHS norm, $\|f\|_k$, is more challenging for an arbitrary neural network. Thus, we follow the approximations proposed in (Bietti et al., 2019). (Bietti et al., 2019) use the result derived in (Bietti & Mairal, 2017) that constructs a kernel whose RKHS contains a CNN, $\bar{f}$, with the same architecture and parameters as $f$ but with activations that are smooth approximations of ReLU. With this approximation, (Bietti et al., 2019) shows tractable bounds on the RKHS norm. We consider their upper bound based on spectral normalization of the weights in $f_W$. With this, Problem (29) can be approximated with the following.

$$\min_\theta \max_{f_W \in \mathcal{F}} \int f_W \mathrm{d}P_\theta - \int f_W \mathrm{d}P_r + \rho_1 \|\nabla f_W\| + \rho_2 \sum_{i=1}^L \frac{1}{\lambda} \|W_i\|_{\mathrm{sp}}^2, \tag{30}$$

where $\|.\|_{\mathrm{sp}}$ denotes the spectral norm and $\rho_1, \rho_2 > 0$. Formulations like (30) have been successfully applied as variants of Spectral Normalized GAN (SN-GAN). This shows the utility of MMD-regularized UOT in generative modelling.

### B.18 Comparison with WAE

The OT problem in WAE (RHS in Theorem 1 in (Tolstikhin et al., 2018)) using our notation is:

$$\min_{\pi \in \mathcal{R}_1^+(\mathcal{X} \times \mathcal{Z})} \int c\left(x, G(z)\right) \, \mathrm{d}\pi(x, z), \text{ s.t. } \pi_1 = P_X, \ \pi_2 = P_Z, \tag{31}$$

where $\mathcal{X}, \mathcal{Z}$ are the input and latent spaces, $G$ is the decoder, and $P_X, P_Z$ are the probability measures corresponding to the underlying distribution generating the given training set and the latent prior (e.g., Gaussian).

(Tolstikhin et al., 2018) employs a one-sided regularization. More specifically, (Tolstikhin et al., 2018, eqn. (4)) in our notation is:

$$\min_{\pi \in \mathcal{R}_1^+(\mathcal{X} \times \mathcal{Z})} \int c\left(x, G(z)\right) \, \mathrm{d}\pi(x, z) + \lambda_2 \mathrm{MMD}_k(\pi_2, P_Z), \text{ s.t. } \pi_1 = P_X. \tag{32}$$

However, in our work, the proposed MMD-UOT formulation corresponding to (31) reads as:

$$\min_{\pi \in \mathcal{R}_1^+(\mathcal{X} \times \mathcal{Z})} \int c\left(x, G(z)\right) \, \mathrm{d}\pi(x, z) + \lambda_1 \mathrm{MMD}_k(\pi_1, P_X) + \lambda_2 \mathrm{MMD}_k(\pi_2, P_Z). \tag{33}$$

It is easy to see that the WAE formulation (32) is a special case of our MMD-UOT formulation (33). Indeed, as $\lambda_1 \to \infty$, both formulations are the same.

The theoretical advantages of MMD-UOT over WAE are that MMD-UOT induces a new family of metrics and can be efficiently estimated from samples at a rate $\mathcal{O}(\frac{1}{\sqrt{m}})$ whereas WAE is not expected to induce a metric as the symmetry is broken. Also, WAE is expected to be cursed with dimensions in terms of estimation, as a marginal is exactly matched, similar to unregularized OT.

We now present the details of estimating (33) in the context of VAEs. The transport plan $\pi$ is factorized as $\pi(x, z) \equiv \pi_1(x)\pi(z|x)$, where $\pi(z|x)$ is the encoder. For the sake of fair comparison, we choose this encoder and the decoder, $G$, to be exactly the same as that in (Tolstikhin et al., 2018). Since $\pi_1(x)$ is not modelled by WAE, we fall back to the default parametrization in our paper of distributions supported over the training points. More specifically, if $\mathcal{D} = \{x_1, \ldots, x_m\}$ is the training set (sampled from $P_X$), then our formulation reads as:

$$\min_{\pi(z|x), \alpha \in \Delta_m} \sum_{i=1}^{m} \alpha_i \int c\left(x_i, G(z)\right) \, \mathrm{d}\pi(z|x_i) + \lambda_1 \mathrm{MMD}_k^2 \left(\alpha, \frac{1}{m}\mathbf{1}\right) + \lambda_2 \mathrm{MMD}_k^2 \left(\sum_{i=1}^{m} \alpha_i \pi(z|x_i), P_Z\right), \tag{34}$$

where $G$ is the gram-matrix over the training set $\mathcal{D}$. We solve (34) using SGD, where the block over the $\alpha$ variables can employ accelerated gradient steps.

## C  Experimental Details and Additional Results

We present more experimental details and additional results in this section. We have followed standard practices to ensure reproducibility. We will open-source the codes to reproduce all our experiments upon acceptance of the paper.

### C.1  Synthetic Experiments

We present more details for the experiments in Section 5.1, along with additional experimental results.

**Transport Plan and Barycenter**  We use squared-Euclidean cost as the ground metric. We take points $[1, 2, \cdots, 50]$ and consider Gaussian distribution over them with mean, and standard deviation as (15, 5) and (35, 3), respectively. The hyperparameters for MMD-UOT are $\lambda$ as 100 and $\sigma^2$ in the RBF kernel $(k(x, y) = \exp\left(\frac{-\|x-y\|^2}{2\sigma^2}\right))$ as 1. The hyperparameters for $\epsilon$KL-UOT are $\lambda$ and $\epsilon$ as 1.

For the barycenter experiment, we take points $[1, 2, \cdots, 100]$ and consider Gaussian distribution over them with mean, and standard deviation as (20, 5) and (60, 8), respectively. The hyperparameters for MMD-UOT are $\lambda$ as 100 and $\sigma^2$ in the RBF kernel as 10. The hyperparameters for $\epsilon$KL-UOT are $\lambda$ as 100 and $\epsilon$ as $10^{-3}$.

**Visualizing the Level Sets**  For all OT variants squared-Euclidean is used as a ground metric. For the level set with MMD, RBF kernel is used with $\sigma^2$ as 3. For MMD-UOT, $\lambda$ is 1 and RBF kernel is used with $\sigma^2$ as 1. For plotting the level set contours, 20 lines are used for all methods.

**Computation Time**  The source and target measures are Uniform distributions from which we sample 5,000 points. The dimensionality of the data is 5. The experiment is done with hyper-parameters as squared-Euclidean distance, squared-MMD regularization with RBF kernel, sigma as 1 and lambda as 0.1. $\epsilon$KL-UOT's entropic regularization coefficient is 0.01, and lambda is 1. We choose entropic regularization coefficient from the set $\{1e-3, 1e-2, 1e-1\}$ and lambda from the set $\{1e-2, 1e-1, 1\}$. This hyper-parameter resulted in the fastest convergence. This experiment was done on an NVIDIA-RTX 2080 GPU.

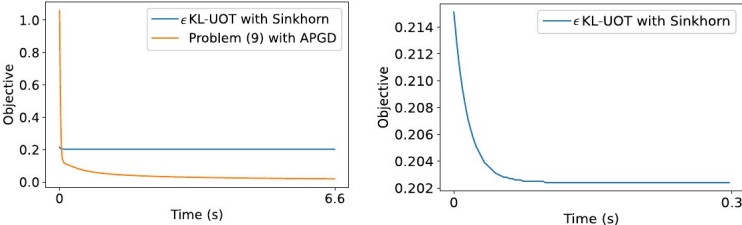

Figure 4: Computation time: Convergence plots with $m = 5000$ for the case of the same source and target measures where the optimal objective is expected to be 0. Left: MMD-UOT Problem (10) solved with accelerated projected gradient descent. Right: $\epsilon$KL-UOT's convergence plot is shown separately. We observe that $\epsilon$KL-UOT's objective plateaus in 0.3 seconds. We note that our convergence to the optimal objective is faster than that of $\epsilon$KL-UOT.

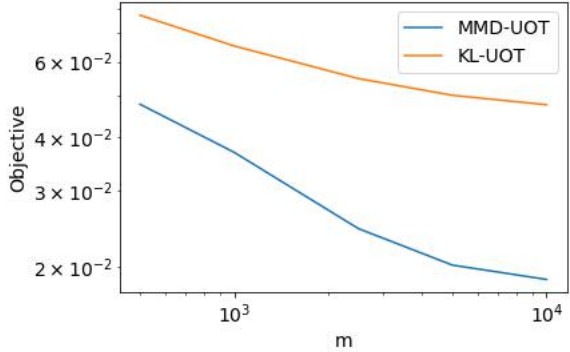

Figure 5: Sample efficiency: Log-log plot of optimal objective vs number of samples. The optimal objective values of MMD-UOT and $\epsilon$KL-UOT formulation are shown as the number of samples increases. The data lies in 10 dimensions, and the source and target measures are both Uniform. MMD-UOT can be seen to have a better rate of convergence.

**Sample Complexity**   In Theorem 4.10 in the main paper, we proved an attractive sample complexity of $\mathcal{O}\left(m^{-\frac{1}{2}}\right)$ for our sample-based estimators. In this section, we present a synthetic experiment to show that the convergence of MMD-UOT's metric towards the true value is faster than that of $\epsilon$KL-UOT. We sample 10-dimensional sources and target samples from Uniform sources and target marginals, respectively. As the marginals are equal, the metrics over measures should converge to 0 as the number of samples increases. We repeat the experiment with an increasing number of samples. We use squared-Euclidean cost. For $\epsilon$KL-UOT, $\lambda = 1, \epsilon = 1e - 2$. For MMD-UOT, $\lambda = 1$ and RBF kernel with $\sigma = 1$ is used. In Figure 5, we plot MMD-UOT's objective and the square root of the $\epsilon$KL-UOT objective on increasing the number of samples. It can be seen from the plot that the MMD-UOT achieves a better rate of convergence compared to $\epsilon$KL-UOT.

**Effect of Regularization**   In Figures 7 and 6, we visualize matching the marginals of MMD-UOT's optimal transport plan. We show the results with both RBF kernel $k(x, y) = \exp\left(\frac{-\|x-y\|^2}{2*10^{-6}}\right)$ and the IMQ kernel $k(x, y) = \left(10^{-6} + \|x - y\|^2\right)^{-0.5}$. As we increase $\lambda$, the matching becomes better for unnormalized measures, and the marginals exactly match the given measures when the measures are normalized. We have also shown the unbalanced case results with KL-UOT. As the POT library (Flamary et al., 2021) doesn't allow including a simplex constraint for KL-UOT, we do not show this.

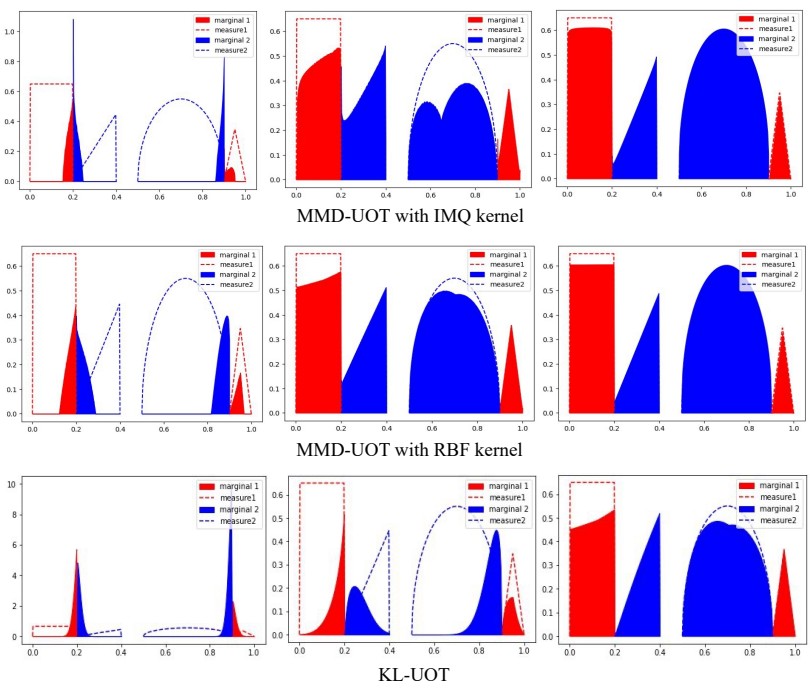

Figure 6: (With unnormalized measures) Visualizing the marginals of transport plans learnt by MMD-UOT and KL-UOT, on increasing $\lambda$.

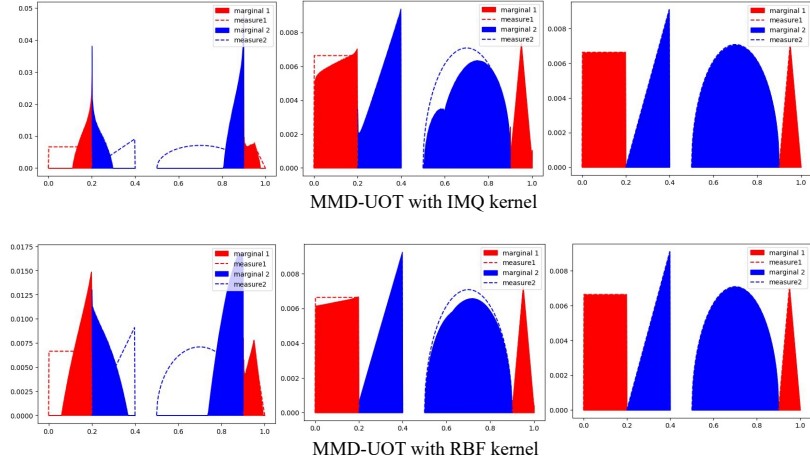

Figure 7: (With normalized measures) Visualizing the marginals of MMD-UOT (solved with simplex constraints) plan on increasing $\lambda$. We do not show KL-UOT here as the Sinkhorn algorithm for solving KL-UOT in the POT library (Flamary et al., 2021) does not incorporate the Simplex constraints on the transport plan.

## C.2   Two-sample Test

Following (Liu et al., 2020), we repeat the experiment 10 times, and in each trial, we randomly sample a validation subset and a test subset of size $N$ from the given real and fake MNIST datasets. We run the two-sample test experiment for type-II error on the test set for a given trial using the hyperparameters chosen for that trial. The hyperparameters were tuned for $N = 100$ for each trial. The hyperparameters

Table 8: Test power (higher is better) for the task of CIFAR-10.1 vs CIFAR 10. The proposed MMD-UOT method achieves the best results.

| ME | SCF | C2ST-S | C2ST-L | MMD | $\epsilon$KL-UOT | MMD-UOT |
|---|---|---|---|---|---|---|
| 0.588 | 0.171 | 0.452 | 0.529 | 0.316 | 0.132 | **0.643** |

for a given trial were chosen based on the average empirical test power (higher is better) over that trial's validation dataset.

We use squared-Euclidean distance for MMD-UOT and $\epsilon$KL-UOT formulations. RBF kernel, $k(x,y) = \exp\left(\frac{-\|x-y\|^2}{2\sigma^2}\right)$, is used for MMD and for MMD-UOT formulation. The hyperparameters are chosen from the following set. For the MMD-UOT and MMD, $\sigma$ was chosen from {median, 40, 60, 80, 100} where the median is the median-heuristic (Gretton et al., 2012). For the MMD-UOT an $\epsilon$KL-UOT, $\lambda$ is chosen from {0.1, 1, 10}. For $\epsilon$KL-UOT, $\epsilon$ was chosen from {1, $10^{-1}, 10^{-2}, 10^{-3}, 10^{-4}$}. Based on validation, $\sigma$ as the median is chosen for MMD at all trials. For $\epsilon$KL-UOT, the best hyperparameters $(\lambda, \epsilon)$ are $(10, 0.001)$ for trial number 3, $(0.1, 0.1)$ for trial number 10 and $(1, 0.1)$ for the remaining the 8 trials. For MMD-UOT, the best hyperparameters $(\lambda, \sigma^2)$ are $(0.1, 60)$ for trial number 9 and $(1, \text{median}^2)$ for the remaining 9 trials.

**Additional Results** Following (Liu et al., 2020), we consider the task of verifying that the datasets CIFAR-10 (Krizhevsky, 2009) and CIFAR-10.1 (Recht et al., 2018) are statistically different. We follow the same experimental setup as given in (Liu et al., 2020). The training is done on 1,000 images from each dataset, and the test is on 1,031 images. The experiment is repeated 10 times, and the average test power is compared with the results shown in (Liu et al., 2020) with the popular baselines: ME (Chwialkowski et al., 2015; Jitkrittum et al., 2016), SCF (Chwialkowski et al., 2015; Jitkrittum et al., 2016), C2ST-S (Lopez-Paz & Oquab, 2017), C2ST-L (Cheng & Cloninger, 2019). We repeat the experiment following the same setup for the MMD and $\epsilon$KL-UOT baselines. The chosen hyperparameters $(\lambda, \epsilon)$ for the 10 different experimental runs $\epsilon$KL-UOT are $(0.1, 0.1), (1, 0.1), (1, 0.1), (1, 0.01), (1, 0.1), (1, 0.1), (1, 0.1), (0.1, 0.1), (1, 0.1), (1, 0.1)$ and $(1, 0.1)$. The chosen $(\lambda, \sigma^2)$ for the 10 different experimental runs of MMD-UOT are $(0.1, \text{median}), (1, 60), (10, 100), (0.1, 80), (0.1, 40), (0.1, 40), (0.1, 40), (1, \text{median}), (0.1, 80)$ and $(1, 40)$. Table 8 shows that the proposed MMD-UOT obtains the highest test power.

## C.3 Single-Cell RNA sequencing

scRNA-seq helps us understand how the expression profile of the cells changes over stages (Schiebinger et al., 2019). A population of cells is represented as a measure of the gene expression space, and as they grow/divide/die, and the measure evolves over time. While scRNA-seq records such a measure at a time stamp, it does so by destroying the cells (Schiebinger et al., 2019). Thus, it is impossible to monitor how the cell population evolves continuously over time. In fact, only a few measurements at discrete timesteps are generally taken due to the cost involved.

We perform experiments on the Embryoid Body (EB) single-cell dataset (Moon et al., 2019). The Embryoid Body dataset comprises data at 5 timesteps with sample sizes as 2381, 4163, 3278, 3665 and 3332, respectively.

The MMD barycenter interpolating between measures $s_0, t_0$ has the closed form solution as $\frac{1}{2}(s_0 + t_0)$. For evaluating the performance at timestep $t_i$, we select the hyperparameters based on the task of predicting for $\{t_1, t_2, t_3\} \setminus t_i$. We use IMQ kernel $k(x,y) = \left(\frac{1+\|x-y\|^2}{K^2}\right)^{-0.5}$. The $\lambda$ hyperparameter for the validation of MMD-UOT is chosen from $\{0.1, 1, 10\}$ and $K^2$ is chosen from $\{1e-4, 1e-3, 1e-2, 1e-1, \text{median}\}$, where median denotes the median of $\{0.5\|x-y\|^2 \forall x, y \in \mathcal{D} \text{ s.t. } x \neq y\}$ over the training dataset $(\mathcal{D})$. The chosen $(\lambda, K^2)$ for timesteps $t_1, t_2, t_3$ are $(1, 0.1)$, $(1, \text{median})$ and $(1, \text{median})$, respectively. The $\lambda$ hyperparameter for the validation of $\epsilon$KL-UOT is chosen from $\{0.1, 1, 10\}$ and $\epsilon$ is chosen from $\{1e-5, 1e-4, 1e-3, 1e-2, 1e-1\}$. The chosen $(\lambda, \epsilon)$ for timesteps $t_1, t_2, t_3$ are $(10, 0.01)$, $(1, 0.1)$ and $(1, 0.1)$ respectively. In Table 9, we compare against additional OT-based methods $\bar{W}_1, \bar{W}_2, \epsilon$OT.

Table 9: Additional OT-based baselines for two-sample test: Average Test Power (between 0 and 1; higher is better) on MNIST. MMD-UOT obtains the highest average test power at all timesteps even with the additional baselines.

| N | $\bar{W}_1$ | $\bar{W}_2$ | $\epsilon$OT | MMD-UOT |
|---|---|---|---|---|
| 100 | 0.111 | 0.099 | 0.108 | **0.154** |
| 200 | 0.232 | 0.207 | 0.191 | **0.333** |
| 300 | 0.339 | 0.309 | 0.244 | **0.588** |
| 400 | 0.482 | 0.452 | 0.318 | **0.762** |
| 500 | 0.596 | 0.557 | 0.356 | **0.873** |
| 1000 | 0.805 | 0.773 | 0.508 | **0.909** |

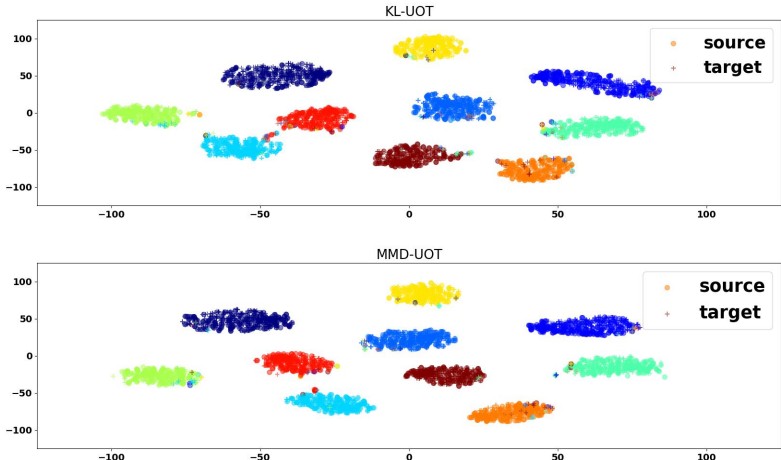

Figure 8: (Best viewed in color) The t-SNE plots of the source and target embeddings learnt for the M-MNIST to USPS domain adaptation task. Different cluster colors imply different classes. The quality of the learnt representations can be judged based on the separation between clusters. The clusters obtained by MMD-UOT seem better separated (for example, the red and the cyan-colored clusters).

## C.4   Domain Adaptation in JUMBOT framework

The experiments are performed with the same seed as used by JUMBOT. For the experiment on the Digits dataset, the chosen hyper-parameters for MMD-UOT are $K^2$ in the IMQ kernel $k(x,y) = \left( \frac{1 + \|x-y\|^2}{K^2} \right)^{-0.5}$ as $10^{-2}$ and $\lambda$ as 100. In Figure 8, we also compare the t-SNE plot of the embeddings learnt with the MMD-UOT and $\epsilon$KL-UOT-based loss. The clusters formed with the proposed MMD-UOT seem better separated (for example, the red and the cyan-colored clusters). For the experiment on the Office-Home dataset, the chosen hyperparameters for MMD-UOT are $\left( \lambda = 100, \text{IMQ kernel with } K^2 = 0.1 \right)$. For the VisDA-2017 dataset, the chosen hyperparameters for MMD-UOT are $\left( \lambda = 1, \text{IMQ kernel with } K^2 \text{ as } 10 \right)$.

For the validation phase on the Digits and the Office-Home datasets, we choose $\lambda$ from the set $\{1, 10, 100\}$ and $K^2$ from the set $\{0.01, 0.1, 10, 100, \text{median}\}$. For the validation phase on VisDA, we choose $\lambda$ from the set $\{1, 10, 100\}$ and $K^2$ from the set $\{0.1, 10, 100\}$.

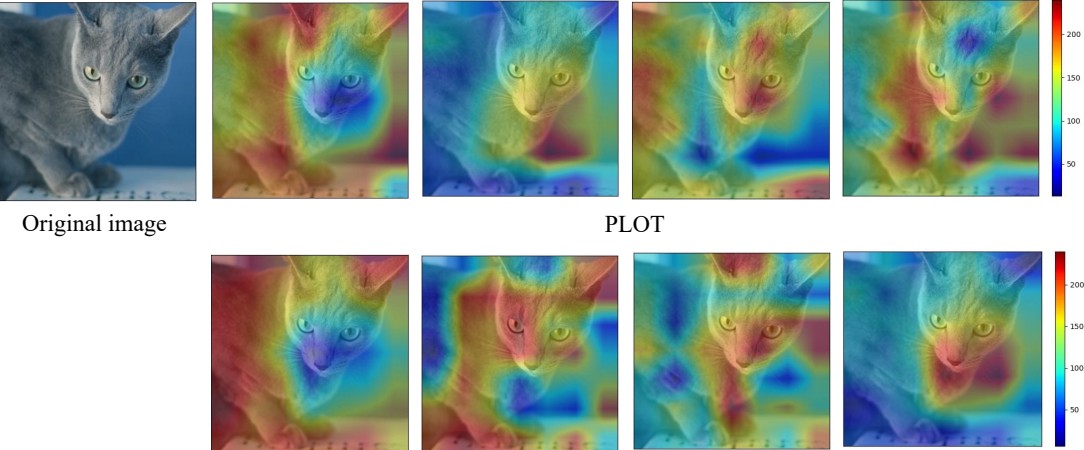

Figure 9: The attention maps corresponding to each of the four prompts for the baseline (PLOT) and the proposed method. The prompts learnt using the proposed MMD-UOT capture diverse attributes for identifying the cat (Oxford-Pets dataset): lower body, upper body, image background and the area near the mouth.

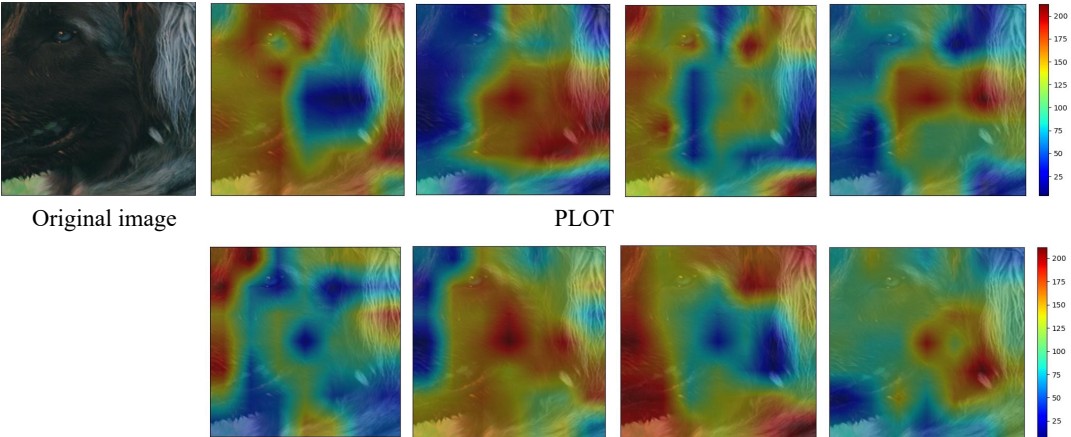

Figure 10: The attention maps corresponding to each of the 4 prompts for the baseline (PLOT) and the proposed method. The prompts learnt using the proposed MMD-UOT capture diverse attributes for identifying the dog (Oxford-Pets dataset): the forehead and the nose, the right portion of the face, the head along with the left portion of the face, and the ear.

## C.5 Prompt Learning

Let $\mathbf{F} = \{\mathbf{f}_m|_{m=1}^M\}$ denote the set of visual features for a given image and $\mathbf{G}_r = \{\mathbf{g}_n|_{n=1}^N\}$ denote the set of textual prompt features for class $r$. Following the setup in the PLOT baseline, an OT distance is computed between empirical measures over 49 image features and 4 textual prompt features, taking cosine similarity cost. Let $d_{OT}(\mathbf{x}, r)$ denote the OT distance between the visual features of image $\mathbf{x}$ and prompt features of class $r$. The prediction probability is given by $p(y = r|\mathbf{x}) = \frac{\exp\left((1-d_{OT}(\mathbf{x},r)/\tau)\right)}{\sum_{r=1}^{T}\exp\left((1-d_{OT}(\mathbf{x},r)/\tau)\right)}$, where $T$ denotes the total no. of classes and $\tau$ is the temperature of softmax. The textual prompt embeddings are then optimized with the cross-entropy loss. Additional results on Oxford-Pets (Parkhi et al., 2012) and UCF101 (Soomro et al., 2012) datasets are shown in Table 11.

Table 10: Hyperparameters (kernel type, kernel hyperparameter, $\lambda$) for the prompt learning experiment.

| Dataset | 1 | 2 | 4 | 8 | 16 |
|---|---|---|---|---|---|
| EuroSAT | (imq2, $10^{-3}$, 500) | (imq1, $10^4$, $10^3$) | (imq1, $10^{-2}$, 500) | (imq1, $10^4$, 500) | (rbf, 1, 500) |
| DTD | (imq1, $10^{-2}$, 10) | (rbf, 100, 100) | (imq2, $10^{-2}$, 10) | (rbf, $10^{-2}$, 10) | (rbf, 0.1, 1) |
| Oxford-Pets | (imq2, 0.01, 500) | (rbf, $10^{-3}$, 10) | (imq, 1, 10) | (imq1, 0.1, 10) | (imq1, 0.01, 1) |
| UCF101 | (rbf, 1, 100) | (imq2, 10, 100) | (rbf, 0.01, 1000) | (rbf, $10^{-4}$, 10) | (rbf, 100, $10^3$) |

Table 11: Additional Prompt Learning results. Average and standard deviation (over 3 runs) of accuracy (higher is better) on the $k$-shot classification task, shown for different values of shots ($k$) in the state-of-the-art PLOT framework. The proposed method replaces OT with MMD-UOT in PLOT, keeping all other hyperparameters the same. The results of PLOT are taken from their paper (Chen et al., 2023).

| Dataset | Method | 1 | 2 | 4 | 8 | 16 |
|---|---|---|---|---|---|---|
| EuroSAT | PLOT | $54.05 \pm 5.95$ | $64.21 \pm 1.90$ | $\mathbf{72.36 \pm 2.29}$ | $78.15 \pm 2.65$ | $82.23 \pm 0.91$ |
| | Proposed | $\mathbf{58.47 \pm 1.37}$ | $\mathbf{66.0 \pm 0.93}$ | $71.97 \pm 2.21$ | $\mathbf{79.03 \pm 1.91}$ | $\mathbf{83.23 \pm 0.24}$ |
| DTD | PLOT | $46.55 \pm 2.62$ | $\mathbf{51.24 \pm 1.95}$ | $56.03 \pm 0.43$ | $61.70 \pm 0.35$ | $65.60 \pm 0.82$ |
| | Proposed | $\mathbf{47.27 \pm 1.46}$ | $51.0 \pm 1.71$ | $\mathbf{56.40 \pm 0.73}$ | $\mathbf{63.17 \pm 0.69}$ | $\mathbf{65.90 \pm 0.29}$ |
| Oxford-Pets | PLOT | $87.49 \pm 0.57$ | $86.64 \pm 0.63$ | $88.63 \pm 0.26$ | $\mathbf{87.39 \pm 0.74}$ | $87.21 \pm 0.40$ |
| | Proposed | $\mathbf{87.60 \pm 0.65}$ | $\mathbf{87.47 \pm 1.04}$ | $\mathbf{88.77 \pm 0.46}$ | $87.23 \pm 0.34$ | $\mathbf{88.27 \pm 0.29}$ |
| UCF101 | PLOT | $\mathbf{64.53 \pm 0.70}$ | $66.83 \pm 0.43$ | $69.60 \pm 0.67$ | $74.45 \pm 0.50$ | $77.26 \pm 0.64$ |
| | Proposed | $64.2 \pm 0.73$ | $\mathbf{67.47 \pm 0.82}$ | $\mathbf{70.87 \pm 0.48}$ | $\mathbf{74.87 \pm 0.33}$ | $\mathbf{77.27 \pm 0.26}$ |
| *Avg acc.* | PLOT | $63.16$ | $67.23$ | $71.66$ | $75.42$ | $78.08$ |
| | Proposed | $\mathbf{64.38}$ | $\mathbf{67.98}$ | $\mathbf{72.00}$ | $\mathbf{76.08}$ | $\mathbf{78.67}$ |

Following the PLOT baseline, we use the last-epoch model. The authors empirically found that learning 4 prompts with the PLOT method gave the best results. In our experiments, we keep the number of prompts and the other neural network hyperparameters fixed. We only choose $\lambda$ and the kernel hyperparameters for prompt learning using MMD-UOT. For this experiment, we also validate the kernel type. Besides RBF, we consider two kernels belonging to the IMQ family: $k(x, y) = \left( \frac{1 + \|x - y\|^2}{K^2} \right)^{-0.5}$ (referred to as imq1) and $k(x, y) = (K^2 + \|x - y\|^2)^{-0.5}$ (referred to as imq2). We choose $\lambda$ from $\{10, 100, 500, 1000\}$ and kernel hyperparameter ($K^2$ or $\sigma^2$) from $\{1e-3, 1e-2, 1e-1, 1, 10, 1e+2, 1e+3\}$. The chosen hyperparameters are included in Table 10.

