# OpenReview forum: "MMD-Regularized Unbalanced Optimal Transport"
_TMLR — Accepted by TMLR_

### Review · Reviewer_rbdr · 2023-10-03

**Summary Of Contributions:**

In this submission, the authors propose a new variant of unbalanced optimal transport (UOT) called MMD-regularized UOT.
Compared with classic KL-divergence-regularized UOT, the proposed method suppresses the curse-of-dimensionality issue and thus may be more suitable for high-dimensional applications.
Additionally, the authors investigated the theoretical property of the MMD-UOT, demonstrating that 1) it is a valid metric, working as a surrogate for both MMD and Wasserstein distance; 2) it achieves an "interpolation" of MMD and Wasserstein distance --- changing the weight of the regularizer leads to different metrics.
In the aspect of the computation, an accelerated projected gradient descent algorithm is applied to compute the MMD-UOT with good empirical convergence, and the barycenter problem based on the MMD-UOT is considered.
Experiments on synthetic and real-world data show the rationality of the proposed method.

**Audience:**

Yes

**Claims And Evidence:**

Yes

**Requested Changes:**

See the weaknesses mentioned above.

**Strengths And Weaknesses:**

Strengths:

1) The proposed method is reasonable and solid, with theoretical supports, and the theory in the submission is proven in details.

2) In the aspect of sample complexity, the proposed MMD-UOT suppress the curve-of-dimensionality issue, which provides a good surrogate of Wasserstein distance and KL-divergence-based UOT in high-dimensional applications.

3) The paper is well-written, with a good logic flow, which is friendly to the readers without much background.

Weaknesses:

1) My main concern is about the experimental part of this work. In particular, the experimental part of this submission is weak to some degree. Firstly, the scale of the datasets is not large enough, which cannot demonstrate the usefulness of the proposed method in large-scale applications. Secondly, in each experiment, the baselines are insufficient --- only the MMD and the KL-UOT are considered. Especially in the generative modeling task, only toy datasets like MNIST are considered, and many OT-based generative models (e.g., WGAN [a] and its IPM extensions) are ignored.

2) Additionally, for the MMD-UOT based barycenter, besides the toy experiment shown in Figure 2, more practical applications and datasets should be considered.

3) In the section of preliminaries, to make the paper more friendly to readers, the authors should show the dual form of Wasserstein distance explicitly.

4) The Wasserstein autoencoder (WAE) in [b] should be added in the literature review and experimental part, which also considers an MMD regularizer when solving an OT problem approximately.

[a] Arjovsky, Martin, Soumith Chintala, and Léon Bottou. "Wasserstein generative adversarial networks." International conference on machine learning. PMLR, 2017.

[b] Tolstikhin, I., et al. "Wasserstein Auto-Encoders." 6th International Conference on Learning Representations (ICLR 2018). OpenReview. net, 2018.

Minors:

1) Figure 1 is meaningless in my opinion, which can be deleted in the revised paper.

2) In the implementation, does the proposed method require the source and target measures to have the same number of samples (i.e., the "m" in the paper)? If the answer is No, a footnote should be added to explain the case with different numbers of samples.

---

> ### Author Response · Authors · 2023-11-16
>
> We thank you for your valuable feedback. Following is our response.
> ### **Including experiments on larger datasets and comparison with more baselines**
>
> **Answer:** We have added experiments on larger datasets and with more baselines.
>
> **For additional domain adaptation experiments**, please refer to Section 5.5 of the revised draft. We now include the VisDA-2017 dataset (Recht et al, 2022), where the source and target domains have 152,397 and 55,388 images, respectively, with 12 object categories. We also include the Office-Home (Venkateswara et al, 2017) dataset, which has 15,500 images from four domains with 65 object categories. Office-Home dataset involves 12 adaptation tasks and is a popular choice for unsupervised domain adaptation (UDA) experiments. In addition to  $\epsilon$KL-UOT based JUMBOT (Fatras et al, 2021), we now include more  baselines for UDA which are based on a ResNet-50 backbone network: DANN (Ganin et al, 2015), CDANN-E (Long et al, 2017), DEEPJDOT (Damodar et al, 2018), ALDA (Chen et al, 2020a), ROT (Balaji et al, 2020) and OT-based state-of-the-art method BombOT (Nguyen et al, 2022). For both of these datasets, the proposed method achieves the best generalization performance.
>
> **For additional Two Sample Test experiments**, please refer to Appendix section C.2 of the revised draft. Following (Liu et al., 2020), we consider the task of verifying that the datasets CIFAR-10 (Krizhevsky, 2009) and CIFAR-10.1 (Recht et al., 2018) are statistically different.
> For this, we also include other popular baselines for two sample test: Classifier two-sample tests, including C2STS-S (Lopez-Paz \& Oquab, 2017) and C2ST-L (Chen \& Cloninger, 2019), Mean Embedding (ME) (Chwialkowski et al, 2015; Jitkrittum et al, 2016), Smooth characteristic functions (SCF) (Chwialkowski et al, 2015; Jitkrittum et al, 2016). The results (Table 8) show that the proposed MMD-UOT obtains the highest test power.
>
>
> ### **Including experiments related to MMD-UOT based barycenter**
>
> **Answer:** Section 5.3 of the main paper described our experiments and obtained results on an application of MMD-UOT based barycenter on cell population dynamics. OT Barycenter based interpolation has been explored for this application in existing works. Our results (Table 3) show that MMD-UOT based barycenter interpolation is more suitable than KL-UOT and MMD based barycenter for this application.
>
> ### **WGAN and its IPM extensions are not discussed**
>
> **Answer:** We had actually included a discussion in the Appendix B.17 of the original submission. Our analysis shows that the popular Spectral Normalized WGAN variant is a special case of MMD-UOT-based generative modelling.
>
>
> ### **On Wasserstein Autoencoder**
>
> **Answer:** We have now included a discussion on the Wasserstein Autoencoder in the related work section and presented details in Section C.6. In particular, we discuss the key differences with the WAE-MMD formulation and present an MMD-UOT-based WAE formulation. We discuss how WAE-MMD is a special case of MMD-UOT-based WAE and note its theoretical advantages.
>
> ### **Dual of Wasserstein in Preliminaries**
>
> **Answer:** We have discussed the dual of 1-Wasserstein in the Preliminaries. We have now highlighted this in the revised draft in the paragraph on Kantorovich metric on page 4.
>
>
> ### **Minor comments: Equal no. of samples in source-target?**
>
> **Answer:** We don't need an equal number of samples from the source and target. Our proofs detailed in the Appendices for the finite-sample case are general. We have added a footnote in the main paper on Page 7.
>
> We would be happy to discuss further.

---

### Review · Reviewer_LYuh · 2023-10-17

**Summary Of Contributions:**

In this work, a study of relaxed optimal transport is conducted where the marginal constraints are distributed into the objective in the form of penalties. While the standard choice to penalize these constraints is the phi-divergence (or the f-divergence as commonly known in the literature), this work studies the case of selecting the kernel-Maximum Mean Discrepancy (MMD). It is shown that the optimization objective admits a dual formulation which has conceptual appeal. Additionally, it is shown how this formulation can be computed with competitive estimation rates of O(m^{-1/2}). Experiments on various applications of machine learning are then demonstrated.

**Audience:**

Yes

**Broader Impact Concerns:**

As this work studies the theory and abstraction of machine learning application, there are no immediate ethical concerns in the work itself.

**Claims And Evidence:**

Yes

**Requested Changes:**

I think it would be better if the authors mention that the optimization relaxed considers not only probability measures but also unsigned Borel measures which is a much larger set. While this is implicit in the study of "unbalanced" optimal transport, it would be better if this more explicitly mentioned. I still think the paper should be accepted in its current state without this. Additionally, I think it would be better if the authors motivate unbalanced OT or how signed (unnormalized) Borel measures appear naturally in machine learning applications.

Page 2: "Such efficient estimators are particularly useful in machine learning applications, where typically only samples from the underlying measures are available"
It would be better if you can specify which machine learning applications are useful here.

I think you should include the objective of the Wasserstein Autoencoder [1] in the discussion of related work since in their work the "WAE-MMD" closely resembles the study where the marginal constraint is penalized with MMD. In particular, marginal constraint being penalized using MMD (in exactly WAE-MMD) was studied in [2] where MMD regularization leads to a variational objective involving RKHS functions  (see Equation (25) ) which is highly relevant to the dual form derived here.

In the proof of Theorem 4.1, I would request the authors to explain the details of using Sion's minimax Theorem. In particular, it is required that at least one of the spaces (either $\mathcal{R}^{+}(\mathcal{X} \times \mathcal{X})$ or $\mathcal{G}$) are compact. I would recommend the authors to clearly state this fact so the compactness assumption of $\mathcal{G}$ can be better understood.

If $\mathcal{X} \times \mathcal{X}$ is assumed to be compact then one can actually rely on the compactness of $\mathcal{R}^{+}(\mathcal{X} \times \mathcal{X})$ to make Sion's theorem work and a slightly different result would be present. I would encourage the authors to add such a discussion.


[1] Tolstikhin, I., Bousquet, O., Gelly, S., & Schoelkopf, B. (2017). Wasserstein auto-encoders. arXiv preprint arXiv:1711.01558.

[2] Husain, H., & Knoblauch, J. (2022, March). Adversarial interpretation of Bayesian inference. In International Conference on Algorithmic Learning Theory (pp. 553-572). PMLR.

**Strengths And Weaknesses:**

Strengths
- This paper makes a strong contribution of considering the popular MMD for constraint regularization which is a very practical choice.
- The dual form derived is very interesting as it translates to constraining the LP dual (Kantorovich-Rubenstein duality) in Optimal Transport to the function classes being in the RKHS and in the case of IPMs the weighted generator class.
- The algorithm presented is very practical and benefits from MMD.

Weaknesses:
- The theoretical result appears to be minimizing over unsigned Borel measures as opposed to probability measures and therefore only a generalization of this variant of OT.
- When using Sion's minimax theorems, compactness of the set is required in one of the sets. This makes the IPM result slightly limited.

---

> ### Author Response · Authors · 2023-11-13
>
> We thank you for your valuable feedback. Following is our response.
>
> ### **Relaxation to unsigned Borel measures**
> **Answer:** As in UOT settings, the measures $s_0,t_0$ may themselves be un-normalized, the transport plan is also allowed to be un-normalized (Chizat, 2017; Liero et al., 2018). This is now explicitly mentioned below Equation (4).
>
> ### **Sion's minimax for IPM generalization in Appendix -- Compactness assumption**
> **Answer:** Yes, the compactness of the generating set helps in the Sion's minimax exchange. The assumption of compactness of $\mathcal{G}$ is mentioned as Assumption A.1 and is mentioned in the statement of Theorem 4.1 in Appendix B.1. The use of this assumption in min-max exchange is explicitly mentioned in the last paragraph of the proof of this theorem discussed in Section B.1.
>
> While this assumption may be slightly restrictive for IPMs, it is satisfied by the generating set of MMD. We note that the main paper focuses on MMD-regularization and the IPM result is a generalization presented in the appendix.
>
> Finally, our proof of Theorem 4.1 for IPMs in Appendix B.1 cannot go through by assuming $\mathcal{R}^+(\mathcal{X}\times \mathcal{X})$ is compact (as suggested). This is because $\mathcal{R}^+(\mathcal{X}\times \mathcal{X})$ is a convex cone and hence unbounded by definition.
>
> ### **On Wasserstein Autoencoder**
> **Answer:** We have now included a discussion on the Wasserstein Autoencoder in the related work section and presented details in Section C.6. In particular, we discuss the key differences with the WAE-MMD formulation and present an MMD-UOT-based WAE formulation. We discuss how WAE-MMD is a special case of MMD-UOT-based WAE and note its theoretical advantages.
>
> ### **Motivating Unbalanced OT**
> **Answer:** As noted in the introduction, classical OT strictly enforces the marginals of the transport plan to be the source and target distributions. However, one would want to relax this constraint when the measures are noisy  (Frogner et al., 2015) or when the source and target are un-normalized. Hence, existing works (Chizat, 2017; Liero et al., 2018) have explored unbalanced optimal transport (UOT) formulations, which perform a regularization-based soft-matching of the transport plan’s marginals with the source and the target distributions. Section 3 (Related Work) of the main draft also discusses UOT formulations with phi-divergence regularization and TV regularization. KL-regularized UOT, in particular, has been explored in several applications (Fatras et al., 2021; Chen et al., 2020b; Nguyen et al., 2022; Arase et al., 2023; De Plaen et al., 2023) and has obtained state-of-the-art results.
>
> ### **Applications involving un-normalized measures**
> **Answer:** Such applications include gradient flows modeling growth phenomena, for instance, the Hele-Shaw model of tumor evolution (Chizat, 2017). Other applications are that of color transfer (with unequal mass color histograms), shape interpolation (where the shapes could be of very different scales), etc.
>
> ### **Machine learning applications where sample efficient estimators are useful**
>
> **Answer:** Typically, in machine learning (ML) applications, distributions are available only through samples. Hence, any OT-based ML application would benefit from sample efficient estimators. Such applications include hypothesis testing, domain adaptation, and model interpolation, to name a few. We have added such application examples on Page 2.
>
> We would be happy to discuss further.

---

### Review · Reviewer_uqvE · 2023-11-02

**Summary Of Contributions:**

The authors study unbalanced optimal transport as a tool to compare probability measures, which is a central task in machine learning. Unbalanced optimal transport (UOT) is a variant of the standard Kantorovich problem, wherein the hard constraints on the transport plans are replaced with soft regularization terms that impose the marginals of the transport plan to be close to the source and target measures. The authors study the analytical and statistical properties of UOT when the regularization terms are defined using the maximum-mean discrepancy (MMD) metric on probability measures corresponding to a reproducing kernel Hilbert space (RKHS). On the analysis side, the authors prove a duality formula for MMD-regularized UOT that is analogous to the duality for the traditional Kantorovich problem in optimal transport. Under certain assumptions on the kernel, they also prove that MMD-regularized UOT is a genuine metric on probability measures, and that it interpolates between the Kantorovich metric and a rescaled version of the MMD metric, depending on the regularization parameter. On the statistics side, the authors propose an estimator for MMD-regularized UOT when the measures are only accessible through data, which can be computed efficiently using convex optimization methods. They prove that their estimator converges to the true metric in the infinite-data limit with a dimension-independent convergence rate. The paper concludes with the discussion of several synthetic experiments, including the computation of barycenters.

**Audience:**

Yes

**Claims And Evidence:**

Yes

**Requested Changes:**

I suggest the authors make changes by addressing the following questions.

-	What are the benefits of employing MMD-UOT in practice as opposed to just using MMD? It seems like they have similar statistical properties, and computing MMD does not require one to solve an optimization problem.
-	I wonder if it is worth considering MMD-regularized UOT with kernels that are not characteristic (e.g., polynomial kernels), for the reasons described earlier. Perhaps you would get a worse statistical convergence rate, but the result might apply to a wider variety of measures. In addition, you would probably lose the metric properties, although it seems like for many applications it doesn’t really matter whether you are using a genuine metric. Have you considered this?
-	Is the IMQ kernel characteristic?
-	In the introduction, you write “Since MMD-UOT can approximate Wasserstein arbitrarily closely (as the regularization hyperparameter goes to $\infty$), our result can also be understood as a way of alleviating the curse of dimensionality problem in Wasserstein.” Do you have any quantitative estimates for the rate of convergence of MMD-UOT to the Wasserstein metric as the regularization parameter increases? If this rate suffers from a curse of dimensionality, then perhaps it is not quite true that using MMD-UOT breaks the curse of dimensionality for the Wasserstein metric.

**Strengths And Weaknesses:**

Strengths:

- The authors identify an interesting problem on which little has been done, namely the theoretical foundations of UOT with MMD-based regularization.
- Questions such as the structure of the underlying metric and statistical consistency of practical estimators are significant, since methods like UOT continue to be employed in real-world generative modeling tasks, and the authors do a nice job at explaining some of this story and providing rigorous proofs.
- I find their duality and interpolation results very interesting from the perspective of traditional optimal transport. It really explains what the metric induced by MMD-regularized UOT is capturing.
- The paper is generally well-written, making it easy to understand their main results.

Weaknesses:

- The main weakness is that the assumption of Theorem 4.10 seems quite unrealistic and thus precludes the application of their work to many interesting settings. In particular, the authors assume that the function $\eta(x,y) := \frac{\pi^{\ast}(x,y)}{s_0(x) t_0(y)}$ belongs to $\mathcal{H}_k \otimes \mathcal{H}_k$, where $s_0$ and $t_0$ are the two measures we wish to compare, $\pi^{\ast}(x,y)$ is the optimal transport plan between $s_0$ and $t_0$, and $\mathcal{H}_k$ is the RKHS. They also assume that the cost function $c(x,y)$ used to define their optimal transport plan belongs to $\mathcal{H}_k \otimes \mathcal{H}_k$. In addition, the error of their estimator has a constant of $\|\eta\|_k \cdot \|c\|_k$, the RKHS norms of $\eta$ and $c$.

- My concern is that this assumption assumes far too much regularity on the target measures and cost function, since the RKHS of a characteristic kernel can be quite small (evidenced, for example, by the $O(n^{-1/2})$ rate for the uniform law of large numbers that the paper cites). It does not allow for non-absolute continuity of the distributions, which is a common assumption in machine learning. The authors state that the assumption is mild because the RKHS is dense in the space of continuous functions. However, they do not provide any quantitative approximation results, i.e., they do not say, for given $f \in C(\mathcal{X}), $\epsilon > 0$, what the RKHS norm of a function $f_{\epsilon} \in \mathcal{H}^k$ that satisfies $\|f-f_{\epsilon}\|_{\infty} < \epsilon$ will be. Also, it is not clear why the cost would belong to the RKHS.

I think the paper would be much improved if the authors:

- Were able to relax the assumptions on $s_0$, $t_0$, and $c$ or weaken the dependence of the sample complexity bound on the RKHS norms.
- Put more classical/realistic assumptions on the target measures and consider approximation as an additional source of error.

If the authors insist that it is a ‘mild’ assumption, then they should provide at least one test example where the assumptions are satisfied. Otherwise, it should be mentioned that the assumption is restrictive, and future improvements should be discussed in the conclusion.

I also have some minor issues with the presentation. First, the notions of characteristic and universal kernels are used throughout the paper but never defined. It would improve readability to have a brief discussion of what kernels are included/excluded from the characteristic assumption, even if it is delayed to the appendix. Also, I think you should mention what kernels you use for your experiments in the main text.

---

> ### Author Response · Authors · 2023-11-13
>
> We thank you for your valuable feedback.
>
> ### **What are the benefits of employing MMD-UOT in practice as opposed to just using MMD?**
>
> **Answer:** The key advantage of employing MMD-UOT is the phenomenon of lifting the ground-metric geometry to that over distributions.
> One such result is visualized in Figure 2, where the MMD-based-interpolate of the two unimodal distributions comes out to be bimodal. This is because MMD's interpolation is the (literal) average of the source and the target densities, irrespective of the kernel. This has been well-established in the literature (Bottou et al., 2017). On the other hand, MMD-UOT obtains a unimodal barycenter. This is a `geometric' interpolation that captures the characteristic aspects of the source and the target distributions. As observed in our experiments (Section 5), MMD-UOT outperforms MMD in various applications.
>
> Another feature of MMD-UOT (and, in general OT-based approaches) is that we obtain a transport plan between the source and the target points, which can be used for various alignment-based applications (e.g., cross-lingual word mapping (Alvarez-Melis and Jakkola, 2018 EMNLP)). On the other hand, it is unclear how MMD can be used to align the source and target data points.
>
> This makes MMD-UOT more applicable than MMD across ML problems. We have included this point in Section 3 (last paragraph).
>
> ### **Using non-characteristic kernels**
>
> **Answer:** Using a non-characteristic kernel for MMD will lead to situations where very different distributions have zero MMD between them. For example, if we use a linear kernel, then all distributions with the same mean will have MMD as zero between them. Hence, in addition to losing metric properties, MMD with a non-characteristic kernel also seems unsuited as a loss function. Hence, we did not explore this direction.
>
>
> ### **Is the IMQ kernel characteristic?**
>
> **Answer:** Yes, IMQ is a characteristic kernel (Sriperumbudur et al., 2011).
>
> ### **Quantitative estimates for the rate of convergence of MMD-UOT to the Wasserstein metric as the regularization parameter increases.**
>
> **Answer:** Our sample complexity bound in Corollary 4.6 shows the dependence on $\lambda$ as $\mathcal{O}(\frac{\lambda}{\sqrt{m}})$. Let $\lambda$ grows as $\mathcal{O}(m^{1/4})$, our sample complexity bound would then be $\mathcal{O}(m^{-1/4})$. Hence, in this setting, our estimator is consistent, and the rate of convergence is indeed dimension-free.
>
> *More responses to follow.*

---

> > ### Author Response · Authors · 2023-11-16
> >
> > ### **Assumption of Theorem 4.10 seems quite unrealistic**
> >
> > **Answer:** Following your suggestions, we were able to relax the assumption of the cost function belonging to the Hilbert space. Now, we only assume that the cost function is continuous. The new theorem statement and its proof are now presented in the revised draft. In addition, we have removed the phrase `mild assumption' from the theorem statement.
> >
> > ### **Discussion on characteristic and universal kernels**
> >
> > **Answer:** We have added a discussion on the characteristic and universal kernels in the Section 2 of the main paper.

---

### Decision · Action_Editor_GYYB · 2023-12-22

**Recommendation:** Accept as is

**Comment:**

Initial reviews praised several aspects of the submissions (theoretical contributions, relevance, scope) and suggested improvements on others (relaxing the assumptions, expanding the numerical experiments). The authors very convincingly addressed all these points in their rebuttals and all three reviewers unanimously recommend acceptance, which I am happy to follow. The paper contributes an interesting strategy to study unbalanced optimal transport problems through regularisation with the MMD. Congratulations on a fine piece of work!

**Audience:**

Optimal transport is a fairly active topic in machine learning and I expect the paper to be relevant to a part of the TMLR audience. I have no doubt this work is relevant for TMLR.

**Claims And Evidence:**

All reviewers praise the technical contributions made by the authors, and all claims are supported by correct proofs.